# Blessings of Many Good Arms in Multi-Objective Linear Bandits

## Abstract

Multi-objective decision-making is often deemed overly complex in bandit settings, leading to algorithms that are both complicated and frequently impractical. In this paper, we challenge that notion by showing that, under a novel *goodness of arms* condition, multiple objectives can facilitate learning, enabling simple near-greedy methods to achieve sub-linear Pareto regret. To our knowledge, this is the first work to demonstrate the effectiveness of near-greedy algorithms for multi-objective bandits and also the first to study the regret of such algorithms for parametric bandits in the absence of context distributional assumptions. We further introduce a framework for *objective fairness*, supported by strong theoretical and empirical evidence, illustrating that multi-objective bandit problems can become both simpler and more efficient.

## 1 Introduction

Multi-objective decision-making problems have become increasingly prevalent in today's complex, real-world applications. From recommendation systems to robotics, decision-makers must often optimize multiple, potentially conflicting objectives simultaneously. This setting naturally gives rise to *multi-objective bandit problems* (Drugan & Nowe, 2013; Turgay et al., 2018; Lu et al., 2019; Xu & Klabjan, 2023; Kim et al., 2023; Cheng et al., 2024; crepon et al., 2024; Zhang, 2024), which generalize the single-objective bandit framework by incorporating several objectives. Although this extension may seem conceptually straightforward, balancing exploration and exploitation across multiple objectives significantly increases the complexity of the problem.

To address multi-objective bandit problems, most existing approaches focus on achieving Pareto optimality (Drugan & Nowe, 2013; Yahyaa & Manderick, 2015; Tekin & Turgay, 2018; Turgay et al., 2018; Lu et al., 2019; Kim et al., 2023; Cheng et al., 2024). However, these methods often involve updating empirical Pareto fronts in each round, leading to substantial computational overhead and limiting their suitability for often real-time and sequential decision-making applications.

While multi-objective problems are generally more complex than their single-objective counterparts, it is natural to ask whether multiple objectives could, in some cases, *facilitate* learning rather than hinder it. Formally, we pose the following research question:

*Can the presence of multiple objectives actually facilitate learning rather than hinder it?*

A priori, the answer is not always *yes*. Nonetheless, there may be scenarios in which multiple objectives can be leveraged to achieve simpler, more efficient solutions. A positive answer to this question could reshape our perspective on multi-objective problems: instead of always resorting to increasingly complex methods, we might exploit a simpler, near-greedy approach to handle multiple objectives more effectively. To our knowledge, this perspective has been largely overlooked, perhaps because it appears counterintuitive that adding objectives could simplify the problem. Consequently, an important research direction is to identify the precise conditions under which multi-objective bandit problems become admissible—even potentially advantageous—for simple algorithms.

In this work, we show that the existence of *good arms for multiple objectives* can enable simpler near-greedy algorithms to achieve strong performance. Such "goodness" means that, for each objective, there is at least one arm that performs sufficiently well (and these arms may differ across objectives), a scenario commonly observed in practice. We show that this condition leads to what we

call *free exploration*—the ability to collect informative feedback without incurring extra exploration cost. Concretely, we propose a novel near-greedy algorithm, MOG (Algorithm 1), and its variants, and prove that, under the suitable goodness assumption, they attain a regret bound of $\widetilde{\mathcal{O}}(\sqrt{T})$. To our knowledge, these are the first *explosion-free* algorithms for multi-objective bandits.

From a broader perspective, our work is related to the literature on exploration-free linear bandits in the single-objective setting (Kannan et al., 2018; Raghavan et al., 2018; Bastani et al., 2021; Kim & Oh, 2025), where greedy (or near-greedy) strategies can be efficient if the contexts are sufficiently diverse. However, our analysis here differs crucially: we do not rely on any stochasticity or diversity in the contexts (features). Notably, our algorithms perform effectively even in *fixed* feature settings, marking the first result (to our knowledge) in which a greedy-type algorithm achieves *no-regret* without any diversity assumptions in parametric bandits. This new insight suggests that multiple objectives can sometimes replace or augment the role that context diversity plays in single-objective settings, which can be of independent interest.

We evaluate our proposed algorithms both theoretically and empirically, analyzing their regret performance and introducing a novel fairness criterion to ensure that no single objective is neglected. Our results show that these simple and efficient methods not only exhibit strong performance but also satisfy fairness guarantees—representing, to the best of our knowledge, the first theoretical result on fairness in multi-objective bandits. Consequently, our work opens new perspectives in multi-objective bandit research by providing a new class of algorithms and offering stronger guarantees on both Pareto optimality and fairness.

## 1.1 CONTRIBUTIONS

- We present and rigorously analyze a novel, sufficient condition on the *goodness of arms* (definition 5) under which near-greedy algorithms achieve statistical efficiency in multi-objective bandit problems *without* relying on the commonly assumed context distributional assumptions in the greedy bandit literature (Kannan et al., 2018; Raghavan et al., 2018; Bastani et al., 2021; Kim & Oh, 2025). Under this condition, *free exploration* can be effectively leveraged. Notably, free exploration persists even in fixed context settings, rather than just stochastic environments. Our key (and somewhat surprising) insight is that having multiple objectives can *enhance* rather than hinder the learning process.

- We propose and analyze three practical algorithms, MOG, MOG-R, and MOG-WR, showing that under the goodness assumption, each algorithm attains $\widetilde{\mathcal{O}}(\sqrt{T})$ Pareto regret, where $T$ is the total number of rounds. Crucially, our proposed algorithms do not require constructing or maintaining empirical Pareto fronts, significantly reducing computational overhead compared to many existing algorithms.

- We introduce the notion of *objective fairness* (definition 13) as a criterion for evaluating multi-objective bandit algorithms. We prove that the MOG and MOG-R algorithm satisfies objective fairness, while the MOG-WR algorithm satisfies a general version of this criterion (definition 6). To our knowledge, this is the first theoretical analysis of fairness in multi-objective bandit problems.

- Through extensive numerical experiments, we demonstrate that MOG, MOG-R and MOG-WR consistently outperform existing multi-objective methods across a wide range of scenarios. These results empirically validate our theoretical claims.

## 1.2 RELATED WORK

The multi-objective bandit problem, an extension of the single-objective bandit framework that captures real-world scenarios with multiple conflicting optimization objectives, was first introduced by Drugan & Nowe (2013). They proposed two approaches using the UCB algorithm: one based on Pareto optimality and the other on scalarization. While the scalarization approach simplifies the problem by reducing it to a single-objective one (Drugan & Nowe, 2013; Yahyaa & Manderick, 2015; Zhang, 2024), the Pareto optimality approach treats all objectives equally, without making any assumptions about their interrelationships. This second approach inspired numerous studies on multi-objective bandits focused on Pareto efficiency (Turgay et al., 2018; Tekin & Turgay, 2018; Lu et al., 2019; Xu & Klabjan, 2023; Kim et al., 2023; Cheng et al., 2024; crepon et al., 2024).

Recent advancements have extended the multi-objective bandit framework to linear contextual settings. Lu et al. (2019) established theoretical regret bounds for the UCB algorithm within the generalized linear bandit framework. Kim et al. (2023) explored Pareto front identification in linear bandit settings. Additionally, Cheng et al. (2024) introduced two algorithms stochastic linear bandits under a hierarchy-based Pareto dominance condition. In a different approach, Zhang (2024) proposed a hypervolume scalarization method in stochastic linear bandit settings and analyzed hypervolume regret, a metric that measures how well the Pareto front is generated.

While these works made important strides, they largely overlook the potential for free exploration that can arise from the existence of good arms for each objective, particularly in the absence of the diversity in context stochasticity. Recent research on single-objective linear contextual bandits with stochastic contexts has shown that when context diversity is sufficiently high, greedy algorithms can achieve near-optimal regret bounds in terms of the total number of rounds. (Bastani et al., 2021; Kannan et al., 2018; Raghavan et al., 2018; Kim & Oh, 2025). However, the extension of these results to multi-objective bandits has been limited by a diversity assumption on the context distribution, leaving a gap in understanding how exploration can occur without the this assumption.

Our work addresses this gap by focusing on free exploration driven by good arms for different objectives, even in the absence of context stochasticity. While Bayati et al. (2020) demonstrated that greedy algorithms perform well in non-contextual single-objective settings when the number of arms is large, they relied on a $\beta$-regularity assumption related to the reward distribution. In contrast, we introduce the concept of $\gamma$-*goodness* (Assumption 3), which generalize the notion of $\beta$-regularity to feature spaces in the multi-objective setting. Unlike Bayati et al. (2020), which provided only Bayesian regret bounds—a weaker notion of regret than the frequentest regret, we rigorously establish the frequentest regret bounds for our proposed algorithms, MOG, MOG-R, and MOG-WR, under this generalized goodness assumption and for multi-objective problem settings. This is the first time that a theoretical guarantee has been provided for exploration-free algorithms in multi-objective linear bandits, without relying on the diversity assumption on context distribution, which is a significant departure from existing literature.

## 2 PROBLEM SETTINGS

### 2.1 NOTATIONS

We denote by $[n] := \{1, \ldots, n\}$ for $n \in \mathbb{N}$. For a vector $x \in \mathbb{R}^d$, we use $\|x\|_2$ and $\|x\|_A = \sqrt{x^\top A x}$ to denote to denote the $l_2$ norm and the weighted norm of $x$ induced by a positive definite matrix $A \in \mathbb{R}^{d \times d}$. We define the $d$-dimensional ball $\mathbb{B}_R^d = \{x \in \mathbb{R}^d \mid \|x\|_2 \leq R\}$. Finally, $\mathbb{1}\{\text{condition}\}$ means the indicator function that takes the value 1 if the condition is true and 0 otherwise.

### 2.2 MULTI-OBJECTIVE LINEAR BANDITS

In each round $t \in [T]$, each feature vector $x_i \in \mathbb{R}^d$ for $i \in [K]$ is associated with stochastic reward $y_{i,m}(t)$ for objective $m \in [M]$ with mean $x_i^\top \theta_m^*$ where $\theta_m^* \in \mathbb{R}^d$ is a fixed, unknown parameter. After the agent pulls an arm $a(t) \in [K]$, the agent receives a stochastic reward vector $y_{a(t)}(t) = \left(y_{a(t),1}(t), \ldots, y_{a(t),M}(t)\right) \in \mathbb{R}^M$ as a bandit feedback, where $y_{a(t),m}(t) = x_{a(t)}^\top \theta_m^* + \eta_{a(t),m}(t)$ and $\eta_{a(t),m}(t) \in \mathbb{R}$ is zero-mean noise for objective $m \in [M]$. To simplify notation, we denote by $x(t) := x_{a(t)}$ and $y(t) := y_{a(t)}(t)$, the selected arm vector in round $t$ and its rewards, respectively, with slight notational overloading. We assume that for all $m \in [M]$, $\eta_{a(t),m}(t)$ is conditionally $\sigma^2$-sub-Gaussian for some $\sigma > 0$, i.e., for all $\lambda \in \mathbb{R}$, $\mathbb{E}[e^{\lambda \eta_{a(t),m}(t)} | \mathcal{F}_{t-1}] \leq \exp\left(\lambda^2 \sigma^2 / 2\right)$ where $\mathcal{F}_t$ is the $\sigma$-algebra generated by $\left(\{x(s)\}_{s \in [t+1]}, \{a(s)\}_{s \in [t]}, \{y(s)\}_{s \in [t]}\right)$.

While we present our problem setting in the fixed-feature setup for clarity of exposition—highlighting our main idea of free exploration without relying on the context distributional diversity assumption—we also provide results under a varying-context setting in Appendix G.

#### 2.2.1 PARETO REGRET METRIC

In this work, we use the notion of Pareto regret (Drugan & Nowe, 2013; Tekin & Turgay, 2018; Turgay et al., 2018; Lu et al., 2019; Xu & Klabjan, 2023; Kim et al., 2023; Cheng et al., 2024;

crepon et al., 2024) as the performance metric for multi-objective bandit algorithms. Before we formally define the Pareto regret, we introduce the notions of Pareto order and Pareto front.

**Definition 1** (Pareto order). *For $u = (u_1, \ldots, u_M)$, $v = (v_1, \ldots, v_M) \in \mathbb{R}^M$, the vector $u$ dominates $v$, denoted by $v \prec u$, if and only if $v_m \leq u_m$ for all $m \in [M]$, and there exists $m' \in [M]$ such that $v_{m'} < u_{m'}$. We use the notation $v \nprec u$ when $v$ is not dominated by $u$, and $u \parallel v$ when $u$ and $v$ are incomparable, i.e., either $u$ or $v$ is not dominated by the other, respectively.*

**Definition 2** (Pareto front). *Let $\mu_i \in \mathbb{R}^M$ be the expected reward vector of arm $i \in [K]$. Then, arm $i$ is Pareto optimal if and only if $\mu_i$ is not dominated by $\mu_{i'}$ for all $i' \in [K]$. The Pareto front is the set of all Pareto optimal arms.*

**Definition 3** (Pareto regret). *We denote **Pareto suboptimality gap** $\Delta_i$ for arm $i \in [K]$ as the infimum of the scalar $\epsilon \geq 0$ such that $\mu_i$ becomes Pareto optimal arm after adding $\epsilon$ to all entries of its expected reward. Formally,*

$$\Delta_i := \inf \{\epsilon \mid (\mu_i + \epsilon) \nprec \mu_{i'}, \forall i' \in [K]\}.$$

*Then, the cumulative **Pareto regret** is defined as $\mathcal{PR}(T) := \sum_{t=1}^{T} \mathbb{E}[\Delta_{a(t)}]$, where $\mathbb{E}[\Delta_{a(t)}]$ represents the expected Pareto suboptimality gap of the arm pulled at round $t$.*

The goal of the agent is to minimize the cumulative Pareto regret while ensuring fairness across objectives, which is described in the next section.

### 2.2.2 OBJECTIVE FAIRNESS

**Beyond Pareto regret.** Pareto regret minimization is often a central goal in multi-objective bandit algorithms, but it does not fully capture the essence of the multi-objective problem. Paradoxically, focusing solely on Pareto regret minimization itself allows algorithms to optimize for a single specific objective, potentially neglecting others. Specifically, an algorithm that behaves with respect to a single-objective bandit problem may perform just as well in the Pareto optimal sense (Xu & Klabjan, 2023), hence defeats the purpose of the multi-objective problem. Therefore, multi-objective bandit algorithms should aim to balance multiple objectives, typically incorporating additional considerations such as fairness, alongside Pareto regret minimization.

**Existing fairness criterion (Drugan & Nowe, 2013).** In multi-objective bandits, how fairly an algorithm handles multiple objectives is considered an important factor. Fairness in multi-objective bandits was first introduced by Drugan & Nowe (2013), who defined it as how evenly the Pareto front is sampled (Definition 7). However, this definition requires tracking the selection frequency of each true Pareto optimal arm, making it unsuitable for theoretical analysis. Many previous studies have mentioned fairness in the selection process, but, to the best of our knowledge, none has provided a theoretical analysis of fairness (Yahyaa & Manderick, 2015; Turgay et al., 2018; Lu et al., 2019).

Furthermore, in practice, the fairness principle requires multi-objective algorithms to compute the empirical Pareto front at each arm selection, resulting in significant computational overhead (Drugan & Nowe, 2013; Yahyaa & Manderick, 2015; Turgay et al., 2018; Lu et al., 2019). Specifically, algorithms that construct the empirical Pareto front in each round incur a time complexity of $\mathcal{O}(K^2)$ per round. This indicates that such algorithms may encounter scalability challenges in real-world applications involving a significantly large arm set.

**Objective fairness.** To address these limitations, we propose a new notion of fairness in multi-objective bandit problems. The fairness we introduce provides theoretical guarantees without imposing additional computational overhead on the algorithms. We present a new notion of fairness based on the near-optimality in each objective in definition 13.

**Definition 4** (Objective fairness). *Let $\mu_{i,m}$ be the expected reward of arm $i$ for objective $m$, $a_m^*$ be the arm that has the highest expected reward for objective $m$, and $\mu_m^* := \mu_{a_m^*, m}$. For all $\epsilon > 0$, we define **the objective fairness index** $\mathrm{OFI}_{\epsilon,T}$ of an algorithm as*

$$\mathrm{OFI}_{\epsilon,T} := \min_{m \in [M]} \left( \frac{1}{T} \mathbb{E}\left[ \sum_{t=1}^{T} \mathbb{1}\{\mu_m^* - \mu_{a(t),m} < \epsilon\} \right] \right).$$

*Then, we say that an algorithm satisfies **objective fairness** if for a given $\epsilon$, there exists a positive lower bound $L_\epsilon$ such that $\lim_{T \to \infty} \mathrm{OFI}_{\epsilon,T} \geq L_\epsilon$.*

Intuitively, our perspective of objective fairness makes sure that the algorithms consistently consider all optimal arms for each objective. The objective fairness index measures the proportion of rounds in which the $\epsilon$-optimal arms are selected for the least selected objective. Therefore, objective fairness is an asymptotic concept that ensures the consistent selection of near-optimal arms for each objective as time progresses. Conversely, if $\lim_{T\to\infty} \text{OFI}_{\epsilon,T} \to 0$, this implies that the algorithm ultimately does not consider optimal arms for at least one objective.

Additionally, we extend Definition 13 to consider not only the directions of objective parameters but also the weighted sum of their directions, introducing the notion of *generalized objective fairness* (definition 6). This extended definition ensures that optimal arms on the positive side of the Pareto front (Figure 2) are selected in proportion to the total rounds $T$. A detailed discussion, including the improvement over Drugan & Nowe (2013)'s fairness notion, is provided in Appendix B.

**Pareto front approximation.** We argue that algorithms pursuing (generalized) objective fairness can address many challenges in real-world problems more efficiently than traditional approaches. A major advantage of these algorithms is that they eliminate the need to construct the empirical Pareto front at each iteration, which is typically required by traditional methods to ensure fairness. Although these algorithms do not approximate the Pareto front in every round, they allow for on-demand approximation of the Pareto front by estimating the parameters of each objective.

**Lemma 1** (Connection from objective parameter estimation to Pareto front approximation). *Suppose $\|x_i\|_2 \leq 1$ holds for all arms $i \in [K]$. Let $\widetilde{\theta}_m(t)$ be an estimator for $\theta_m^*$ for $m = 1, \ldots, M$, and we define the empirical Pareto front $\widetilde{\mathcal{O}}(t)$ with respect to the estimated expected reward $\left(x_i^\top \widetilde{\theta}_1(t), \ldots, x_i^\top \widetilde{\theta}_M(t)\right)$ for each arm $i \in [K]$ in round $t$. If $\|\widetilde{\theta}_m(t) - \theta_m^*\|_2 < \epsilon$ holds for all $m \in [M]$, then, for all arms $i \in \widetilde{\mathcal{O}}(t)$, the suboptimality gap satisfies $\Delta_i \leq 2\epsilon$.*

# 3 PROPOSED ALGORITHM

## 3.1 MULTI-OBJECTIVE GREEDY (MOG) ALGORITHM

We propose a new algorithm named the MOG algorithm, that greedily selects arms based on a target objective in each round. The default setting for determining the target objective is a round-robin approach, where in each round $t$ we use the modulo operator (mod) to cycle through the objectives as targets (Line 3). Initially, the algorithm greedily selects arms based on the initial parameters $\beta_1, \ldots, \beta_M$ until the minimum eigenvalue of the Gram matrix $V_{t-1} = \sum_{s=1}^{t-1} x(s)x(s)^\top$ exceeds a certain threshold $B$ (Line 5). After the initial rounds, the algorithm greedily selects arms iteratively using the OLS estimators $\hat{\theta}_m(t)$ of $\theta_m^*$ (Line 8).

This simple approach is presented without loss of generality: various selection strategies can also be employed, depending on specific problem requirements. For instance, if certain objectives are more (or less) important, their selection frequency can be adjusted accordingly. Alternatively, the target objective can be chosen randomly, an approach we denote by MOG-R (Algorithm 2), and further describe in Appendix E.

The MOG algorithm is easy to implement and generic, making it easily extendable. While Algorithm 1 represents the simplest yet efficient approach, we propose a more general algorithm, MOG-WR (Algorithm 3). This algorithm selects arms greedily based on the direction of the weighted sum of the estimated objective parameters. If the bandit problem has a convex Pareto front, MOG-WR can fully explore the entire Pareto front. In general, MOG-WR explores the positive side of the Pareto front (since one does not need to explore non-positive sides). Detailed descriptions and analyses of MOG-WR can be found in Appendix F.

If the feature vectors do not span $\mathbb{R}^d$, the minimum eigenvalue of the Gram matrix in Line 4 remains zero. In such cases, the MOG algorithm can be implemented using the more general formulation presented in Algorithm 4. Further details are provided in Appendix I.

Most existing algorithms regarding Pareto efficiency construct the empirical Pareto front on each round, resulting in complex algorithm structure and less practicality. Compared to other multi-objective bandit algorithms, our algorithms are very easy to implement and have significantly lower computational overhead. Aside from these advantages, surprisingly, our simple algorithms can

---

**Algorithm 1** Multi-Objective Greedy algorithm (`MOG`)

---

**Require:** Total rounds $T$, Eigenvalue threshold $B$
 1: Initialize $V_0 \leftarrow 0 \times I_d$, and $\beta_1, \ldots, \beta_M \in \mathbb{R}^d$
 2: **for** $t = 1$ **to** $T$ **do**
 3:     Select the target objective $m \leftarrow t \bmod M$ {If $m = 0$, then $m \leftarrow M$}
 4:     **if** $\lambda_{\min}(V_{t-1}) < B$ **then**
 5:        Select action $a(t) \in \arg\max_{i \in [K]} x_i^\top \beta_m$
 6:     **else**
 7:        Update the OLS estimators $\hat{\theta}_1(t), \ldots, \hat{\theta}_M(t)$
 8:        Select action $a(t) \in \arg\max_{i \in [K]} x_i^\top \hat{\theta}_m(t)$
 9:     **end if**
10:    Observe $y(t) = \big(y_{a(t),1}(t), \ldots, y_{a(t),M}(t)\big)$
11:    Update $V_t \leftarrow V_{t-1} + x(t)x(t)^\top$
12: **end for**

---

achieve theoretical performance guarantees, which are typically obtained by more complex algorithms, when good arms exist for each objective.

### 3.2 Free exploration induced by many good arms

The `MOG` algorithm (Algorithm 1) is built on the insight that exploration can arise naturally, even when the algorithm is focused solely on exploitation, as long as the multi-objective bandit problem has many good arms. In most existing multi-objective bandit studies, as the number of objectives increases, the problem setup becomes more complex, leading to more sophisticated algorithms, particularly in comparison to single-objective bandits.

However, we observe a surprising and beneficial effect: as the number of objectives increases, unlike the single-objective case, the multiple directions of good arms can naturally induce free exploration. This allows simple near-greedy algorithms like `MOG` to achieve statistical efficiency (see Theorem 1).

The core idea is that, for each objective, rounds where greedy selections for other objectives can simultaneously function as exploration rounds for the remaining objectives. In each round, exploitation occurs for one objective, while inherently providing exploration for the others. This mechanism enables automatic exploration without incurring additional Pareto regret, which offers a significant performance advantage.

This phenomenon is intuitive, yet it has not been rigorously examined in multi-objective settings so far. Our work is the first to formalize the conditions under which natural exploration can occur in the presence of good arms for multiple objectives, paving the way for simpler and more efficient algorithms for multi-objective bandit problems.

## 4 Analysis

In this section, we analyze the algorithm `MOG` from the perspective of Pareto regret and objective fairness. Our analysis is established in the fixed feature setup to expose our main idea clearly, however, we also present similar results in a stochastic environment in Appendix G. We start with a boundedness assumption similar to those used in the linear bandit literature (Abbasi-Yadkori et al., 2011; Chu et al., 2011b; Agrawal & Goyal, 2013; Abeille & Lazaric, 2017; Li et al., 2017).

**Assumption 1** (Boundedness). *For all $i \in [K]$ and $m \in [M]$, $\|x_i\|_2 \leq 1$ and $\|\theta_m^*\|_2 = 1$.*

Assumption 1 is used to make a clean analysis for convenience and the first part of it is in fact standard in bandit literature. Notably, we can obtain a regret bound of the proposed algorithm that differs by at most a constant factor under the conditions $\|x_i\|_2 \leq x_{\max}$ and $l \leq \|\theta_m^*\|_2 \leq L$ for all $i \in [K]$ and $m \in [M]$. Our analysis focuses on the insights that multiple objectives may enhance learning under certain regularity (e.g. goodness of arms in Definition 5) rather than always posing hindrance. In light of these insights, the lower bound $l$ represents the minimum contribution of each objective. We will later discuss how to extend our analysis to arbitrary bounds for feature vectors and objective parameters in Appendix H.

As stated earlier in the Introduction and Section 3.2, we are interested in the problem setting where there exist good arms for multiple objectives. We start with a simple condition that there are enough objectives to span the feature space without loss of generality.

**Assumption 2.** *We assume $\theta_1^*, \ldots, \theta_M^*$ span $\mathbb{R}^d$.*

It is important to note that Assumption 2 is used without loss of generality. We can actually relax Assumption 2 so that $\theta_1^*, \ldots, \theta_M^*$ span the space of feature vectors, $span(\{x_1, \ldots, x_K\})$ (see details in Appendix I). That is, it can be sufficient to assume that $\theta_1^*, \ldots, \theta_M^*$ span a strict subspace of $\mathbb{R}^d$ if the feature vectors span such a subspace. Yet, for clear exposition of our main idea, we work with Assumption 2 and define $\lambda := \lambda_{\min}(\frac{1}{M} \sum_{m=1}^M \theta_m^* (\theta_m^*)^\top)$, which is positive under Assumption 2.

Next, we introduce the $\gamma$-goodness condition of arms with feature vectors in multi-objective linear bandits. In brief, this condition ensures the presence of good arms in every objective direction.

**Definition 5** (Goodness of arms). *For fixed $\gamma \in (0, 1]$, we say that the feature vectors of the arms $\{x_1, \ldots, x_K\}$ satisfy $\gamma$-goodness condition if there exists $\alpha > 0$ such that*

$$\text{for all } \beta \in \mathbb{B}_\alpha^d(\theta_1^*) \cup \ldots \cup \mathbb{B}_\alpha(\theta_M^*), \text{ there exists } k \in [K] \text{ such that } x_k^\top \frac{\beta}{\|\beta\|_2} \geq \gamma,$$

*and denote such $x_k$ as the $\gamma$-good arm for direction $\beta$.*

**Assumption 3** (Goodness). *The feature vectors $\{x_1, \ldots, x_K\}$ satisfy $\gamma$-goodness for some $\gamma \geq 1 - \frac{\lambda^2}{18}$.*

Assumption 3 states that there exists at least one $\gamma$-good arm for directions in the neighborhoods of objective parameters. We relax the $\gamma$-goodness condition by relaxing the requirement of the existence of $\gamma$-good arms and instead allowing a positive probability of their existence in a stochastic setting (see Assumption 4 in Appendix G).

**Practical implication of arm goodness.** The $\gamma$-goodness condition often arises in real-world applications where each objective has at least one arm (or item) that performs reasonably well. For example, in a personalized recommendation system optimizing multiple metrics such as click-through rates, watch time, and user satisfaction, it is plausible to assume that there exists at least one item among many that delivers high click-through rates, another (possibly different from the first one) that increases watch time, and so on. Consequently, the existence of "good arms" across different objective directions naturally aligns with many practical scenarios, reinforcing the applicability of our theoretical findings.

**Remark 1.** *The notion of $\gamma$-goodness is related to the concept of $\beta$-regularity introduced by Bayati et al. (2020) in the non-contextual multi-armed bandit framework. Specifically, they assume that the prior distribution $\Gamma$ for each arm's expected reward $\mu$ satisfies $\mathbb{P}_\mu[\mu > 1 - \epsilon] = \Theta(\epsilon^\beta)$ for every $\epsilon > 0$. Our $\gamma$-goodness generalizes this idea to linear reward bandit problems with multiple objectives. Comparing Assumption 4 under $d = M = 1$ with $\beta$-regularity in the stochastic context setting shows that $\gamma$-goodness is a* weaker *(and thus more general) condition than $\beta$-regularity. Appendix C.2 provides a detailed discussion contrasting these assumptions.*

**Remark 2.** *It is worthy noting that the above assumptions are irrelevant to the diversity assumption on context distribution which is commonly used in the existing greedy bandit literature (Kannan et al., 2018; Raghavan et al., 2018; Hao et al., 2020; Bastani et al., 2021). In particular, we explain cases where $\gamma$-goodness holds but the traditional diversity assumptions do not in Appendix C.3.*

Before we start our analysis, let $\alpha$ denote the value that satisfies the goodness condition defined in Definition 5, together with $\gamma$ as specified in Assumption 3. If $\alpha$ is greater than $\psi(\lambda, \gamma) := \sqrt{\frac{\lambda^2}{9} - \frac{\lambda^4}{324}} \, \gamma - \left(1 - \frac{\lambda^2}{18}\right) \sqrt{1 - \gamma^2}$, then we replace the value of $\alpha$ with $\psi(\lambda, \gamma)$. Since a larger $\alpha$ tightens the goodness condition, the condition remains valid even if $\alpha$ is reduced.

### 4.1 REGRET ANALYSIS OF MOG

We establish the lower bound of the minimum eigenvalue on the Gram matrix that grows linearly with respect to $t$. Specifically, instead of assuming contextual diversity as in prior greedy bandit

work, we leverage the presence of good arms for each objective to guarantee a constant lower bound on the growth of the Gram matrix's minimum eigenvalue over a single round-robin cycle. Let $T_0$ denote the number of initial rounds required until the condition $\lambda_{\min}(V_{t-1}) \geq B$ holds.

**Lemma 2** (Increment of the minimum eigenvalue of the Gram matrix). *Suppose that Assumptions 1, 2, and 3 hold. If the OLS estimator satisfies $\|\hat{\theta}_m(s) - \theta_m^*\| \leq \alpha$, for all $m \in [M]$ and for all $s \geq T_0 + 1$, then the selected arms for a single cycle $s = t_0, t_0 + 1, \ldots, t_0 + M - 1$ ($t_0 \geq T_0 + 1$) by Algorithm 1 satisfy*

$$\lambda_{\min}\left(\sum_{s=t_0}^{t_0+M-1} x(s)x(s)^\top\right) \geq \frac{\lambda}{3}M.$$

The proof of the lemma is provided in Appendix D.1.

It is well known that if the minimum eigenvalue of the Gram matrix increases linear with $t$, a regret bound of $\widetilde{\mathcal{O}}(\sqrt{T})$ can be derived. The following theorem establishes the Pareto regret of MOG.

**Theorem 1** (Pareto regret bound of MOG). *Suppose that Assumptions 1, 2, and 3 hold. If we run Algorithm 1 with $B = \min\left\{\frac{\sigma}{\alpha}\sqrt{2dT\log(dT^2)}, \frac{4\sigma^2}{\alpha^2}\left(\frac{d}{2}\log\left(1 + \frac{2T}{d}\right) + \log(T)\right)\right\}$, then the Pareto regret of Algorithm 1 is upper-bounded by*

$$\mathcal{PR}(T) \leq \frac{24\sigma}{\lambda}\sqrt{2dT\log(dT)} + 4T_0 + 10M.$$

The proof of the theorem is provided in Appendix D.2.

**Discussion of Theorem 1.** The theorem demonstrates that the cumulative Pareto regret bound of MOG is $\widetilde{\mathcal{O}}(\sqrt{dT})$ in terms of $d$ and $T$. To the best of our knowledge, our study is the first to prove the frequentiest regret bound of a greedy algorithm in the linear reward setting without relying on context stochasticity. Notably, this bound does not include the term dependent on $K$, and our algorithm performs well even when the number of arms is infinite. Furthermore, we show in Appendix J matching lower bound of $\Omega(\sqrt{dT})$ under our problem setting. Theorem 1 provides the theoretical foundation that if there are many good arms, simple near-greedy algorithms can outperform even more complicated exploration-based algorithms for multi-objective linear bandits (see Section 5).

Next, we analyze how quickly exploration can be completed. It is generally challenging to precisely determine a bound on the number of initial rounds $T_0$. However, in the MOG algorithm, if the feature vectors selected during the initial rounds span $\mathbb{R}^d$, $T_0$ can be upper-bounded by $\mathcal{O}(B)$.

**Corollary 1** (Number of initial rounds). *Suppose that Assumptions 1, 2, and 3 hold. If the feature set $S$ selected during the initial rounds in Algorithm 1 spans $\mathbb{R}^d$, then $T_0$ can be bounded by $T_0 \leq \lfloor B/\lambda_{\min}\left(\frac{1}{M}\sum_{x_i \in S} x_i(x_i)^\top\right)\rfloor + M$.*

The proof of the corollary is given in Appendix D.4.

The above corollary implies that $T_0$ in the bound stated in Theorem 1 is of the order $\log T$. Notably, Corollary 1 is valid for the deterministic version of MOG (Algorithm 1) in the case of fixed arms; however, similar results can be derived with high probability for randomized algorithms or in stochastic (varying) context settings (see Corollary 3).

**Remark 3.** *We can conclude the initial phase with minimal number of rounds by employing the most diverse set of $M$ feature vectors during the initial rounds.*

### 4.2 OBJECTIVE FAIRNESS OF MOG

We have confirmed that the MOG algorithm satisfies objective fairness. In the regret analysis of the MOG algorithm, we derived $l_2$ bounds on the estimators of each objective parameter (see Lemma 8). This implies that for a given $\epsilon > 0$, there exists $T_\epsilon$ such that, after round $T_\epsilon$, only $\epsilon$-optimal arms are selected with high probability. The following theorem establishes a lower bound on the objective fairness index.

**Theorem 2** (Objective fairness of MOG). *Suppose that Assumptions 1, 2, and 3 hold. If we run Algorithm 1 using $B$ as given in Theorem 1, the objective fairness index of Algorithm 1 is bounded*

*below by*

$$\text{OFI}_{\epsilon,T} \geq \left(\frac{T - T_\epsilon - M}{MT}\right)\left(1 - \frac{3M}{T}\right),$$

*where $T_\epsilon = \max\left(\lfloor\frac{288\sigma^2 d\log(dT)}{\lambda^2\epsilon^2}\rfloor + T_0 + M,\ 2T_0 + 2M\right)$.*

The proof of the theorem is provided in Appendix D.3.

**Discussion of Theorem 2.** The theorem shows that Algorithm 1 satisfies objective fairness. Notably, for any given $\epsilon > 0$, $\lim_{T\to\infty}\text{OFI}_{\epsilon,T} = \frac{1}{M}$ and the limit does not include a term with $K$. Furthermore, we prove that with high probability, our algorithm selects near-optimal arms for each objective at a ratio of $\frac{1}{M}$ in the long run (this phenomenon is also observed in the experiments in section 5), and it selects only $\epsilon$-optimal arms of an objective after a certain number of rounds $T_\epsilon$. To our knowledge, this is the first theoretical analysis of fairness in multi-objective bandits.

## 5 EXPERIMENT

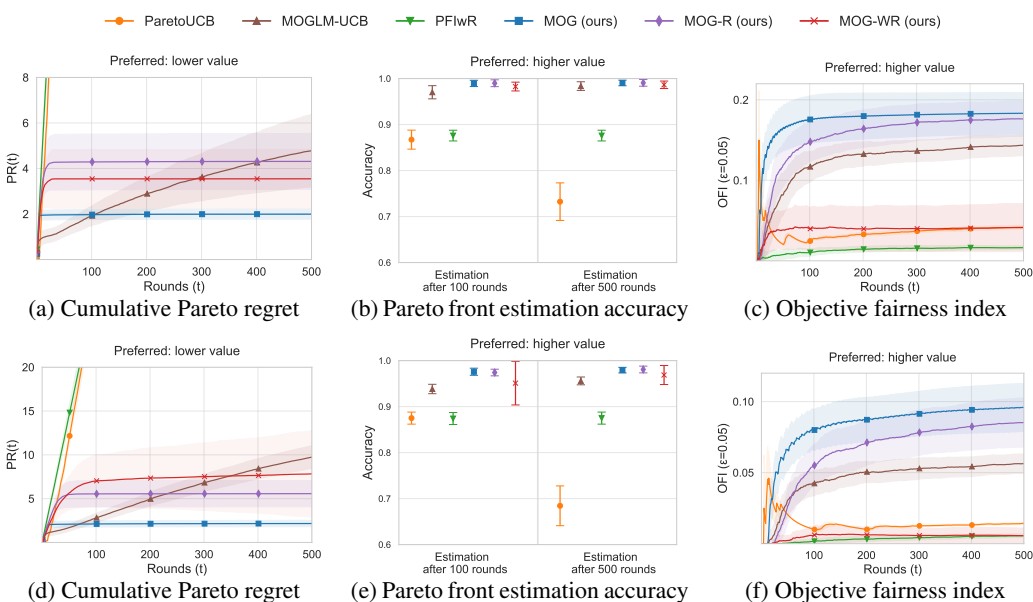

Figure 1: Evaluation of multi-objective bandit algorithms with tuned parameters. The top row shows results for $d = 5$, $K = 50$, $M = 5$, and the bottom row shows results for $d = 10$, $K = 100$, $M = 10$. The shaded areas and the error bars indicate $\pm$ half the standard deviation for each algorithm.

We evaluate our proposed algorithms –MOG, MOG-R, and MOG-WR– in both fixed and stochastic context settings, comparing them with ParetoUCB (Drugan & Nowe, 2013), MOGLM-UCB (Lu et al., 2019), and PFIwR (Kim et al., 2023). Performance is assessed in terms of Pareto regret, Pareto front estimation accuracy, and objective fairness under the linear bandit model $y_m(t) \sim \mathcal{N}(x_i^\top\theta_m^*, 0.1^2)$ for all $i \in [K]$ and $m \in [M]$. The detailed experimental settings are provided in Appendix K.1. Figures 1 illustrates the performance of each algorithm in the fixed feature setup.

The result in Figures 1a and 1d clearly demonstrates that our proposed algorithms outperform the others empirically, despite their simpler structure. Moreover, in Figures 1b and 1e, we observe that MOG, MOG-R and MOG-WR approach the true Pareto front more efficiently than the empirical Pareto fronts used by other algorithms. Additionally, we confirm that the objective fairness indices of MOG and MOG-R (with uniform objective distribution) converge to approximately $\frac{1}{M}$, regardless of $K$ (Figures 1c and 1f). Additional results, including performance evaluations in various settings and under stochastic contexts, the effect of parameter settings on algorithm performance, experiment results based on real-world data, and a more in-depth analysis are presented in Appendix K.

## LLM USAGE

We employed an LLM for typo correction and grammar checking.

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

CONTENTS OF APPENDIX

## A    ADDITIONAL NOTATIONS

We define the $d$-dimensional ball $\mathbb{B}_R^d = \{x \in \mathbb{R}^d \mid \|x\|_2 \leq R\}$ and the $(d-1)$-dimensional sphere $\mathbb{S}_R^{d-1} = \{x \in \mathbb{R}^d \mid \|x\|_2 = R\}$. $R$ can be omitted for simplicity if $R = 1$, i.e. $\mathbb{B}^d := \mathbb{B}_1^d$, and $\mathbb{S}^{d-1} := \mathbb{S}_1^{d-1}$. We define $\Delta^M$ as the $M$-dimensional simplex, given by $\{(w_1, \ldots, w_M) \in \mathbb{R}^d \mid \sum_{m \in [M]} w_i = 1, \ w_1, \ldots w_M \geq 0\}$. We also denote the positive orthant of $\mathbb{R}^d$ by $\mathbb{R}_+^d$. For matrices $A$ and $B$, we write $A \succeq B$ to indicate that $A - B$ is positive definite. The $i$-th unit vector in $\mathbb{R}^d$ is denoted by $e_i^{(d)}$, and when the dimension $d$ is clear from the context, we simply write $e_i$. We define the spanning space of feature vectors $x_1, \ldots, x_K$ as $S_x$, and its orthogonal complement as $S_x^\perp$. The projection map onto $S_x$ is denoted by $\pi_{S_x} : \mathbb{R}^d \to S_x$, and when the space $S_x$ is clear from the context, we simply write $\pi_s$.

## B    GENERALIZED OBJECTIVE FAIRNESS

In Section 2.2.2, we defined the objective fairness of a multi-objective algorithm. Objective fairness guarantees that the near-optimal arms in each objective direction are consistently selected without neglecting any objective. Building on this principle, we propose a generalized objective fairness criterion that ensures a multi-objective algorithm continues to select the near-optimal arms across all weight-sum directions of the objectives.

**Definition 6** (Generalized objective fairness). *Given a weight vector $w \in \Delta^M$, let $\mu_{i,w} := \sum_{m \in [M]} w_m x_i^\top \theta_m^*$ be the expect weighted reward of arm $i$, $a_w^*$ be the arm that has the largest expected weighted reward with respect to the weight vector $w$, and $\mu_w^* := \mu_{a_w^*,m}$. For all $\epsilon > 0$, define **the generalized objective fairness index** $\mathrm{GOFI}_{\epsilon,T}$ of an algorithm as*

$$\mathrm{GOFI}_{\epsilon,T} := \inf_{w \in \Delta^M} \left( \frac{1}{T} \mathbb{E} \left[ \sum_{t=1}^T \mathbb{1}\{\mu_w^* - \mu_{a(t),w} < \epsilon\} \right] \right).$$

*Then, we say that the algorithm satisfies the **generalized objective fairness** if for given $\epsilon$, there exists a positive lower bound $L_\epsilon$ that satisfies $\lim_{T \to \infty} \mathrm{GOFI}_{\epsilon,T} \geq L_\epsilon$.*

Intuitively, generalized objective fairness considers intermediate arms on the Pareto front, which are not optimal for individual objectives. It ensures the consistent selection of the optimal arms corresponding to some weight-sum reward functions. The next lemma explains that the generalized objective fairness criterion guarantees that the algorithm consistently selects Pareto near-optimal arms that lie within the *positive side* (Figure 2) of the Pareto front.

**Lemma 3** (Boyd & Vandenberghe (2004)). *Consider a multi-criterion problem, minimizing $F(x) = (f_1(x), \ldots, f_m(x))$ with respect to $\mathbb{R}_+^m$. In scalarization, we choose a positive vector $\widetilde{w}$, and minimize the scalar function $\widetilde{w}^\top F(x)$. Then, any minimizer for scalarization is guaranteed to be Pareto optimal, and conversely, every Pareto optimal of a convex multi-criterion problem minimizes the function $\widetilde{w}^\top F(x)$ for some nonnegative weight vector $\widetilde{w}$.*

**Corollary 2.** *In a multi-objective bandit problem, the optimal arms corresponding to weight-sum scalarized reward functions are contained in Pareto Front. Conversely, every Pareto optimal arms that lie within the positive side of the Pareto front are optimal for some weight-sum scalarized reward function.*

**Remark 4.** *In a multi-objective bandit problem, if the true Pareto front is convex, the generalized objective fairness ensures consistent selection of the entire Pareto front.*

In some cases, it may be important to determine whether an algorithm fully explores the entire Pareto front. If we aim to evaluate whether an algorithm consistently selects the entire Pareto front, we can further extend the concept of GOF. In such cases, fairness can be redefined by employing alternative scalarization methods that enable full exploration of the Pareto front (Paria et al., 2020; Golovin & Zhang, 2020; Zhang, 2024), rather than relying on the weighted sum.

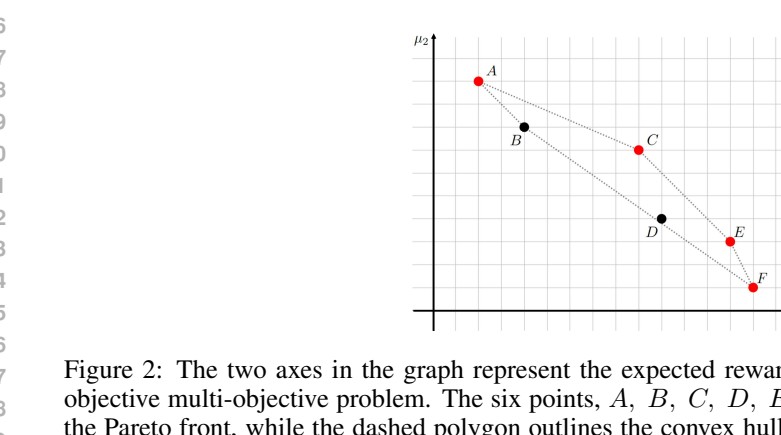

Figure 2: The two axes in the graph represent the expected rewards for each objective in a two-objective multi-objective problem. The six points, $A$, $B$, $C$, $D$, $E$, and $F$, in the figure represent the Pareto front, while the dashed polygon outlines the convex hull of the Pareto front. Within the Pareto front, *the positive side of the Pareto front* refers to the points located on the positive side of the convex hull, highlighted in red, corresponding to points $A$, $C$, $E$, and $F$.

### B.1 GENERALIZED OBJECTIVE FAIRNESS VS FAIRNESS SUGGESTED BY DRUGAN & NOWE (2013)

In this section, we explain how our fairness criterion differs from and improves upon the one proposed by Drugan & Nowe (2013). The fairness criterion defined by Drugan & Nowe (2013) is as follows, and we refer to it as *Pareto front fairness*.

**Definition 7** (Pareto front fairness (Drugan & Nowe, 2013)). *Let $T_i^*(n)$ be the number of rounds an optimal arm $i$ is pulled, and $\mathbb{E}[T^*(n)]$ be the expected number of times optimal arms are selected. The unfairness of a multi-objective bandit algorithm is defined as the variance of the arms in Pareto front $\mathcal{A}^*$,*

$$\phi = \frac{1}{|\mathcal{A}^*|} \sum_{i \in \mathcal{A}^*} \left( T_i^*(n) - \mathbb{E}[T^*(n)] \right)^2.$$

*For a perfectly fair usage of optimal arms, we have that $\phi \to 0$.*

Now, we compare our generalized objective fairness (GOF) with Pareto front fairness (PFF). The key differences and improvements are summarized as follows:

- GOF guarantees consistent selection of Pareto-optimal arms lies within the positive side of the Pareto front, while PFF considers the entire Pareto front (see Corollary 2).

- Statistical analysis is feasible with GOF but not with PFF, as PFF requires the number of times each true optimal arm is pulled, which can only be computed in simulated studies. In contrast, the definition of GOF incorporates an $\epsilon$ argument, enabling theoretical analysis. Detailed theoretical analysis of fairness is provided in Appendices D.3, E.3, and F.4.

- GOF accommodates differences in the importance of objectives, whereas PFF assumes equal importance across objectives. These differences are reflected in the indices of the two fairness criteria. GOF uses the lower bound of the selection ratio for each optimal arm as its index, whereas PFF employs the variance in the frequency of selecting each optimal arm.

- Algorithms based on the GOF perspective do not require computing the empirical Pareto front, whereas PFF-based algorithms incur additional computational costs for empirical Pareto front estimation.

## C  $\gamma$-GOODNESS

In this section, we introduce the concept of $\gamma$-goodness, compare it with the alternative regularity condition employed in another greedy bandit study Bayati et al. (2020), and clarify the distinction between $\gamma$-goodness and context diversity (Assumption 3 in Bastani et al. 2021) assumption, which is commonly used in the existing greedy bandit literature.

We first extend the definition of a $\gamma$-good arm in Definition 5 to an arbitrary vector.

**Definition 8** ($\gamma$-good vectors). *For fixed $\gamma \in (0, 1]$, we say that the vector $x \in \mathbb{R}^d$ is $\gamma$-good for the direction of $\theta \in \mathbb{R}^d$ if $x^\top \frac{\theta}{\|\theta\|_2} \geq \gamma$ holds.*

The following naturally arises from the definition; however, it plays a pivotal role in applying the goodness assumption to the analysis.

**Proposition 1.** *If there exists a $\gamma$-good arm for $\theta$, then the optimal arm for $\theta$ is also $\gamma$-good.*

**Proposition 2.** *Suppose $x$ is a random variable that can only take values corresponding to $\gamma$-good arms for $\theta$. Then, $\mathbb{E}[x]$ is also $\gamma$-good for $\theta$.*

The above proposition holds because the region $\{x \in \mathbb{B}^d \mid x^\top \frac{\theta}{\|\theta\|_2} \geq \gamma\}$ is convex.

## C.1 $\gamma$-GOODNESS CONDITION FOR STOCHASTIC CONTEXTS SETUP

Before explaining the meaning of $\gamma$-goodness, we first extend the $\gamma$-goodness condition to be applicable in a stochastic context setup. In a multi-objective linear contextual bandit framework, the stochastic context setup assumes that the context set $\chi(t) = \{x_i(t) \in \mathbb{R}^d, i \in [K]\}$ in each round $t$ is drawn from some unknown distribution $P_\chi(t)$. Detailed explanations regarding this problem can be found in Section G.1. Under the stochastic context setup, we introduce the definition of goodness with respect to the context distribution and present the $\gamma$-goodness assumption as follows.

**Definition 9** (Goodness of arms – stochastic context version). *For fixed $\gamma \leq 1$, we say that the distribution $P_\chi(t)$ of feature vector set $\chi(t)$ satisfies $\gamma$-goodness condition if there exists a positive number $q_\gamma$ that satisfies*

$$\text{for all } \beta \in \mathbb{S}^{d-1}, \ \mathbb{P}_{\chi(t)}[\exists i \in [K], \ x_i(t)^\top \beta \geq \gamma] \geq q_\gamma.$$

**Assumption 4** ($\gamma$-goodness – stochastic context version). *We assume $P_\chi(t)$ satisfies $\gamma$-goodness condition for all $t \in [T]$, with $\gamma > 1 - \frac{\lambda^2}{18}$.*

Different from fixed version, the goodness condition requires only the positive probability $q_\gamma$ of the presence of $\gamma$-good arms not the existence of them (i.e. $q_\gamma = 1$). Instead, the condition requires $\gamma$-good arms for not only the neighborhood of objective parameters but also all directions. In other words, $\gamma$-goodness signifies that for any direction $\beta \in \mathbb{S}^{d-1}$, there exists at least one $\gamma$-good arm with a probability of at least $q_\gamma$. Intuitively, if the union of the supports of each arm $x_i(t)$ for $i \in [K]$ covers all of $\mathbb{S}^{d-1}$, $\gamma$-goodness will be guaranteed for all $\gamma < 1$. The following lemma formalizes this concept.

**Lemma 4.** *Suppose $x_1(t), \ldots, x_K(t)$ are continuous variables with density function $f_1, \ldots, f_K$. If $f = f_1 + \ldots + f_K$ is a bounded function and positive near $\mathbb{S}^{d-1}$ (i.e., there exist $r \in (0, 1)$ satisfies $f$ is always positive at $\{x \in \mathbb{R}^d \mid r < \|x\|_2 < 1\}$), then $P_\chi(t)$ satisfies $\gamma$-goodness for all $\gamma \in (0, 1)$.*

*Proof.* Fix $\gamma \in (0, 1)$. From the definition of $f$, $f/K$ is the probability density function of $X = \text{Uniform}(x_1(t), \ldots, x_K(t))$. Define $p_\beta = \mathbb{P}_{\chi(t)}[X^\top \beta \geq \gamma]$ for unit vector $\beta \in \mathbb{S}^{d-1}$. Then,

$$p_\beta = \mathbb{P}_{\chi(t)}[X^\top \beta \geq \gamma] = \int_{\{x \in \mathbb{B}^R \mid x^\top \beta \geq \gamma\}} \frac{f(x)}{K} dx \geq \int_{\{x \in \mathbb{B}^R \mid x^\top \beta \geq \max(\gamma, r)\}} \frac{f(x)}{K} dx > 0,$$

for all $\beta \in \mathbb{S}^{d-1}$.

Consider the function $F : \beta \xrightarrow{F} p_\beta$. From the boundedness of $f$, we can easily check $F$ is continuous. By the fact that the compactness is preserved by continuous functions, $\{p_\beta \mid \beta \in \mathbb{S}^{d-1}\}$ is compact. Define $q_\gamma := \min\{p_\beta \mid \beta \in \mathbb{S}^{d-1}\}$, then we have $q_\gamma > 0$ since $p_\beta > 0$ for all $\beta \in \mathbb{S}^{d-1}$. Then, for all $\beta \in \mathbb{S}^{d-1}$

$$\mathbb{P}_{\chi(t)}[\exists i \in [K], \ x_i(t)^\top \beta \geq \gamma] \geq \mathbb{P}_{\chi(t)}[X^\top \beta \geq \gamma] = p_\beta \geq q_\gamma$$

$\square$

**Remark 5.** *The above lemma states that if the set of arm $\chi(t)$ includes just a single continuous variable that can cover $\mathbb{S}^{d-1}$, then $\gamma$-goodness will hold for all $\gamma < 1$ regardless of the distributions of the remaining arms.*

## C.2 $\gamma$-GOODNESS VS $\beta$-REGULARITY

In Bayati et al. (2020), they assume the prior distribution $\Gamma$ of the expected reward $\mu$ of each arm satisfies $\mathbb{P}_\mu[\mu > 1 - \epsilon] = \Theta(\epsilon^\beta)$ for all $\epsilon > 0$ in non-contextual MAB setting. Let's compare this with $\gamma$-goodness when $m = d = 1$. We claim that $\gamma$-goodness can be considered weaker than $\beta$-regularity from three perspectives.

The most significant difference is that in $\beta$-regularity, the probability that the expected reward $\mu_i$ exceeds $1 - \epsilon$ is required for all arm $i \in [K]$, along with the assumption that $\mu_i$'s are drawn independently from prior $\Gamma$. In contrast, in $\gamma$-goodness, it is sufficient to ensure that the probability that one of the $K$ arms satisfies $x_i(t)^\top \beta \geq \gamma$, without the need for the independence assumption between arm vectors. Secondly, unlike $\beta$-regularity, $\gamma$-goodness does not require a specific relationship like $\Theta(1 - \gamma)$ between the probability of the existence of near-optimal arms $\mathbb{P}_{\chi(t)}[\exists i \in [K], \, x_i(t)^\top \beta \geq \gamma]$ and the threshold $\gamma$; instead, it focuses on the existence of a positive lower bound $q_\gamma$. Lastly, the $\beta$-regularity assumes the probability of $\mu > 1 - \epsilon$ for all $\epsilon > 0$, while this work does not mandate $\gamma$-goodness for $\gamma$ very close to 1; it is sufficient to hold $\gamma$-goodness only for some $\gamma \geq 1 - (\frac{\lambda}{18})^2$.

## C.3 $\gamma$-GOODNESS VS CONTEXT DIVERSITY

In recent years, there has been significant interest in the optimality of the Greedy algorithm in single-objective bandit problems (Bastani et al., 2021; Kannan et al., 2018; Raghavan et al., 2018; Hao et al., 2020). A common theme among these studies is the assumption that feature vectors follow a distribution satisfying specific diversity conditions. For example, Bastani et al. (2021) assume the existence of a positive constant $\lambda$ such that for each vector $u \in \mathbb{R}^d$ and context vector $x_i(t)$, $\lambda_{\min}\big(\mathbb{E}[x_i(t)x_i(t)^\top \mathbb{1}\{x_i(t)^\top u \geq 0\}]\big) \geq \lambda$. The $\gamma$-goodness condition fundamentally differs from traditional context diversity assumptions. Below, we provide examples where the $\gamma$-goodness condition holds, while traditional diversity conditions do not.

**Example 1** (Containing fixed arms) Imagine a situation where one feature vector is a continuous variable while the other arms are fixed. For example, let $x_1(t)$ be uniformly distributed over $\mathbb{B}^d$ while $x_2(t) = x_2, \ldots, x_K(t) = x_K$ are fixed at some points in $\mathbb{S}^{d-1}$. By Lemma 4, $P_\chi(t)$ satisfies $\gamma$-goodness for all $\gamma \in (0, 1)$. However, it is easy to see that diversity is not satisfied because $\lambda_{\min}\big(\mathbb{E}[x_2(t)x_2(t)^\top \mathbb{1}\{x_2(t)^\top u \geq 0\}]\big) = \lambda_{\min}(x_2 x_2^\top \mathbb{1}\{x_2^\top u \geq 0\}) \leq \lambda_{\min}(x_2 x_2^\top) = 0$.

**Example 2** (Low-randomness distribution) Consider a scenario where the feature vectors are drawn from a finite set of discrete points. Despite the lack of diversity, if these points are strategically chosen to cover $\mathbb{S}^{d-1}$ adequately, the goodness condition can still be satisfied. For example, suppose there is a set of points $P = \{a_1, a_2, \ldots, a_N\}$ that contains $\sqrt{1 - \gamma^2}$-net of $\mathbb{S}^{d-1}$. Assume that $x_1(t)$ be chosen uniformly from the $d - 1$ points and other arms $x_2(t), \ldots, x_K(t)$ be chosen from the remaining points. Obviously, $P_\chi(t)$ satisfies $\gamma$-goodness with $q_\gamma \geq \frac{1}{N}$. In contrast, $\lambda_{\min}\big(\mathbb{E}[x_1(t)x_1(t)^\top \mathbb{1}\{x_1(t)^\top u \geq 0\}]\big) = 0$ since there are only $d - 1$ candidates that can be $x_1(t)$. Therefore, context diversity does not hold in this scenario.

Although $\gamma$-goodness encompasses cases where the traditional context diversity assumption is not covered, there is no inclusion relationship between the two conditions. Here is an example where $\gamma$-goodness does not hold, but context diversity does.

**Example 3** (Proper support) Consider a case where 1 is given as the upper bound of the $l_2$ norm of feature vectors, but the actual support of feature vectors is smaller. For instance, if $x_i(t)$ follows a uniform distribution over $\mathbb{B}_{1/2}^d$ for all $i \in [K]$ and $t \in [T]$, then context diversity still holds (Bastani et al. (2021)), but $\gamma$-goodness does not hold for $\gamma > 1/2$.

# D  ANALYSIS OF MOG WITH FIXED FEATURES

## D.1  PROOF OF LINEAR GROWTH OF MINIMUM EIGENVALUE OF THE GRAM MATRIX

The proof of Lemma 2 is presented in Section D.1.2 and its supporting lemmas are presented in Section D.1.1.

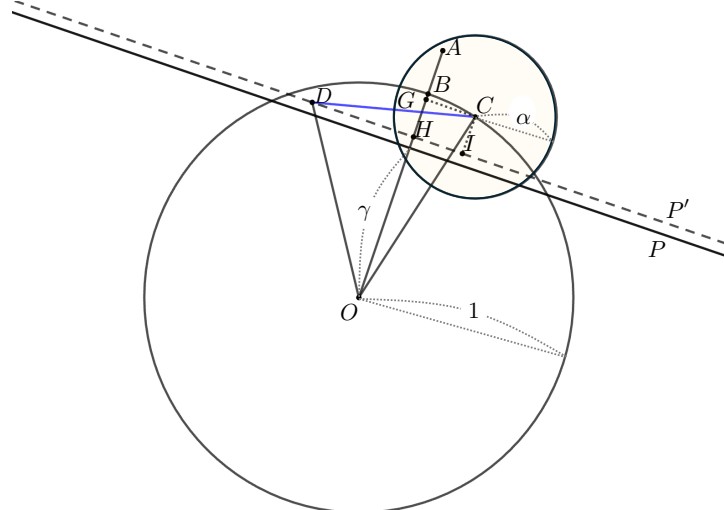

Figure 3: The larger circle represents the unit sphere in $\mathbb{R}^d$ while the interior of smaller circle indicates the region where $\hat{\theta}_m(s)$ may exist. Then, the blue line illustrates the distance between $\theta_m^*$ and the $\gamma$-good arm for $\hat{\theta}_m(s)$.

### D.1.1   Technical lemmas for Lemma 2

The following lemma states that, after sufficient exploration rounds, the distance between the $\gamma$-good arms for the OLS estimator of the objective parameters and the respective true objective parameters can be bounded.

**Lemma 5.** *Given Assumptions 1, assume the OLS estimator satisfies $\|\hat{\theta}_m(s) - \theta_m^*\| \leq \alpha$, for $m \in [M]$ and $s \geq T_0 + 1$. If $x \in \mathbb{B}^d$ is $\gamma$-good for $\hat{\theta}_m(s)$, then the distance between $x$ and $\theta_m^*$ is bounded by*

$$\|\theta_m^* - x\|_2 \leq \sqrt{2 + 2\alpha\sqrt{1 - \gamma^2} - 2\gamma\sqrt{1 - \alpha^2}}.$$

*Proof.* Let the origin be denoted by $O$, and define $\hat{\theta}_m(s) := \vec{OA}$, $\frac{\hat{\theta}_m(s)}{\|\hat{\theta}_m(s)\|} := \vec{OB}$, $\theta_m^* = \vec{OC}$, and let a $\gamma$-good arm $x$ be denoted by $\vec{OD}$. By Assumption 1, $C$ lies on the unit $d$-dimension sphere centered at $O$(sphere $O$). By the assumption of the lemma, $A$ lies on or inside the sphere centered $C$ with radius $\alpha$(sphere $C$), and $B$ is the intersection point of the extension of $OA$ with sphere $O$. Define the hyperplane $P$, orthogonal to $OB$, that passes through the point dividing $OB$ in the ratio $\gamma : 1 - \gamma$. Then, by the definition of $\gamma$-good arms, point $D$ lies on or inside the unit sphere $O$, and must be located on or beyond the hyperplane $P$.

Let $G$ and $H$ denote the foot of the perpendiculars from $C$ and $D$ onto $OB$, respectively. Since $D$ is located on or beyond the hyperplane $P$, $OH \geq \gamma$ and (the distance between $D$ and $OB$ )= $DH = \sqrt{OD^2 - OH^2} \leq \sqrt{1 - \gamma^2}$. Letting (the distance between $C$ and $OB$ )= $CG = l$ $(\leq \alpha)$, we have $GH = OG - OH = \sqrt{OC^2 - CG^2} - OH \leq \sqrt{1 - l^2} - \gamma$. The equality holds when $OD = 1$ and $OH = \gamma$, i.e., $D$ is lying on the hyperplane $P$.

Now, consider the hyperplane $P'$ that passes through both $D$ and $H$ and is orthogonal to $OB$ (Figure 3 illustrates the case when $P = P'$). Let $I$ be the foot of the perpendicular from $C$ to $P'$. Then, $\square CGHI$ is a rectangle and we can bound $DI \leq DH + IH = DH + CG \leq \sqrt{1 - \gamma^2} + l$. The equality holds for both inequality when $D$, $H$, and $I$ lie on the same line, in that order, and $D$ is lying on the hyperplane $P$.

Therefore, by the Pythagorean theorem,

$$\|\theta_m^* - x\|_2^2 = CD^2 = DI^2 + CI^2 = DI^2 + GH^2$$

$$\leq (\sqrt{1-\gamma^2} + l)^2 + (\sqrt{1-l^2} - \gamma)^2$$

$$= 1 - \gamma^2 + l^2 + 2(\sqrt{1-\gamma^2})l + 1 - l^2 + \gamma^2 - 2(\sqrt{1-l^2})\gamma$$

$$= 2 + 2(\sqrt{1-\gamma^2})l - 2(\sqrt{1-l^2})\gamma$$

$$\leq 2 + 2(\sqrt{1-\gamma^2})\alpha - 2(\sqrt{1-\alpha^2})\gamma.$$

The last inequality holds since $l \leq \alpha$. $\qquad\square$

Now, we will demonstrate that the $\gamma$-good arms for multiple objectives spans $\mathbb{R}^d$ by deriving a lower bound on the minimum eigenvalue of the Gram matrix constructed from $\gamma$-good arms.

**Lemma 6.** *Given Assumptions 1 and 2, assume the OLS estimator satisfies $\|\hat{\theta}_m(s) - \theta_m^*\| \leq \alpha$, for all $m \in [M]$ and $s \geq T_0 + 1$. If $x_{r(1)}, \ldots, x_{r(M)} \in \mathbb{B}^d$ are $\gamma$-good for $\hat{\theta}_1(s_1), \ldots, \hat{\theta}_M(s_M)$ for some $s_1, \ldots, s_M \geq T_0 + 1$, respectively, then the following holds*

$$\lambda_{\min}\left(\sum_{m \in [M]} x_{r(m)}\left(x_{r(m)}\right)^\top\right) \geq \frac{\lambda}{3}M.$$

*Proof.* For all $m \in [M]$, we can get $\|x_{r(m)} - \theta_m^*\| \leq \sqrt{2 + 2\alpha\sqrt{1-\gamma^2} - 2\gamma\sqrt{1-\alpha^2}}$ by Lemma 5.

Then, for any unit vector $u \in \mathbb{B}^d$,

$$u^\top\left(\sum_{m \in [M]} x_{r(m)}\left(x_{r(m)}\right)^\top\right)u = \sum_{m \in [M]} \langle u, x_{r(m)}\rangle^2$$

$$= \sum_{m \in [M]} \langle u, \theta_m^* + (x_{r(m)} - \theta_m^*)\rangle^2$$

$$= \sum_{m \in [M]} \{\langle u, \theta_m^*\rangle^2 + \langle u, x_{r(m)} - \theta_m^*\rangle^2 + 2\langle u, \theta_m^*\rangle\langle u, x_{r(m)} - \theta_m^*\rangle\}$$

$$\geq u^\top\left(\sum_{m \in [M]} \theta_m^*\left(\theta_m^*\right)^\top\right)u + 0 - 2\sqrt{2 + 2\alpha\sqrt{1-\gamma^2} - 2\gamma\sqrt{1-\alpha^2}}M$$

$$\geq \lambda M - 2\sqrt{2 + 2\alpha\sqrt{1-\gamma^2} - 2\gamma\sqrt{1-\alpha^2}}M.$$

We define $\alpha$ in Section 4 as having a value less than or equal to $\psi(\lambda, \gamma) := \sqrt{\frac{\lambda^2}{9} - \frac{\lambda^4}{324}}\,\gamma - \left(1 - \frac{\lambda^2}{18}\right)\sqrt{1-\gamma^2}$. This leads the inequality $\lambda - 2\sqrt{2 + 2\alpha\sqrt{1-\gamma^2} - 2\gamma\sqrt{1-\alpha^2}} \geq \frac{\lambda}{3}$. Therefore, we have

$$\lambda_{\min}\left(\sum_{m \in [M]} x_{r(m)}\left(x_{r(m)}\right)^\top\right) \geq \left(\lambda - 2\sqrt{2 + 2\alpha\sqrt{1-\gamma^2} - 2\gamma\sqrt{1-\alpha^2}}\right)M \geq \frac{\lambda}{3}M.$$

$\qquad\square$

### D.1.2 PROOF OF LEMMA 2.

The previous lemma shows that the minimum eigenvalue of the Gram matrix increases at a rate of $\mathcal{O}(\lambda)$. It is well known that if the minimum eigenvalue of the Gram matrix increases linear with $t$, a regret bound of $\widetilde{\mathcal{O}}(\sqrt{T})$ can be derived.

*Proof.* For $s = t_0, \ldots, t_0 + M - 1$ $(t_0 \geq T_0 + 1)$, $\|\hat{\theta}_m(s) - \theta_m^*\| < \alpha$ for all $m \in [M]$. Then, by Assumption 3 and Proposition 1, the selected arm $x(s)$ are $\gamma$-good arms for the corresponding target objectives in round $s = t_0, \ldots, t_0 + M - 1$. Since the target objectives in round $s = t_0, \ldots, t_0 + M - 1$ are all different $M$ objectives, we have

$$\lambda_{\min} \left( \sum_{s=t_0}^{t_0+M-1} x(s)x(s)^\top \right) \geq \frac{\lambda}{3} M,$$

by Lemma 6. $\qquad \square$

## D.2 PROOF OF THE REGRET BOUND

Theorem 1 is proven by deriving an $l_2$ bound on $\hat{\theta}_m(t) - \theta_m^*$. This is enabled by Lemma 2, which shows that the minimum eigenvalue of the Gram matrix grows linearly with $t$ with high probability, thereby allowing us to obtain the desired bound. The proof of Theorem 1 is presented in Section D.2.2 and its supporting lemmas are presented in Section D.2.1.

### D.2.1 TECHNICAL LEMMAS FOR THEOREM 1

To apply Lemma 2, a sufficient number of initial exploration is required to ensure its preconditions are satisfied. We discuss this requirement in the next section (Section D.4). In the current section, we assume this condition is met via Lemma 9, and proceed to prove Theorem 1.

**Lemma 7** (Minimum eigenvalue growth). *Suppose Assumptions 1, 2, and 3 hold, and fix $\delta > 0$. If we run Algorithm 1 with $B = \min \left[ \frac{\sigma}{\alpha} \sqrt{2dT \log(\frac{dT}{\delta})}, \ \frac{4\sigma^2}{\alpha^2} \left( \frac{d}{2} \log \left( 1 + \frac{2T}{d} \right) + \log \left( \frac{1}{\delta} \right) \right) \right]$, then with probability $1 - 2M\delta$, the following holds for the minimum eigenvalue of the Gram matrix*

$$\lambda_{\min} \left( \sum_{s=1}^{t-1} x(s)x(s)^\top \right) \geq B + \frac{\lambda}{3}(t - T_0 - M),$$

*for $T_0 + M \leq t \leq T$.*

*Proof.* If we choose $B$ as stated in the lemma, the OLS estimator satisfies $\|\hat{\theta}_m(s) - \theta_m^*\| \leq \alpha$ for all $s \geq T_0 + 1$ and $m \in [M]$ with probability $1 - 2M\delta$, by Lemma 9. Thus, by applying Lemma 2 to every single round after exploration, we have, for $t \geq T_0 + M$,

$$\lambda_{\min} \left( \sum_{s=1}^{t-1} x(s)x(s)^\top \right) \geq \lambda_{\min} \left( \sum_{s=1}^{T_0} x(s)x(s)^\top \right) + \lambda_{\min} \left( \sum_{s=T_0+1}^{t-1} x(s)x(s)^\top \right)$$

$$\geq B + \left[ \frac{t - 1 - T_0}{M} \right] \times \frac{\lambda}{3} M,$$

$$\geq B + \frac{\lambda}{3}(t - T_0 - M).$$

$\square$

With Lemma 7, we are ready to derive the $l_2$ bound of $\hat{\theta}_m(t) - \theta_m^*$ for $m \in [M]$.

**Lemma 8.** *Fix $\delta > 0$. Under the same conditions as those in Lemma 7, with probability at least $1 - 3M\delta$, for all $m \in [M]$ and $t \geq 2T_0 + 2M$, the OLS estimator $\hat{\theta}_m(t)$ of $\theta_m^*$ satisfies*

$$\left\| \hat{\theta}_m(t) - \theta_m^* \right\|_2 \leq \frac{6\sigma}{\lambda} \sqrt{\frac{d \log(dt/\delta)}{t - T_0 - M}}.$$

*Proof.* From the closed form of the OLS estimators, for all $m \in [M]$,

$$\left\| \hat{\theta}_m(t) - \theta_m^* \right\|_2 = \left\| \left( \sum_{s=1}^{t-1} x(s)x(s)^\top \right)^{-1} \sum_{s=1}^{t-1} x(s)\eta_{a(s),m}(s) \right\|_2$$

$$\leq \frac{1}{\lambda_{\min} \left( \sum_{s=1}^{t-1} x(s)x(s)^\top \right)} \left\| \sum_{s=1}^{t-1} x(s)\eta_{a(s),m}(s) \right\|_2$$

For the denominator, we have $\lambda_{\min}(V_{t-1}) \geq B + \frac{\lambda}{3}(t - T_0 - M)$ for $t \geq T_0 + M$, with probability at least $1 - 2M\delta$, by Lemma 7. To bound the $l_2$ norm of $S_{t-1,m} := \sum_{s=1}^{t-1} x(s)\eta_{a(s),m}(s)$, we can use Lemma 23, the martingale inequality of Kannan et al. (2018). The lemma states that for fixed $m \in [M]$, $\|S_{t-1,m}\|_2 \leq \sigma\sqrt{2dt \log(dt/\delta)}$ holds with probability at least $1 - \delta$. Therefore, with probability at least $1 - 3M\delta$, for all $m \in [M]$ and $t \geq 2T_0 + 2M$,

$$\left\| \hat{\theta}_m(t) - \theta_m^* \right\|_2 \leq \frac{\sigma\sqrt{2dt \log(dt/\delta)}}{B + \lambda(t - T_0 - M)/3} \leq \frac{6\sigma}{\lambda} \sqrt{\frac{d \log(dt/\delta)}{t - T_0 - M}}.$$

The last inequality holds when $t \geq 2T_0 + 2M$. $\qquad\square$

### D.2.2 PROOF OF THEOREM 1

*Proof.* Let $E$ be the event that $\left\| \hat{\theta}_m(t) - \theta_m^* \right\|_2 \leq \frac{6\sigma}{\lambda} \sqrt{\frac{d \log(dtT)}{t - T_0 - M}}$ holds for all $m \in [M]$ and $t \geq 2T_0 + 2M$. Then, $\mathbb{P}(\bar{E}) \leq \frac{3M}{T}$ by Lemma 8 with $\delta = \frac{1}{T}$.

Let $m(t)$ be the target objective for round $t$ and $a_m^*$ be the optimal arm with respect to objective $m$. Then, the suboptimality gap on round $t$ is bounded by

$$\Delta_{a(t)}(t) \leq \left( x_{a_{m(t)}^*} \right)^\top \theta_{m(t)}^* - x(t)^\top \theta_{m(t)}^* \leq 2\|\hat{\theta}_{m(t)}(t) - \theta_{m(t)}^*\|_2.$$

Let $\Delta_{\max}$ be the maximum suboptimality gap. For $t \geq 2T_0 + 2M$,

$$\mathbb{E}[\Delta_{a(t)}(t)] \leq \mathbb{E}[\Delta_{a(t)}(t) \mid E] + \mathbb{P}(E)\Delta_{\max}$$

$$\leq 2\mathbb{E}[\|\hat{\theta}_{m(t)}(t) - \theta_{m(t)}^*\|_2 \mid E] + \frac{3M}{T}\Delta_{\max}$$

$$\leq \frac{12\sigma}{\lambda} \sqrt{\frac{d \log(dtT)}{t - T_0 - M}} + \frac{3M}{T}\Delta_{\max}.$$

Then, the Pareto regret is bounded by

$$\mathcal{PR}(T) = \sum_{t=2T_0+2M+1}^{T} \mathbb{E}[\Delta_{a(t)}(t)] + (2T_0 + 2M)\Delta_{\max}$$

$$\leq \sum_{t=2T_0+2M+1}^{T} \frac{12\sigma}{\lambda} \sqrt{\frac{d \log(dtT)}{t - T_0 - M}} + \{(\frac{3M}{T})T + 2T_0 + 2M\}\Delta_{\max}$$

$$\leq \frac{12\sigma}{\lambda} \sqrt{2d \log(dT)} \int_0^T \frac{1}{\sqrt{t}} dt + \{2T_0 + 5M\}\Delta_{\max}$$

$$\leq \frac{24\sigma}{\lambda} \sqrt{2dT \log(dT)} + 2\{2T_0 + 5M\}.$$

The last inequality holds because we have $\Delta_{\max} \leq 2$ under Assumption 1. $\qquad\square$

### D.3 PROOF OF THEOREM 2

*Proof.* Define the event $\Omega_{m,t}$ for all $m \in [M]$ as

$$\Omega_{m,t} := \{\omega \in \Omega \mid \text{Objective } m \text{ is a target objective for round } t\}.$$

Then, $\mathbb{P}(\Omega_{m,t}) = \mathbb{1}\{t \equiv m \mod M\}$ from the Round-Robin process.

Let $E$ be the event that $\left\|\hat{\theta}_m(t) - \theta_m^*\right\|_2 \leq \frac{6\sigma}{\lambda}\sqrt{\frac{d\log(dtT)}{t-T_0-M}}$ holds for all $m \in [M]$ and $t \geq 2T_0+2M$. Then, $\mathbb{P}(\bar{E}) \leq \frac{3M}{T}$ by Lemma 8 with $\delta = \frac{1}{T}$. We know that on $\Omega_{m,t} \cap E$, for $t \geq 2T_0 + 2m$,

$$\mu_m^* - \mu_{a(t),m} \leq 2\|\hat{\theta}_m(t) - \theta_m^*\|_2 \leq \frac{12\sigma}{\lambda}\sqrt{\frac{d\log(dtT)}{t-T_0-M}} \leq \frac{12\sigma}{\lambda}\sqrt{\frac{2d\log(dT)}{t-T_0-M}}.$$

Let $T_\epsilon = \max(\lfloor\frac{288\sigma^2 d\log(dT)}{\lambda^2\epsilon^2}\rfloor+T_0+M,\ 2T_0+2M)$. Then, on $\Omega_{m,t}\cap E$, we have $\mu_m^* - \mu_{a(t),m} < \epsilon$ for all $t > T_\epsilon$. Therefore, for all $m \in [M]$,

$$\frac{1}{T}\mathbb{E}\left[\sum_{t=1}^T \mathbb{1}\{\mu_m^* - \mu_{a(t),m} < \epsilon\}\right] = \frac{1}{T}\sum_{t=1}^T \mathbb{E}\left[\mathbb{1}\{\mu_m^* - \mu_{a(t),m} < \epsilon\}\right]$$

$$\geq \frac{1}{T}\sum_{t=1}^T \mathbb{E}[\mathbb{1}\{\mu_m^* - \mu_{a(t),m} < \epsilon\} \mid \Omega_{m,t}]\,\mathbb{P}(\Omega_{m,t})$$

$$\geq \frac{1}{T}\sum_{t=T_\epsilon+1}^T \mathbb{E}[\mathbb{1}\{\mu_m^* - \mu_{a(t),m} < \epsilon\} \mid \Omega_{m,t}]\,\mathbb{P}(\Omega_{m,t})$$

$$\geq \frac{1}{T}\sum_{t=T_\epsilon+1,\ M|t-m}^T \mathbb{E}[\mathbb{1}\{\mu_m^* - \mu_{a(t),m} < \epsilon\} \mid \Omega_{m,t} \cap E]\,\mathbb{P}(E)$$

$$\geq \frac{1}{T}\sum_{t=T_\epsilon+1,\ M|t-m}^T \mathbb{P}(E)$$

$$\geq \frac{1}{T}\left[\frac{T-T_\epsilon}{M}\right]\left(1 - \frac{3M}{T}\right)$$

$$\geq \left(\frac{T-T_\epsilon-M}{MT}\right)\left(1 - \frac{3M}{T}\right)$$

$\square$

## D.4 THE PARAMETER $B$ AND THE NUMBER OF INITIAL ROUNDS

In this section, we discuss the appropriate value of $B$, the threshold of the minimum eigenvalue of the Gram matrix. For convenience, denote $V_t := \sum_{s=1}^t x(s)x(s)^\top$ and $S_t := \sum_{s=1}^t x(s)\eta_{a(s)}(s)^\top$. When the minimum eigenvalue of the empirical covariance matrix $V_{T_0-1}$ exceeds a certain threshold, we can guarantee the $l_2$ bound of the OLS estimator $\hat{\theta}(t)$ of $\theta_*$ for $t \geq T_0$ with high probability. I.e.,

$$\lambda_{\min}(V_{T_0-1}) \geq f(a) \quad \Rightarrow \quad \text{for all } t \geq T_0, \quad \left\|\hat{\theta}(t) - \theta_*\right\|_2 \leq a \tag{1}$$

If we set $B = f(\alpha)$, then with high probability, $\|\hat{\theta}_m(t) - \theta_m^*\| \leq \alpha$ after initial rounds.

Kveton et al. (2020) suggest $f(a)$ that satisfies Eq.(1) using a bound of $\|S_t\|_{V_{t-1}^{-1}}$. However, a small mistake was made in their process: the bound they derived by modifying Theorem 1 of Abbasi-Yadkori et al. (2011) is actually a bound for $\|\sum_{s=\tau_0+1}^t x(s)\eta_{a(s)}(s)^\top\|_{V_{t-1}^{-1}}$, where $\tau_0 = \min\{t \geq 1 : V_t \succ 0\}$, not $\|S_t\|_{V_{t-1}^{-1}}$. To address this problem, the simplest approach would be to use the bound of $\|S_t\|_2$ suggested by Kannan et al. (2018). Alternatively, we can use the bound of $\|S_t\|_{V_{t-1}^{-1}}$ proposed by Li et al. (2017). The following lemma explains how the theoretical value of the initial parameter $B$, given by $\widetilde{\mathcal{O}}(\min(\sqrt{dT}, d\log T))$, can be derived through these two approaches.

**Lemma 9.** *Given Assumption 1, for any $a > 0$ and $\delta > 0$, if we run Algorithm 2 with*

$$B = \min\left[\frac{\sigma}{\alpha}\sqrt{2dT\log(\frac{dT}{\delta})},\ \frac{4\sigma^2}{\alpha^2}\left(\frac{d}{2}\log\left(1 + \frac{2T}{d}\right) + \log\left(\frac{1}{\delta}\right)\right)\right],$$

*then with probability at least $1 - 2M\delta$, the OLS estimator satisfies $\|\hat{\theta}_m(t) - \theta_m^*\|_2 \leq \alpha$ for all $m \in [M]$ and $t \geq T_0 + 1$.*

*Proof.* First we will bound $B$ using the fact

$$\left\|\hat{\theta}_m(t) - \theta_m^*\right\|_2 = \left\|(V_{t-1})^{-1}S_{t-1,m}\right\|_2 \leq \frac{1}{\lambda_{\min}(V_{t-1})}\|S_{t-1,m}\|_2,$$

where $S_{t,m} := \sum_{s=1}^{t} x(s)\eta_{a(s),m}(s)^\top$.

Since for fixed $m \in [M]$, $\|S_{t-1,m}\|_2 \leq \sigma\sqrt{2dt\ln(td/\delta)}$ holds for all $t \leq T$ with probability at least $1 - \delta$ by Lemma 23 and it is obvious that $\lambda_{\min}(V_{t-1}) \geq \lambda_{\min}(V_{T_0-1})$ for $t \geq T_0$, we have $\left\|\hat{\theta}_m(t) - \theta_m^*\right\|_2 \leq \alpha$ for all $m \in [M]$ and $t \geq T_0 + 1$ with probability at least $1 - M\delta$ when the value of $B$ set to $\frac{\sigma}{a}\sqrt{2dT\log(dT/\delta)}$.

Alternatively, we can use the fact

$$\left\|\hat{\theta}_m(t) - \theta_m^*\right\|_2^2 = (S_{t-1,m})^\top V_{t-1}^{-1}V_{t-1}^{-1}S_{t-1,m} \leq \frac{1}{\lambda_{\min}(V_{t-1})}\|S_{t-1,m}\|_{V_{t-1}^{-1}}^2.$$

By Lemma 24, for fixed $m \in [M]$, $\|S_{t-1,m}\|_{V_{t-1}^{-1}}^2 \leq 4\sigma^2(\frac{d}{2}\log(1 + \frac{2t}{d}) + \log(\frac{1}{\delta}))$ holds for all $t \leq T$ with probability at least $1 - \delta$, and hence, we have $\left\|\hat{\theta}_m(t) - \theta_m^*\right\|_2 < a$ for all $m \in [M]$ and $t \geq T_0 + 1$ with probability at least $1 - M\delta$ by setting $B$ to $\frac{4\sigma^2}{a^2}(\frac{d}{2}\log(1 + \frac{2T}{d}) + \log(\frac{1}{\delta}))$.

Therefore, if we set $B = \min\left[\frac{\sigma}{\alpha}\sqrt{2dT\log(\frac{dT}{\delta})},\ \frac{4\sigma^2}{\alpha^2}\left(\frac{d}{2}\log\left(1 + \frac{2T}{d}\right) + \log\left(\frac{1}{\delta}\right)\right)\right]$, we have $\left\|\hat{\theta}_m(t) - \theta_m^*\right\|_2 < a$ for all $m \in [M]$ and $t \geq T_0 + 1$ with probability at least $1 - 2M\delta$. $\qquad\square$

### D.4.1 Proof of Corollary 1

*Proof.* Let $S$ be the feature set selected during initial rounds and $\lambda_S := \lambda_{\min}\left(\frac{1}{M}\sum_{x_i \in S} x_i(x_i)^\top\right)$ Then, for any $T_1 \geq \lfloor\frac{B}{\lambda_S}\rfloor + M$, if we keep playing with feature vectors in $S$ in a Round-Robin manner for $T_1$ rounds,

$$\lambda_{\min}\left(\sum_{s=1}^{T_1-1} x(s)x(s)^\top\right) \geq \left[\frac{T_1 - 1}{M}\right] \times \lambda_S M \geq \lambda_S(T_1 - M) \geq B.$$

Hence, we have $T_0 \leq \lfloor\frac{B}{\lambda_S}\rfloor + M$. $\qquad\square$

## E   Randomized version of MOG algorithm

### E.1   Multi-Objective Greedy algorithm – Randomized version

We propose a randomized version of MOG algorithm named the MOG-R algorithm, which selects target objective randomly for each round (Line 3). The algorithm takes as input the probability mass function $(p_1, \ldots, p_M)$ of selecting each objective, which can be uniformly set to $\frac{1}{M}$ in the absence of specific information. The other aspects remain identical to the original MOG algorithm.

The MOG-R algorithm can be interpreted as a greedy algorithm operating in a multi-objective setting, where the prioritized objective changes in each round. The statistical guarantees of the MOG-R algorithm demonstrate that applying a greedy algorithm to the prioritized objective can be an efficient strategy when there exist good arms for multiple objectives. This suggests that in real-world scenarios where the dominant objective changes across rounds, an algorithm can still achieve strong performance in terms of regret, even when solely exploiting the dominant objective in each round.

---

**Algorithm 2** Multi-Objective Greedy algorithm – Randomized version (`MOG-R`)

---

**Require:** Total rounds $T$, Eigenvalue threshold $B$, Objective distribution $(p_1, \ldots, p_M)$
1: Initialize $V_0 \leftarrow 0 \times I_d$, and $\beta_1, \ldots, \beta_M \in \mathbb{R}^d$
2: **for** $t = 1$ **to** $T$ **do**
3:     Randomly select the target objective $m \in [M]$ from the distribution $(p_1, \ldots, p_M)$.
4:     **if** $\lambda_{\min}(V_{t-1}) < B$ **then**
5:         Select action $a(t) \in \arg\max_{i \in [K]} x_i^\top \beta_m$
6:     **else**
7:         Update the OLS estimators $\hat{\theta}_1(t), \ldots, \hat{\theta}_M(t)$
8:         Select action $a(t) \in \arg\max_{i \in [K]} x_i^\top \hat{\theta}_m(t)$
9:     **end if**
10:   Observe $y(t) = \big(y_{a(t),1}(t), \ldots, y_{a(t),M}(t)\big)$
11:   Update $V_t \leftarrow V_{t-1} + x(t)x(t)^\top$
12: **end for**

---

### E.2 PARETO REGRET BOUND OF `MOG-R`

The following theorem demonstrates that the `MOG-R` algorithm possesses near optimal regret with respect to $T$.

**Theorem 3** (Pareto regret bound of `MOG-R`). *Suppose Assumptions 1, 2, and 3 hold. If we run Algorithm 1 with $B = \min\left[\frac{\sigma}{\alpha}\sqrt{2dT\log(dT^2)},\ \frac{4\sigma^2}{\alpha^2}\left(\frac{d}{2}\log\left(1 + \frac{2T}{d}\right) + \log(T)\right)\right]$, then the Pareto regret of Algorithm 2 is bounded by*

$$\mathcal{PR}(T) \leq \frac{48\sigma}{\lambda p^*}\sqrt{2dT\log(dT)} + 4T_0 + 6M + \frac{60d}{\lambda p^*},$$

*where $p^* = \min_{m \in [M]}(p_m)M$.*

**Discussion of Theorem 3.** The theorem establishes that `MOG-R` has $\widetilde{\mathcal{O}}\big(\frac{\sqrt{dT}}{\lambda}\big)$ Pareto regret bound, which matches the bound for the original deterministic version of `MOG`. In other words, this implies that even when the target objective in `MOG` is determined stochastically, a similar level of statistical guarantee can be maintained.

**Remark 6.** *The value of $p^*$ becomes smaller as the probability differences among the objectives selected by the algorithm increase. Conversely, if a uniform distribution is used for selecting the target objective, $p^*$ takes a value of $1$.*

**Corollary 3** (Number of initial rounds). *Suppose Assumptions 1, 2, and 3 hold. If the feature set $S$ selected during the initial rounds in Algorithm 2 spans $\mathbb{R}^d$, then $T_0$ can be bounded by $T_0 \leq \left\lfloor 2B/p^*\lambda_{\min}\left(\frac{1}{M}\sum_{x_i \in S} x_i(x_i)^\top\right)\right\rfloor$.*

The proof of Theorem 3 is presented in Section E.2.2 and its supporting lemmas are presented in Section E.2.1, and the proof of corollary 3 is presented in Section E.2.3

### E.2.1 TECHNICAL LEMMAS FOR THEOREM 3

To prove Theorem 3, we first establish the lower bound of the minimum eigenvalue of Gram matrix that increases linearly with respect to $t$, in a slightly different way from the case of `MOG`. In the previous analysis for `MOG`, we construct a constant lower bound for the increment of minimum eigenvalue during one round robin cycle. For the randomized version, we make a constant lower bound for $\lambda_{\min}\big(\mathbb{E}[x(t)x(t)^\top | \mathcal{H}_{t-1}]\big)$ in each round, like existing greedy bandit approaches. However, it is important to note that the expectation of the lemma below arises not from the randomness of the contexts, but rather from the randomness associated with the selection of the target objective in each round.

**Lemma 10** (Increment of the minimum eigenvalue of the Gram matrix). *Suppose Assumptions 1, 2, and 3 hold. If the OLS estimator satisfies $\|\hat{\theta}_m(s) - \theta_m^*\| \leq \alpha$, for all $m \in [M]$ and $s \geq T_0 + 1$, then the arm selected by Algorithm 2 satisfies*

$$\lambda_{\min}(\mathbb{E}[x(s)x(s)^\top | \mathcal{H}_{s-1}]) \geq \frac{\lambda p^*}{3},$$

*where $p^* = \min_{m \in [M]}(p_m)M$.*

*Proof.* For all $s \geq T_0 + 1$ and $m \in [M]$, let $E_m(s)$ be the event that the objective $m$ is a target objective in round $s$. Then,

$$\mathbb{E}[x(s)x(s)^\top | \mathcal{H}_{s-1}] = \sum_{m=1}^{M} \mathbb{E}[x(s)x(s)^\top | E_m(s), \mathcal{H}_{s-1}]\mathbb{P}[E_m(s)|\mathcal{H}_{s-1}]$$

$$= \sum_{m=1}^{M} p_m \mathbb{E}[x(s)x(s)^\top | E_m(s), \mathcal{H}_{s-1}]$$

$$\succeq \min_{m \in [M]}(p_m) \sum_{m=1}^{M} \mathbb{E}[x(s)|E_m(s), \mathcal{H}_{s-1}]\mathbb{E}[x(s)|E_m(s), \mathcal{H}_{s-1}]^\top.$$

, The final line is validated by Lemma 27.

By Assumption 3, there always exists $\gamma$-good arm for $\hat{\theta}_m(s)$ for all $m \in [M]$. Hence, on the event $E_m(s)$, the selected arm $x(s)$ is $\gamma$-good for $\hat{\theta}_m(s)$ by Proposition 1. Therefore, $\mathbb{E}[x(s)|E_m(s), \mathcal{H}_{s-1}]$ is also $\gamma$-good for $\hat{\theta}_m(s)$ by Proposition 2, and so we can apply Lemma 6 to derive the minimum eigen value of above matrix by

$$\lambda_{\min}(\mathbb{E}[x(s)x(s)^\top | \mathcal{H}_{s-1}]) \geq \min_{m \in [M]}(p_m) \lambda_{\min}\left(\sum_{m=1}^{M} \mathbb{E}[x(s)|E_m(s), \mathcal{H}_{s-1}]\mathbb{E}[x(s)|E_m(s), \mathcal{H}_{s-1}]^\top\right)$$

$$\geq \frac{\min_{m \in [M]}(p_m)\lambda M}{3}.$$

$\square$

The following lemma shows that the minimum eigenvalue of the Gram matrix increases at a rate $O(\lambda)$.

**Lemma 11** (Minimum eigenvalue growth of Gram matrix). *Suppose Assumptions 1, 2, and 3 hold. Assume the OLS estimator satisfies $\|\hat{\theta}_m(s) - \theta_m^*\| \leq \alpha$ for all $m \in [M]$ and $s \geq T_0 + 1$. Then for $t \geq T_0$, the following holds for the minimum eigenvalue of the Gram matrix of arms selected by Algorithm 1*

$$\mathbb{P}\left[\lambda_{\min}\left(\sum_{s=1}^{t} x(s)x(s)^\top\right) \leq B + \frac{\lambda q^*}{6}(t - T_0)\right] \leq de^{\frac{-\lambda q^*(t-T_0)}{30}},$$

*where $C = \lambda - 2\sqrt{2 + 2\alpha\sqrt{1 - \gamma^2} - 2\gamma\sqrt{1 - \alpha^2}}$ and $p^* = \min_{m \in [M]}(p_m)M$.*

*Proof.* By the subadditivity of minimum eigenvalue and Lemma 10, for $t \geq T_0 + 1$,

$$\lambda_{\min}\left(\sum_{s=T_0+1}^{t} \mathbb{E}[x(s)x(s)^\top | \mathcal{H}_{s-1}]\right) \geq \sum_{s=T_0+1}^{t} \lambda_{\min}(\mathbb{E}[x(s)x(s)^\top | \mathcal{H}_{s-1}]) \geq \frac{\lambda p^*}{3}(t - T_0)$$

In other words, $\mathbb{P}[\lambda_{\min}(\sum_{s=T_0+1}^{t} \mathbb{E}[x(s)x(s)^\top | \mathcal{H}_{s-1}]) \geq \frac{\lambda p^*}{3}(t - T_0)] = 1$ holds for $t \geq T_0 + 1$. By applying Lemma25 to compute the lower bound of the minimum eigenvalue of the Gram matrix after exploration, we have

$$\mathbb{P}\left[\lambda_{\min}\left(\sum_{s=T_0+1}^{t} x(s)x(s)^\top\right) \leq \frac{\lambda p^*}{6}(t - T_0)\right] \leq d(\frac{e^{0.5}}{0.5^{0.5}})^{-\frac{\lambda p^*}{3}(t-T_0)} \leq de^{\frac{-\lambda p^*(t-T_0)}{30}}.$$

Therefore, by subadditivity of minimum eigenvalue, for $t \geq T_0$

$$\mathbb{P}\left[\lambda_{\min}\left(\sum_{s=1}^{t} x(s)x(s)^\top\right) \leq B + \frac{\lambda p^*}{6}(t - T_0)\right] \leq de^{\frac{-\lambda p^*(t-T_0)}{30}}.$$

$\square$

The following lemma establishes the $l_2$-bound for the estimated objective parameters, a critical requirement for solving greedy bandit problems.

**Lemma 12.** *Suppose Assumptions 1, 2, and 3 hold. Assume the OLS estimator satisfies $\|\hat{\theta}_m(s) - \theta_m^*\| \leq \alpha$ for all $m \in [M]$ and $s \geq T_0 + 1$ for some $\alpha > 0$. Then for any $\delta > 0$, $m \in [M]$ and $t \geq T_0$, with probability at least $1 - M\delta - de^{\frac{-\lambda p^*(t-T_0)}{30}}$, the OLS estimator $\hat{\theta}_m(t)$ of $\theta_m^*$ satisfies*

$$\left\|\hat{\theta}_m(t+1) - \theta_m^*\right\|_2 \leq \frac{12\sigma}{\lambda p^*}\sqrt{\frac{d\log(dt/\delta)}{t - T_0}},$$

*where $p^* = \min_{m \in [M]}(p_m)M$.*

*Proof.* From the closed form of the OLS estimators, for all $m \in [M]$,

$$\left\|\hat{\theta}_m(t+1) - \theta_m^*\right\|_2 = \left\|\left(\sum_{s=1}^{t} x(s)x(s)^\top\right)^{-1}\sum_{s=1}^{t} x(s)\eta_{a(s),m}(s)\right\|_2$$

$$\leq \frac{1}{\lambda_{\min}\left(\sum_{s=1}^{t} x(s)x(s)^\top\right)}\left\|\sum_{s=1}^{t} x(s)\eta_{a(s),m}(s)\right\|_2$$

For the denominator, we have $\lambda_{\min}(V_t) \geq B + \frac{\lambda p^*}{6}(t - T_0)$ for $t \geq T_0$, with probability at least $1 - de^{\frac{-\lambda p^*(t-T_0)}{30}}$, by Lemma 7. To bound the $l_2$ norm of $S_{t,m} := \sum_{s=1}^{t} x(s)\eta_{a(s),m}(s)$, we can use Lemma 23, the martingale inequality of Kannan et al. (2018). The lemma states for fixed $m \in [M]$, $\|S_{t,m}\|_2 \leq \sigma\sqrt{2dt\log(dt/\delta)}$ holds with probability at least $1 - \delta$. Therefore, with probability at least $1 - M\delta - de^{\frac{-\lambda p^*(t-T_0)}{30}}$, for all $m \in [M]$ and $t \geq 2T_0$,

$$\left\|\hat{\theta}_m(t+1) - \theta_m^*\right\|_2 \leq \frac{\sigma\sqrt{2dt\log(dt/\delta)}}{B + \lambda p^*(t - T_0)/6} \leq \frac{12\sigma}{\lambda p^*}\sqrt{\frac{d\log(dt/\delta)}{t - T_0}}.$$

The last inequality holds when $t \geq 2T_0$. $\qquad\square$

### E.2.2 PROOF OF THEOREM 3

*Proof.* By Lemma 9, if $B$ is set by $B = \min\left[\frac{\sigma}{\alpha}\sqrt{2dT\log(dT^2)},\ \frac{4\sigma^2}{\alpha^2}\left(\frac{d}{2}\log\left(1 + \frac{2T}{d}\right) + \log(T)\right)\right]$, we have $\|\hat{\theta}_m(t) - \theta_m^*\| \leq \alpha$ for all $m \in [M]$ and $t \geq T_0 + 1$ with probability at least $1 - \frac{2M}{T}$. Let $E$ be the event that $\|\hat{\theta}_m(t+1) - \theta_m^*\| \leq \frac{12\sigma}{\lambda p^*}\sqrt{\frac{d\log(dtT)}{t-T_0}}$ holds for all $t \geq 2T_0$ and $m \in [M]$. Then, $\mathbb{P}(\bar{E}) \leq \frac{2M}{T} + \frac{M}{T} + de^{\frac{-\lambda p^*(t-T_0)}{30}}$ by Lemma 12.

Let $\Delta_{\max}$ be the maximum suboptimality gap and $m(t)$ be the target objective in round $t$. For $t \geq 2T_0$,

$$\mathbb{E}[\Delta_{a(t+1)}(t+1)] \leq \mathbb{E}[\Delta_{a(t+1)}(t+1) \mid E] + \mathbb{P}(E)\Delta_{\max}$$

$$\leq 2\mathbb{E}[\ \|\hat{\theta}_{m(t+1)}(t+1) - \theta_{m(t+1)}^*\|_2 \mid E] + \left(\frac{3M}{T} + de^{\frac{-\lambda p^*(t-T_0)}{30}}\right)\Delta_{\max}$$

$$\leq \frac{24\sigma}{\lambda p^*}\sqrt{\frac{d\log(dtT)}{t - T_0}} + \left(\frac{3m}{T} + de^{\frac{-\lambda p^*(t-T_0)}{30}}\right)\Delta_{\max}.$$

Then, the Pareto regret is bounded by

$$
\mathcal{PR}(T) = \sum_{t=2T_0}^{T-1} \mathbb{E}[\Delta_{a(t+1)}(t+1)] + 2T_0\Delta_{\max}
$$

$$
\leq \sum_{t=2T_0}^{T} \frac{24\sigma}{\lambda p^*} \sqrt{\frac{d\log(dtT)}{t-T_0}} + \{(\frac{3M}{T})T + \sum_{t=2T_0}^{T} de^{\frac{-\lambda p^*(t-T_0)}{30}} + 2T_0\}\Delta_{\max}
$$

$$
\leq \frac{24\sigma}{\lambda p^*} \sqrt{2d\log(dT)} \int_0^T \frac{1}{\sqrt{t}}dt + \left(2T_0 + 3M + \sum_{t=2T_0}^{T} de^{\frac{-\lambda p^*(t-T_0)}{30}}\right)\Delta_{\max}
$$

$$
\leq \frac{48\sigma}{\lambda p^*} \sqrt{2dT\log(dT)} + \left(2T_0 + 3M + \frac{30d}{\lambda p^*}\right)\Delta_{\max}
$$

$$
\leq \frac{48\sigma}{\lambda p^*} \sqrt{2dT\log(dT)} + 2\left(2T_0 + 3M + \frac{30d}{\lambda p^*}\right).
$$

The last inequality holds because we have $\Delta_{\max} \leq 2$ under Assumption 1. $\qquad\square$

### E.2.3  PROOF OF COROLLARY 3

*Proof.* Let $S$ be the feature set selected during initial rounds and $\lambda_S := \lambda_{\min}\left(\frac{1}{M}\sum_{x_i \in S} x_i(x_i)^\top\right)$. Then, $\lambda_{\min}\left(\sum_{s=1}^{t} x(s)x(s)^\top\right) \geq p^*\lambda_S t/2$ with probability at least $1 - de^{-\lambda_S p^*/10}$ by Lemma 25.

Then, for any $T_0 \geq \lfloor \frac{2B}{p^*\lambda_S}\rfloor$, if we keep playing with the initial values for $T_0$ rounds,

$$
\lambda_{\min}\left(\sum_{s=1}^{T_0} x(s)x(s)^\top\right) \geq \frac{p^*\lambda_S}{2}\lfloor\frac{2B}{p^*\lambda_S}\rfloor \geq B.
$$

Hence, we have $T_0 \leq \lfloor\frac{2B}{p^*\lambda_S}\rfloor$ with probability at least $1 - de^{-\lambda p^*/30}$. $\qquad\square$

### E.3  OBJECTIVE FAIRNESS OF MOG-R

We confirmed that the MOG-R algorithm satisfies the objective fairness. The following theorem shows the lower bound on the objective fairness index.

**Theorem 4** (Objective fairness of MOG-R). *Suppose Assumptions 1, 2, and 3 hold. Then, the objective fairness index of Algorithm 1 satisfies for all $m \in [M]$,*

$$
\mathrm{OFI}_{\epsilon,T} \geq \min_{m\in[M]}(p_m)\left(\frac{T - T_\epsilon}{T}\right)\left(1 - \frac{3M}{T} - d(\frac{1}{dT})^{\frac{40\sigma^2 d}{\lambda p^*\epsilon^2}}\right),
$$

*where $T_\epsilon = \max(\lfloor\frac{1152\sigma^2 d\log(dT)}{\lambda^2(p^*)^2\epsilon^2}\rfloor + T_0,\ 2T_0)$ in the same setting as Theorem 3.*

**Discussion of Theorem 4.** The theorem demonstrates that Algorithm 2 satisfies objective fairness, since for any given $\epsilon > 0$, $\lim_{T\to\infty}\mathrm{OFI}_{\epsilon,T} = \min_{m\in[M]}(p_m)$. We show that with high probability, our algorithm selects near-optimal arms for each objective $m$ at a ratio of $p_m$ as time grows, and it selects only $\epsilon$-optimal arms of an objective after a certain rounds $T_\epsilon$.

*Proof.* Define the event $\Omega_{m,t}$ for all $m \in [M]$ as

$$
\Omega_{m,t} := \{\omega \in \Omega \mid \text{Objective } m \text{ is a target objective for round } t\}.
$$

Then, $\mathbb{P}(\Omega_{m,t}) = p_m$ for all $m \in [M]$ and $t \leq T$.

Let $E$ be the event that $\|\hat{\theta}_m(t+1) - \theta_m^*\| \leq \frac{12\sigma}{\lambda p^*}\sqrt{\frac{d\log(dtT)}{t-T_0}}$ holds for all $t \geq 2T_0$ and $m \in [M]$. Then, $\mathbb{P}(\bar{E}) \leq \frac{3M}{T} + de^{\frac{-\lambda p^*(t-T_0)}{30}}$ by Lemma 9 and Lemma 12.

We know that for $t \geq 2T_0$, on $\Omega_{m,t+1} \cap E$,

$$\mu_m^* - \mu_{a(t+1),m} \leq 2\|\hat{\theta}_m(t+1) - \theta_m^*\|_2 \leq \frac{24\sigma}{\lambda p^*}\sqrt{\frac{d\log(dtT)}{t-T_0}} \leq \frac{24\sigma}{\lambda p^*}\sqrt{\frac{2d\log(dT)}{t-T_0}}.$$

Let $T_\epsilon = \max(\lfloor \frac{1152\sigma^2 d\log(dT)}{\lambda^2 (p^*)^2 \epsilon^2} \rfloor + T_0, \ 2T_0)$. Then, on $\Omega_{m,t+1} \cap E$, we have $\mu_m^* - \mu_{a(t+1),m} < \epsilon$ for all $t > T_\epsilon$.

Then, for all $m \in [M]$,

$$\frac{1}{T}\mathbb{E}\left[\sum_{t=1}^{T} \mathbb{1}\{\mu_m^* - \mu_{a(t),m} < \epsilon\}\right] \geq \frac{1}{T}\sum_{t=1}^{T}\mathbb{E}[\mathbb{1}\{\mu_m^* - \mu_{a(t),m} < \epsilon\} \mid \Omega_{m,t}] \, \mathbb{P}(\Omega_{m,t})$$

$$\geq \frac{p_m}{T}\sum_{t=T_\epsilon}^{T-1}\mathbb{E}[\mathbb{1}\{\mu_m^* - \mu_{a(t+1),m} < \epsilon\} \mid \Omega_{m,t+1} \cap E] \, \mathbb{P}(E)$$

$$\geq \frac{p_m}{T}\sum_{t=T_\epsilon}^{T-1} 1 \cdot \mathbb{P}(E)$$

$$\geq \frac{p_m}{T}\sum_{t=T_\epsilon}^{T-1}\left(1 - \frac{3M}{T} - de^{\frac{-\lambda p^*(T_\epsilon - T_0)}{30}}\right)$$

$$\geq \frac{p_m}{T}(T - T_\epsilon)\left(1 - \frac{3M}{T} - d(\frac{1}{dT})^{\frac{40\sigma^2 d}{\lambda p^* \epsilon^2}}\right).$$

Therefore, the objective fairness index can be bounded by

$$\text{OFI}_{\epsilon,T} \geq \min_{m \in [M]}(p_m)\left(\frac{T - T_\epsilon}{T}\right)\left(1 - \frac{3M}{T} - d(\frac{1}{dT})^{\frac{40\sigma^2 d}{\lambda p^* \epsilon^2}}\right),$$

$\qquad\qquad\qquad\qquad\qquad\qquad\qquad\qquad\qquad\qquad\qquad\qquad\qquad\qquad\qquad\qquad\qquad\quad\square$

# F   LINEAR SCALARIZED VERSION OF MOG ALGORITHM

## F.1   MULTI-OBJECTIVE GREEDY ALGORITHM – WEIGHTED RANDOMIZED VERSION

We propose another multi-objective near-greedy algorithm, named MOG-WR. While both MOG and MOG-R focus solely on selecting optimal arms in specific objective directions, MOG-WR extends this by also considering optimal arms in weighted objective directions. The algorithm takes as input a distribution $\mathcal{D}$ from which the weight vectors are sampled. In each round, the algorithm selects the arm that maximizes the weighted estimated reward based on the weight vector $w$ drawn from $\mathcal{D}$ (Line 5, 8). The rest of the algorithm structure remains identical to the original MOG.

The MOG-R algorithm can be regarded as a special case of the MOG-WR algorithm, where the distribution $\mathcal{D}$ is set to

$$\mathbb{P}_{w \sim \mathcal{D}}(w) := \begin{cases} p_m & \text{if } w = e_m^{(M)}, \\ 0 & \text{otherwise.} \end{cases}$$

The MOG-WR algorithm, like previously proposed scalarized multi-objective bandit algorithms, selects the optimal arms corresponding to the reward functions generated in each round (Drugan & Nowe, 2013; Yahyaa & Manderick, 2015; Zhang, 2024). We confirm that even with the application of weighted scalarization, the greedy algorithm performs effectively, through both theoretical and empirical validation, where good arms exist for multiple objectives. Additionally, we prove that the MOG-WR algorithm satisfies generalized objective fairness.

## F.2   REGULARITY INDICES

Before we start analysis, we first define two regularity indices of a distribution for weight vectors.

---

**Algorithm 3** Multi-Objective Greedy algorithm – Weighted Randomized version (`MOG-WR`)

---

**Require:** Total rounds $T$, Eigenvalue threshold $B$, Weight distribution $\mathcal{D}$
1: Initialize $V_0 \leftarrow 0 \times I_d$, and $\beta_1, \ldots, \beta_M \in \mathbb{R}^d$
2: **for** $t = 1$ **to** $T$ **do**
3:     Sample a weight vector $w = (w_1, \ldots, w_M)$ from the distribution $\mathcal{D}$.
4:     **if** $\lambda_{\min}(V_{t-1}) < B$ **then**
5:         Select action $a(t) \in \arg\max_{i \in [K]} \left( \sum_{m \in [M]} w_m x_i^\top \beta_m \right)$
6:     **else**
7:         Update the OLS estimators $\hat{\theta}_1(t), \ldots, \hat{\theta}_M(t)$
8:         Select action $a(t) \in \arg\max_{i \in [K]} \left( \sum_{m \in [M]} w_m x_i^\top \hat{\theta}_m(t) \right)$
9:     **end if**
10:    Observe the reward vector $y(t) = \left( y_{a(t),1}(t), \ldots, y_{a(t),M}(t) \right)$
11:    Update $V_t \leftarrow V_{t-1} + x(t)x(t)^\top$
12: **end for**

---

**Definition 10** (Regularity indices of a distribution)**.** *Let $\mathcal{D}$ be a distribution on $M$-dimensional simplex, $\Delta^M = \{(w_1, \ldots, w_M) \in \mathbb{R}^d | \sum_{m \in [M]} w_i = 1, \ w_1, \ldots w_M \geq 0\}$. For given $\epsilon > 0$, We define the two regularity indices of distribution $\mathcal{D}$, $V_{\epsilon,\mathcal{D}}$ and $I_{\epsilon,\mathcal{D}}$ as*

$$V_{\epsilon,\mathcal{D}} := \min_{\bar{m} \in [M]} \mathbb{P}_{w \sim \mathcal{D}} \left( \left\| \sum_{m \in [M]} w_m \theta_m^* - \theta_{\bar{m}}^* \right\| < \epsilon \right)$$

$$I_{\epsilon,\mathcal{D}} := \inf_{\bar{w} \in \Delta^M} \mathbb{P}_{w \sim \mathcal{D}} \left( \left\| \sum_{m \in [M]} w_m \theta_m^* - \sum_{m \in [M]} \bar{w}_m \theta_m^* \right\| < \epsilon \right).$$

Intuitively, the regularity indices described above explain how evenly the weight distribution generates weighted objectives. Specifically, $V_{\epsilon,\mathcal{D}}$ measures whether the weighted objectives are well-sampled near the parameter space of each objective, while $I_{\epsilon,\mathcal{D}}$ captures how uniformly all possible weighted objectives are sampled. By definition, it is straightforward to confirm that $V_{\epsilon,\mathcal{D}} \geq I_{\epsilon,\mathcal{D}}$ always holds.

The following lemma demonstrates that for any continuous distribution $\mathcal{D}$ with positive density function, the regularity indices are always positive.

**Lemma 13.** *If $\mathcal{D}$ has a continuous density function $f$ which is positive on $\Delta^M$, then both regularity indices $V_{\epsilon,\mathcal{D}}$ and $I_{\epsilon,\mathcal{D}}$ are positive.*

*Proof sketch.* It is enough to show $I_{\epsilon,\mathcal{D}} > 0$. Fix $\epsilon > 0$ and define $g : \Delta^M \to \mathbb{R}^d$ such that $g(\bar{w}) := \mathbb{P}_{w \sim \mathcal{D}} \left( \left\| \sum_{m \in [M]} w_m \theta_m^* - \sum_{m \in [M]} \bar{w}_m \theta_m^* \right\| < \epsilon \right)$. Then, we can show that $g$ is a positive continuous function. Since $\Delta^M$ is compact, we have $\inf_{w \in \Delta^M} g(w) = \min_{w \in \Delta^M} g(w) > 0$.

### F.3 Pareto regret bound of `MOG-WR`

The following corollary shows that it is possible to achieve a $\widetilde{\mathcal{O}}(\sqrt{T})$ regret bound Algorithm 3, if the weight distribution $\mathcal{D}$ satisfies $V_{\alpha/2,\mathcal{D}} > 0$.

**Corollary 4** (Pareto regret bound of `MOG-WR`)**.** *Suppose Assumptions 1, 2, and 3 hold. If we run Algorithm 3 with $B = \min \left[ \frac{2\sigma}{\alpha} \sqrt{2dT \log(dT^2)}, \ \frac{16\sigma^2}{\alpha^2} \left( \frac{d}{2} \log \left( 1 + \frac{2T}{d} \right) + \log(T) \right) \right]$, then the Pareto regret of Algorithm 3 is bounded by*

$$\mathcal{PR}(T) \leq \frac{48\sigma}{\lambda v^*} \sqrt{2dT \log(dT)} + 4T_0 + 6M + \frac{60d}{\lambda v^*},$$

*where $v^* = V_{\alpha/2,\mathcal{D}} M$.*

We can establish the regret bound for `MOG-WR` using the same arguments employed for the regret bound of `MOG-R`, with the aid of the following two lemmas. The first lemma pertains to the linear growth of the minimum eigenvalue of the Gram matrix.

**Lemma 14** (Increment of the minimum eigenvalue of the Gram matrix). *Suppose Assumptions 1, 2, and 3 hold. If the OLS estimator satisfies $\|\hat{\theta}_m(s) - \theta_m^*\| \leq \alpha$, for all $m \in [M]$ and $s \geq T_0 + 1$, then the arm selected by Algorithm 3 satisfies*

$$\lambda_{\min}(\mathbb{E}[x(s)x(s)^\top | \mathcal{H}_{s-1}]) \geq \frac{\lambda v^*}{3},$$

*where $v^* = V_{\alpha/2,\mathcal{D}} M$.*

*Proof.* For $s \geq T_0 + 1$ and $m \in [M]$, let $E'_{\bar{m}}(s)$ be the event that the weighted objective $\sum_{m \in [M]} w_m \theta_m^*$ in round $s$ satisfies $\left\| \sum_{m \in [M]} w_m \theta_m^* - \theta_{\bar{m}}^* \right\| < \alpha/2$. Then, on $E'_{\bar{m}}(s)$, if $\|\hat{\theta}_m(s) - \theta_m^*\| \leq \alpha/2$ holds, then the following holds.

$$\left\| \sum_{m \in [M]} w_m \hat{\theta}_m(t) - \theta_{\bar{m}}^* \right\| \leq \left\| \sum_{m \in [M]} w_m \hat{\theta}_m(t) - \sum_{m \in [M]} w_m \theta_m^* \right\| + \left\| \sum_{m \in [M]} w_m \theta_m^* - \theta_{\bar{m}}^* \right\|$$

$$\leq \sum_{m \in [M]} w_m \left\| \hat{\theta}_m(t) - \theta_m^* \right\| + \left\| \sum_{m \in [M]} w_m \theta_m^* - \theta_{\bar{m}}^* \right\|$$

$$< \frac{\alpha}{2} + \frac{\alpha}{2} = \alpha$$

Thus, by Assumption 3, there exists $\gamma$-good arm for the weighted objective $\sum_{m \in [M]} w_m \hat{\theta}_m(t)$ in round $t$.

Since the arm selected by Algorithm 3 satisfies

$$\mathbb{E}[x(s)x(s)^\top | \mathcal{H}_{s-1}] = \sum_{m=1}^{M} \mathbb{E}[x(s)x(s)^\top | E'_m(s), \mathcal{H}_{s-1}] \mathbb{P}[E'_m(s) | \mathcal{H}_{s-1}]$$

$$\succeq V_{\alpha/2,\mathcal{D}} \sum_{m=1}^{M} \mathbb{E}[x(s)x(s)^\top | E'_m(s), \mathcal{H}_{s-1}]$$

$$\succeq V_{\alpha/2,\mathcal{D}} \sum_{m=1}^{M} \mathbb{E}[x(s) | E'_m(s), \mathcal{H}_{s-1}] \mathbb{E}[x(s) | E'_m(s), \mathcal{H}_{s-1}]^\top,$$

, and $\mathbb{E}[x(s) | E'_m(s), \mathcal{H}_{s-1}]$ is $\gamma$-good for the weighted objective $\sum_{m \in [M]} w_m \hat{\theta}_m(s)$ in round $s$, we have

$$\lambda_{\min}(\mathbb{E}[x(s)x(s)^\top | \mathcal{H}_{s-1}]) \geq V_{\alpha/2,\mathcal{D}} \, \lambda_{\min} \left( \sum_{m=1}^{M} \mathbb{E}[x(s) | E'_m(s), \mathcal{H}_{s-1}] \mathbb{E}[x(s) | E'_m(s), \mathcal{H}_{s-1}]^\top \right)$$

$$\geq \frac{V_{\alpha/2,\mathcal{D}} \lambda M}{3},$$

by Lemma 6. □

Then, we can drive $l_2$ bound of $\hat{\theta}_m(t) - \theta_m^*$ with above lemma. The next lemma shows how to bound the Pareto regret with the bound on $\|\hat{\theta}_m(t) - \theta_m^*\|_2$.

**Lemma 15.** *Given Assumption 1, for all round $t$, the Pareto suboptimality gap of Algorithm 3 can be bounded by $\Delta_{a(t)}(t) \leq 2 \left\| \sum_{m \in [M]} w_m \hat{\theta}_m(t) - \sum_{m \in [M]} w_m \theta_m^* \right\|_2$, where $w$ is the generated weight vector in round $t$. Furthermore, if there exists an upper bound $U$ that satisfies $\|\hat{\theta}_m(t) - \theta_m^*\|_2 < U$ for all $m \in [M]$, we have $\Delta_{a(t)}(t) \leq 2U$.*

*Proof.* Fix round $t \in [T]$. Let $w \in \Delta^M$ be the generated weight vector in round $t$, and $a_w^*$ be the true optimal arm for the weighted objective $\sum_{m \in [M]} w_i \theta_m^*$. Then, by Corollary 2, $a_w^*$ is in the Pareto

Front, and so we have

$$\Delta_{a(t)}(t) \leq \min_{m \in [M]} \left( x_{a_w^*}^\top \theta_m^* - x_{a(t)}^\top \theta_m^* \right) \leq \sum_{m \in [M]} \left( w_m x_{a_w^*}^\top \theta_m^* - w_m x_{a(t)}^\top \theta_m^* \right)$$

$$= x_{a_w^*}^\top \left( \sum_{m \in [M]} w_m \theta_m^* \right) - x_{a(t)}^\top \left( \sum_{m \in [M]} w_m \theta_m^* \right)$$

$$\leq 2 \left\| \sum_{m \in [M]} w_m \hat{\theta}_m(t) - \sum_{m \in [M]} w_m \theta_m^* \right\|_2,$$

with Assumption 1.

The latter part of the lemma can be directly derived using the triangle inequality. $\qquad\square$

### F.4 OBJECTIVE FAIRNESS OF MOG-WR

**Corollary 5** (Generalized objective fairness of MOG-WR). *Suppose Assumptions 1, 2, and 3 hold. Then, the objective fairness index of Algorithm 3 satisfies for all $m \in [M]$,*

$$\text{GOFI}_{\epsilon,T} \geq I_{\alpha/2,\mathcal{D}} \left( \frac{T - T_\epsilon}{T} \right) \left( 1 - \frac{3M}{T} - d \left( \frac{1}{dT} \right)^{\frac{40\sigma^2 d}{\lambda v^* \epsilon^2}} \right),$$

*where $T_\epsilon = \max(\lfloor \frac{1152\sigma^2 d \log(dT)}{\lambda^2 (v^*)^2 \epsilon^2} \rfloor + T_0, \; 2T_0)$ and $v^* = V_{\alpha/2,\mathcal{D}} M$ in the same setting as Theorem 4.*

We can prove Corollary 5 using the same approach as in MOG-R, based on Lemma 14 and the definition of the index $I_{\epsilon,\mathcal{D}}$.

## G STOCHASTIC CONTEXTS SETUP

We verified that our proposed algorithms are statistically efficient even in stochastic context settings. In this section, we demonstrate the Pareto regret bound and objective fairness of MOG algorithm in a stochastic context setting. Notably, MOG-R and MOG-WR can also be analyzed theoretically using the same approach.

### G.1 PROBLEM SETTING

In multi-objective linear contextual bandit under stochastic contexts setup, the set of feature vectors $\chi(t) = \{x_i(t) \in \mathbb{R}^d, i \in [K]\}$ is drawn from some unknown distribution $P_\chi(t)$ in each round $t = 1, \ldots, T$. Each arm's feature $x_i(t) \in \chi(t)$ for $i \in [K]$ need not be independent of each other and can possibly be correlated. In this case, we denote $x_{a(t)}(t)$ as $x(t)$. Other settings are identical to the fixed arms case in Section 2.2.

**Pareto regret metric**  Pareto regret can be defined in the same way as in the fixed-arm case (Tekin & Turgay, 2018; Turgay et al., 2018; Lu et al., 2019; Cheng et al., 2024). The key difference is that in the fixed-arm setting, each arm's expected reward remains constant over time, and hence the Pareto front does not change. In contrast, in the contextual setup, the expected reward of each arm varies across rounds, and consequently the Pareto front also evolves. Therefore, the definition of Pareto regret is taken with respect to the Pareto front at each round.

**Definition 11** (Pareto front). *Let $\mu_i(t) \in \mathbb{R}^M$ be the expected reward vector of arm $i \in [K]$ in round $t$. Then, arm $i$ is Pareto optimal if and only if $\mu_i(t)$ is not dominated by $\mu_{i'}(t)$ for all $i' \in [K]$. The Pareto front is the set of all Pareto optimal arms in round $t$.*

**Definition 12** (Pareto regret). *We denote **Pareto suboptimality gap** $\Delta_i(t)$ for arm $i \in [K]$ as the infimum of the scalar $\epsilon \geq 0$ such that $\mu_i(t)$ becomes Pareto optimal arm after adding $\epsilon$ to all entries of its expected reward. Formally,*

$$\Delta_i(t) := \inf \{\epsilon \mid (\mu_i(t) + \epsilon) \nprec \mu_{i'}(t), \forall i' \in [K]\}.$$

*Then, the cumulative **Pareto regret** is defined as $\mathcal{PR}(T) := \sum_{t=1}^{T} \mathbb{E}[\Delta_{a(t)}(t)]$, where $\mathbb{E}[\Delta_{a(t)}(t)]$ represents the expected Pareto suboptimality gap of the arm pulled at round $t$.*

**Objective fairness**   Objective fairness can also be defined in the same way as in the fixed-arm case. For each round, the fairness index is defined with respect to the optimal arm for each objective, which may vary over time.

**Definition 13** (Objective fairness). *For each round $t \in [T]$, let $\mu_{i,m}(t)$ be the expected reward of arm $i$ for objective $m$, $a_m^*(t)$ be the arm that has the highest expected reward for objective $m$, and $\mu_m^*(t) := \mu_{a_m^*,m}(t)$. For all $\epsilon > 0$, we define **the objective fairness index** $\mathrm{OFI}_{\epsilon,T}$ of an algorithm as*

$$\mathrm{OFI}_{\epsilon,T} := \min_{m \in [M]} \left( \frac{1}{T} \mathbb{E} \left[ \sum_{t=1}^{T} \mathbb{1}\{\mu_m^*(t) - \mu_{a(t),m}(t) < \epsilon\} \right] \right).$$

*Then, we say that an algorithm satisfies **objective fairness** if for a given $\epsilon$, there exists a positive lower bound $L_\epsilon$ such that $\lim_{T \to \infty} \mathrm{OFI}_{\epsilon,T} \geq L_\epsilon$.*

## G.2   PARETO REGRET BOUND OF MOG WITH STOCHASTIC CONTEXTS

To analyze MOG under stochastic setup, the following assumption is essential to guarantee that the feature vectors in round $t$ are not influenced by previous rounds $s = 1, \ldots, t-1$.

**Assumption 5** (Independently distributed contexts). *The context sets $\chi(1), \ldots, \chi(T)$, drawn from unknown distribution $P_\chi(1), \ldots, P_\chi(T)$ respectively, are independently distributed across time.*

All of the greedy linear contextual bandit with stochastic contexts assumes the independence of context sets. It is important to note that feature vectors within the same round are allowed to be dependent, even under Assumption 5.

As in the regret bounds of MOG, MOG-R, and MOG-WR in fixed arm setting, the key to deriving the regret bound of MOG in the stochastic contextual setup is to establish the linear growth of the minimum eigenvalue of the Gram matrix.

**Lemma 16** (Increment of the minimum eigenvalue of the Gram matrix). *Suppose Assumptions 1, 2, 4, and 5 hold. If the OLS estimator satisfies $\|\hat{\theta}_m(s) - \theta_m^*\| \leq \alpha$, for all $m \in [M]$ and $s \geq T_0 + 1$, then the selected arms for a single cycle $s = t_0, t_0 + 1, \ldots, t_0 + M - 1$ ($t_0 > T_0$) by Algorithm 1 satisfies*

$$\lambda_{\min}\left( \sum_{s=t_0}^{t_0+M-1} \mathbb{E}[x(s)x(s)^\top | \mathcal{H}_{s-1}] \right) \geq \frac{\lambda q_\gamma M}{3},$$

*where $q_\gamma$ is defined in Definition 9.*

*Proof.* For $s \geq T_0 + 1$, let $m(s)$ be the target objective for iteration $s$ and $R(s)$ be the event that there exist $\gamma$-good arm for $\hat{\theta}_{m(s)}(s)$. Then,

$$\begin{aligned}
&\mathbb{E}[x(s)x(s)^\top | \mathcal{H}_{s-1}] \\
&\succeq \mathbb{E}[x(s)x(s)^\top | R(s), \mathcal{H}_{s-1}] \, \mathbb{P}(R(s)|\mathcal{H}_{s-1}) \\
&\succeq q_\gamma \mathbb{E}[x(s)x(s)^\top | R(s), \mathcal{H}_{s-1}] \\
&\succeq q_\gamma \mathbb{E}[x(s)|R(s), \mathcal{H}_{s-1}] \mathbb{E}[x(s)|R(s), \mathcal{H}_{s-1}]^\top.
\end{aligned}$$

Thus, we have $\sum_{s=t_0}^{t_0+M-1} \mathbb{E}[x(s)x(s)^\top | \mathcal{H}_{s-1}] \succeq q_\gamma \sum_{s=t_0}^{t_0+M-1} \mathbb{E}[x(s)|R(s), \mathcal{H}_{s-1}] \mathbb{E}[x(s)|R(s), \mathcal{H}_{s-1}]^\top$. Since $\mathbb{E}[x(s)|R(s), \mathcal{H}_{s-1}]$ is $\gamma$-good for $\hat{\theta}_{m(s)}(s)$ by Proposition 1 and 2, so we can apply Lemma 6 by

$$\lambda_{\min}\left( \sum_{s=t_0}^{t_0+M-1} \mathbb{E}[x(s)x(s)^\top | \mathcal{H}_{s-1}] \right) \geq q_\gamma \, \lambda_{\min}\left( \sum_{s=t_0}^{t_0+M-1} \mathbb{E}[x(s)|R(s), \mathcal{H}_{s-1}] \mathbb{E}[x(s)|R(s), \mathcal{H}_{s-1}]^\top \right)$$

$$\geq \frac{q_\gamma \lambda M}{3}.$$

$\square$

Then, the regret bound can then be derived by the same logic as in the proof of Theorems 1 and 3. The following corollary demonstrates that the `MOG` algorithm also possesses a $\widetilde{\mathcal{O}}(\sqrt{T})$-regret bound in the case of stochastic contexts.

**Corollary 6** (Pareto Regret Bound of `MOG` with Stochastic Contexts). *Suppose Assumptions 1, 2, 4, and 5 hold. If we run Algorithm 1 with $B = \min\left[\frac{\sigma}{\alpha}\sqrt{2dT\log(dT^2)}, \frac{4\sigma^2}{\alpha^2}\left(\frac{d}{2}\log\left(1+\frac{2T}{d}\right)+\log(T)\right)\right]$ where $\alpha = \sqrt{\frac{\lambda^2}{9}-\frac{\lambda^4}{324}}$ $\gamma - \left(1-\frac{\lambda^2}{18}\right)\sqrt{1-\gamma^2}$, the Pareto regret of Algorithm 1 is bounded by*

$$\mathcal{PR}(T) \leq \frac{48\sigma}{\lambda q_\gamma}\sqrt{2dT\log(dT)} + 2\left(2T_0 + 5M + \frac{30d}{\lambda q_\gamma}\right),$$

*where $q_\gamma$ in Definition 9.*

The corollary demonstrates that the cumulative Pareto regret bound of `MOG` is $\widetilde{\mathcal{O}}(\frac{\sqrt{dT}}{\lambda})$. Additionally, in a stochastic setup, $T_0$ can also be bounded at a scale of $\mathcal{O}(B)$ with high probability.

### G.3  OBJECTIVE FAIRNESS OF `MOG` WITH STOCHASTIC CONTEXTS

The objective fairness index can be bounded by combining the arguments from Theorems 2 and 4 with Lemma 16. The following corollary implies that Algorithm 1 satisfies objective fairness.

**Corollary 7** (Objective Fairness of `MOG` with Stochastic Contexts). *Suppose Assumptions 1, 2, 4, and 5 hold. Then, the objective fairness index of Algorithm 1 satisfies for all $m \in [M]$,*

$$\mathrm{OFI}_{\epsilon,T} \geq \left(\frac{T-T_\epsilon-M}{MT}\right)\left(1-\frac{3M}{T}-d(\frac{1}{dT})^{\frac{40\sigma^2 d}{\lambda q_\gamma \epsilon^2}}\right),$$

*where $T_\epsilon = \max(\lfloor\frac{1152\sigma^2 d\log(dT)}{\lambda^2 q_\gamma^2 \epsilon^2}\rfloor + T_0 + M, \ 2T_0 + 2M)$ in the same setting as Theorem 6.*

Notably, for any given $\epsilon > 0$, $\lim_{T\to\infty}\mathrm{OFI}_{\epsilon,T} = \frac{1}{M}$.

## H  RELAXATION OF THE BOUNDEDNESS ASSUMPTION

In this section, we explain how to release the boundedness assumption, Assumption 1. In conclusion, we can obtain results of the same scale as Theorems 1 and 2 for any arbitrary bound $\|x_i\|_2 \leq x_{\max}$ and $l \leq \theta_m^* \leq L$ for all $m \in [M]$. For clarity, we will separately discuss how to release the $l_2$ norm bounds on the feature vector and the objective parameters in Appendix H.1 and H.2, respectively. However, It is important to note that there is no issue in applying the same argument even when the bound on the feature vectors and the bound on the objective parameters are released simultaneously. We present how to release the boundedness assumption in fixed features setting, but the same reasoning can be applied to the case of stochastic contexts.

### H.1  RELEASING BOUND ON FEATURE VECTORS

We demonstrate how the minimum eigenvalue of the Gram matrix can increase linearly when the $l_2$ norm of the feature vectors is bounded by an arbitrary upper bound $x_{\max}$. Since the $\gamma$-goodness assumption is related to the scale of the feature, we modify the $\gamma$-goodness assumption correspondingly.

**Assumption 6** (Boundedness). *For all $i \in [K]$ and $m \in [M]$, $\|x_i\|_2 \leq x_{\max}$ and $\|\theta_m^*\|_2 = 1$.*

**Assumption 7** ($\gamma$-Goodness). *We assume $\{x_1, \ldots, x_K\}$ satisfies $\gamma$-goodness with $\gamma > \frac{x_{\max}}{\lambda}\sqrt{2\sqrt{1+\lambda^2}-2}$.*

The following lemma is the key to the release process.

**Lemma 17.** *Given Assumptions 6, assume the OLS estimator satisfies $\|\hat{\theta}_m(s) - \theta_m^*\| \leq \alpha$, for $m \in [M]$ and $s \geq T_0 + 1$. If $x \in \mathbb{B}_{x_{\max}}^d$ satisfies $x^\top\frac{\hat{\theta}_m(s)}{\|\hat{\theta}_m(s)\|} \geq \gamma$, then the distance between $\frac{x}{\gamma}$ and*

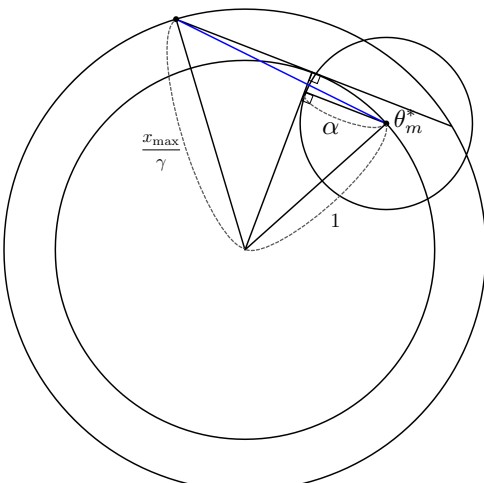

Figure 4: The interior of the circle with radius $\frac{x_{\max}}{\gamma}$ represents the region where $\frac{x}{\gamma}$ may exist in $\mathbb{R}^d$, while that of the smallest circle indicates the region where $\hat{\theta}_m(s)$ may exist. Then, the blue line illustrates the case when $\frac{x}{\gamma}$ is farthest from the $\theta_m^*$.

$\theta_m^*$ *is bounded by*

$$\left\| \theta_m^* - \frac{x}{\gamma} \right\|_2 \leq \sqrt{1 + (\frac{x_{\max}}{\gamma})^2 + 2\alpha\sqrt{(\frac{x_{\max}}{\gamma})^2 - 1} - 2\sqrt{1 - \alpha^2}}.$$

*Proof.* Consider the case when $\frac{x}{\gamma}$ is the farthest from $\theta_m^*$. As we easily can see from Figure 4,

$$\left\| \theta_m^* - \frac{x}{\gamma} \right\|_2^2 \leq \left( \alpha + \sqrt{(\frac{x_{\max}}{\gamma})^2 - 1} \right)^2 + (1 - \sqrt{1 - \alpha^2})^2$$

$$= 1 + (\frac{x_{\max}}{\gamma})^2 + 2\alpha\sqrt{(\frac{x_{\max}}{\gamma})^2 - 1} - 2\sqrt{1 - \alpha^2}.$$

$\square$

**Corollary 8.** *Suppose Assumptions 2, 6, and 7 hold. Assume the OLS estimator satisfies* $\|\hat{\theta}_m(s) - \theta_m^*\| \leq \alpha$, *for all* $m \in [M]$ *and* $s \geq T_0 + 1$. *If* $x_{r(1)}, \ldots, x_{r(M)} \in \mathbb{B}_{x_{\max}}^d$ *are* $\gamma$-good for $\hat{\theta}_1(s_1), \ldots, \hat{\theta}_M(s_M)$ *for some* $s_1, \ldots, s_M \geq T_0 + 1$, *respectively, then the following holds*

$$\lambda_{\min} \left( \sum_{m \in [M]} x_{r(m)} (x_{r(m)})^\top \right) \geq \left( \lambda\gamma^2 - 2x_{\max}\sqrt{\gamma^2 + x_{\max}^2 + 2\alpha\sqrt{x_{\max}^2 - \gamma^2} - 2\gamma^2\sqrt{1 - \alpha^2}} \right) M.$$

*Proof.* By Lemma 17, $\left\| \theta_m^* - \frac{x_{r(m)}}{\gamma} \right\|_2 \leq \sqrt{1 + (\frac{x_{\max}}{\gamma})^2 + 2\alpha\sqrt{(\frac{x_{\max}}{\gamma})^2 - 1} - 2\sqrt{1 - \alpha^2}}$ holds for all $m \in [M]$. Then,

$$\lambda_{\min} \left( \sum_{m \in [M]} x_{r(m)} (x_{r(m)})^\top \right) = \gamma^2 \lambda_{\min} \left( \sum_{m \in [M]} \frac{x_{r(m)}}{\gamma} \left( \frac{x_{r(m)}}{\gamma} \right)^\top \right)$$

$$\geq \gamma^2 \left[ \lambda_{\max} \left( \sum_{m \in [M]} \theta_m^* (\theta_m^*)^\top \right) - 2M \left( \frac{x_{\max}}{\gamma} \right) \left\| \theta_m^* - \frac{x_{r(m)}}{\gamma} \right\| \right]$$

$$\geq \lambda\gamma^2 - 2x_{\max}\sqrt{\gamma^2 + x_{\max}^2 + 2\alpha\sqrt{x_{\max}^2 - \gamma^2} - 2\gamma^2\sqrt{1 - \alpha^2}}.$$

$\square$

The above corollary means that even when Assumptions 1 and 3 are replaced by Assumptions 6 and 7, respectively, we can still obtain a regret bound that differs by at most a constant factor. Furthermore, using the same argument as before, we can also verify the objective fairness with replaced assumptions.

## H.2 RELEASING BOUND ON OBJECTIVE PARAMETERS

In this section, we present how to handle objective parameters with varying $l_2$ norm sizes. The $\gamma$-goodness assumption is related to the scale of the objectives either, the $\gamma$-goodness assumption is modified again correspondingly.

**Assumption 8** (Boundedness). *For all $i \in [K]$ and $m \in [M]$, $\|x_i\|_2 \leq 1$ and $l \leq \|\theta_m^*\|_2 \leq L$.*

**Assumption 9** ($\gamma$-Goodness). *We assume $\{x_1, \dots, x_K\}$ satisfies $\gamma$-goodness with $\gamma > 1 - \frac{\lambda^2}{8L^4}$.*

The following lemma is the key to the release process.

**Lemma 18.** *Given Assumptions 8, assume the OLS estimator satisfies $\|\hat{\theta}_m(s) - \theta_m^*\| \leq \alpha$, for $m \in [M]$ and $s \geq T_0 + 1$. If $x \in \mathbb{B}^d$ is $\gamma$-good for $\hat{\theta}_m(s)$, then the distance between $x$ and $\frac{\theta_m^*}{\|\theta_m^*\|_2}$ is bounded by*

$$\left\| \frac{\theta_m^*}{\|\theta_m^*\|_2} - x \right\|_2 \leq \sqrt{2 + \frac{2\alpha}{l}\sqrt{1 - \gamma^2} - 2\gamma\sqrt{1 - \frac{\alpha^2}{l^2}}}.$$

*Proof.* Consider the case when $x$ is the farthest from $\frac{\theta_m^*}{\|\theta_m^*\|_2}$. As we easily can see from Figure 5,

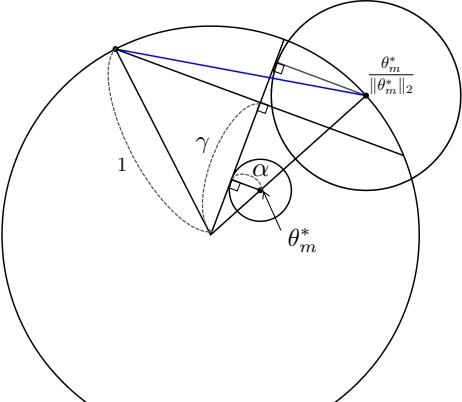

Figure 5: The larger circle represents the unit sphere in $\mathbb{R}^d$ while the interior of the smallest circle indicates the region where $\hat{\theta}_m(s)$ may exist. Then, the blue line illustrates the case when $x$ is farthest from the $\frac{\theta_m^*}{\|\theta_m^*\|_2}$.

we can obtain the following result from Lemma 5 by replacing $\alpha$ by $\frac{\alpha}{l}$.

$$\left\| \frac{\theta_m^*}{\|\theta_m^*\|_2} - x \right\|_2 \leq \sqrt{2\left(1 + \left(\frac{\alpha}{l}\right)\sqrt{1 - \gamma^2} - \gamma\sqrt{1 - \left(\frac{\alpha}{l}\right)^2}\right)}.$$

$\square$

**Corollary 9.** *Suppose Assumptions 2, 8, and 9 hold. Assume the OLS estimator satisfies* $\|\hat{\theta}_m(s) - \theta_m^*\| \leq \alpha$, *for all* $m \in [M]$ *and* $s \geq T_0 + 1$. *If* $x_{r(1)}, \ldots, x_{r(M)} \in \mathbb{B}^d$ *are* $\gamma$-good *for* $\hat{\theta}_1(s_1), \ldots, \hat{\theta}_M(s_M)$ *for some* $s_1, \ldots, s_M \geq T_0 + 1$, *respectively, then the following holds*

$$\lambda_{\min}\left(\sum_{m \in [M]} x_{r(m)}\big(x_{r(m)}\big)^\top\right) \geq \left(\frac{\lambda}{L^2} - 2\sqrt{2 + \frac{2\alpha}{l}\sqrt{1 - \gamma^2}} - 2\gamma\sqrt{1 - \frac{\alpha^2}{l^2}}\right) M.$$

The corollary can be derived from Lemma 18 and $\lambda_{\min}\left(\frac{1}{M}\sum_{m=1}^M \left(\frac{\theta_m^*}{\|\theta_m^*\|_2}\right)\left(\frac{\theta_m^*}{\|\theta_m^*\|_2}\right)^\top\right) \geq \frac{\lambda}{L^2}$.

Therefore, we can still obtain a regret bound that differs by at most a constant factor and the objective fairness criterion when Assumptions 1 and 3 are replaced by Assumptions 8 and 9, respectively.

# I   RELAXATION OF ASSUMPTION 2

Until now, we have conducted the analysis under the assumption that the feature vectors span $\mathbb{R}^d$. Although this assumption is not explicitly stated, it is implied by Lemma 6. In this section, we present a sufficient condition under which our proposed algorithms perform well when the feature vectors do not span $\mathbb{R}^d$ and explain how this leads to regret bounds and objective fairness.

**Intuition.** It is evident that any bandit algorithm cannot obtain information about the true objective parameters in the direction of $S_x^\perp$ while interacting with feature vectors $x_1, \ldots, x_K$. In other words, during the process of estimating the objective parameters, no estimator can converge to the true parameters in the direction of space $S_x^\perp$. Interestingly, from the perspective of regret and optimality, this poses no significant issue. This can be expressed mathematically as for any pair of arms $i, j \in [K]$ and $m \in [M]$,

$$x_i^\top \theta_m^* - x_j^\top \theta_m^* = x_i^\top \left(\pi_S(\theta_m^*)\right) - x_j^\top \left(\pi_S(\theta_m^*)\right).$$

The above equation explains why regret and optimality are determined solely by the projection vector of the objective parameters onto $S_x$.

---

**Algorithm 4** Multi-Objective Greedy algorithm (MOG)

---

**Require:** Total rounds $T$, Threshold $B$
   Initialize $V_0 \leftarrow 0 \times I_d$, and $\beta_1, \ldots, \beta_M \in \mathbb{R}^d$
   **for** $t = 1$ **to** $T$ **do**
      Select the target objective $m \leftarrow t \bmod M$ {If $m = 0$, then $m \leftarrow M$}
      **if** $\min_{\|\beta\|=1,\ \beta \in S_x}\left(\sum_{s=1}^{t-1} \langle \beta,\ x(s)\rangle^2\right) < B$ **then**
         Select action $a(t) \in \arg\max_{i \in [K]} x_i^\top \beta_m$
      **else**
         Update the OLS estimator $\hat{\theta}_m(t)$, arbitrary solution of $\left(\sum_{s=1}^{t-1} x(s)x(s)^\top\right)\theta = \sum_{s=1}^{t-1} x(s)y_{a(s),m}(s)$, for $m \in [M]$
         Select action $a(t) \in \arg\max_{i \in [K]} x_i^\top \hat{\theta}_m(t)$
      **end if**
      Observe $y(t) = \left(y_{a(t),1}(t), \ldots, y_{a(t),M}(t)\right)$
      Update $V_t \leftarrow V_{t-1} + x(t)x(t)^\top$
   **end for**

---

Algorithm 4 provides a general formulation of the MOG algorithm for use when the feature vectors do not span $\mathbb{R}^d$. In this case, it is impossible to satisfy $\lambda_{\min}(V_{t-1}) > B\ (> 0)$, which is the initial exploration criterion stated in Algorithm 1. Therefore, the initial exploration criterion should be modified. Instead of $\lambda_{\min}(V_{t-1}) > B$, we can use $\min_{\|\beta\|=1,\ \beta \in S_x}\left(\sum_{s=1}^{t-1} \langle \beta,\ x(s)\rangle^2\right) > B$. Additionally, under this condition, a unique least squares solution no longer exists. Therefore, for each round $t$, we use an arbitrary solution $\hat{\theta}_m(t)$ of the equation $\left(\sum_{s=1}^{t-1} x(s)x(s)^\top\right)\theta = \sum_{s=1}^{t-1} x(s)y_{a(s),m}(s)$. Notably, at least one solution exists after initial phase since $x(1), \ldots, x(t-1)$ span $S_x$. By extending Algorithm 1 in this way, we can conduct the same analysis as before.

The following are the revised versions of Assumptions 1, 2, and 3 when the feature vectors do not span $\mathbb{R}^d$.

**Assumption 10** (Boundedness). *For all $i \in [K]$ and $m \in [M]$, $\|x_i\|_2 \leq 1$ and $\|\pi_s(\theta_m^*)\|_2 = 1$.*

Once again, the above assumption is intended for a clear analysis. The analyses conducted in this section can be also extended to arbitrary bounds $\|x_i\|_2 \leq x_{\max}$ and $l \leq \pi_s(\theta_m^*) \leq L$ for all $m \in [M]$ by the same process in Appendix H.

**Assumption 11.** *We assume $\theta_1^*, \ldots, \theta_M^*$ span $S_x$.*

In the following analysis, we define $\lambda_1 := \min_{\|\beta\|=1, \, \beta \in S_x} \left( \frac{1}{M} \sum_{m=1}^M \langle \beta, \, \theta_m^* \rangle^2 \right)$. Then, given Assumption 11, $\lambda_1$ is always positive and clearly, $\lambda_1 = \min_{\|\beta\|=1, \, \beta \in S_x} \left( \frac{1}{M} \sum_{m=1}^M \langle \beta, \, \pi_S(\theta_m^*) \rangle^2 \right)$.

Next, we reconsider how to define $\gamma$-goodness. If the feature vectors do not span $\mathbb{R}^d$, it becomes important to determine whether $\gamma$-good arms exist near the direction of $\pi_S(\theta_m^*)$ rather than $\theta_m^*$. The following definition clarifies this concept.

**Definition 14** ($\gamma$-goodness). *For fixed $\gamma \in (0,1]$, we say that the set of feature vectors $\{x_1, \ldots, x_K\}$ satisfies $\gamma$-goodness condition when there exists $\alpha > 0$ that satisfies*

$$\text{for all } \beta \in \mathbb{B}_\alpha(\pi_S(\theta_1^*)) \cup \ldots \cup \mathbb{B}_\alpha(\pi_S(\theta_M^*)), \text{ there exists } i \in [K], \quad x_i^\top \frac{\beta}{\|\beta\|_2} \geq \gamma. \quad (2)$$

**Assumption 12** ($\gamma$-goodness). *We assume $\{x_1, \ldots, x_K\}$ satisfies $\gamma$-regular with $\gamma > 1 - \frac{\lambda_1^2}{18}$.*

Once again, in the following analysis, $\alpha$ denote the value that satisfies the goodness condition defined in Definition 14, in conjunction with $\gamma$ as specified in Assumption 12. Again, if $\alpha$ is greater than $\psi(\lambda_1, \gamma) := \sqrt{\frac{\lambda_1^2}{9} - \frac{\lambda_1^4}{324}} \, \gamma - \left( 1 - \frac{\lambda_1^2}{18} \right) \sqrt{1 - \gamma^2}$, then we replace the value of $\alpha$ with $\psi(\lambda_1, \gamma)$.

The only question is how to construct an $l_2$ bound on $\pi_S(\hat{\theta}_m(s)) - \pi_S(\theta_m^*)$ without utilizing the minimum eigenvalue of the Gram matrix, which is zero when $S_x \subsetneq \mathbb{R}^d$. The key idea is that we can use $\min_{\|\beta\|=1, \, \beta \in S_x} \left( \sum_{s=1}^{t-1} \langle \beta, \, x(s) \rangle^2 \right)$ to fulfill the role previously played by the minimum eigenvalue. We present 2 Lemmas, Lemma 19 and Lemma 20, to explain the idea. First, The following demonstrates the linear growth of $\min_{\|\beta\|=1, \, \beta \in S_x} \left( \sum_{s=1}^{t-1} \langle \beta, \, x(s) \rangle^2 \right)$.

**Lemma 19.** *Suppose Assumptions 10, 11, and 12 hold. Assume a least square solution $\hat{\theta}_m(s)$ satisfies $\|\pi_S(\hat{\theta}_m(s)) - \pi_S(\theta_m^*)\| \leq \alpha$, for all $m \in [M]$ and $s \geq T_0 + 1$. If $x_{r(1)}, \ldots, x_{r(M)} \in \mathbb{B}^d$ are $\gamma$-good for $\pi_s(\hat{\theta}_1(s_1)), \ldots, \pi_s(\hat{\theta}_M(s_M))$, for some $s_1, \ldots, s_M \geq T_0 + 1$, respectively, then*

$$\min_{\|\beta\|=1, \, \beta \in S_x} \left( \sum_{m \in [M]} \langle \beta, \, x_{r(m)} \rangle^2 \right) \geq \frac{\lambda_1}{3} M.$$

*Proof.* Since the greedy selection of $\hat{\theta}_m(s)$ is equal to that of $\pi_S(\hat{\theta}_m(s))$, for the same reason as Lemma 5, we can get $\|x_{r(m)} - \pi_S(\theta_m^*)\|_2 \leq \sqrt{2 \left( 1 + \alpha\sqrt{1-\gamma^2} - \gamma\sqrt{1-\alpha^2} \right)}$.

Then, for any unit vector $\beta \in S_x$,

$$\beta^\top \left( \sum_{m=1}^M x_{r(m)} x_{r(m)}^\top \right) \beta$$

$$= \sum_{m=1}^M \{ \left\langle \beta, \, \pi_S(\theta_{m(s)}^*) \right\rangle^2 + \left\langle \beta, \, x_{r(m)} - \pi_S(\theta_{m(s)}^*) \right\rangle^2 + 2 \left\langle \beta, \, \pi_S(\theta_{m(s)}^*) \right\rangle \left\langle \beta, \, x_{r(m)} - \pi_S(\theta_{m(s)}^*) \right\rangle \}$$

$$\geq M\lambda_1 - 2\sqrt{2 \left( 1 + \alpha\sqrt{1-\gamma^2} - \gamma\sqrt{1-\alpha^2} \right)} M.$$

$\square$

The next lemma shows how to derive $l_2$ bound on $\pi_S\big(\hat{\theta}_m(s)\big) - \pi_S\big(\theta_m^*\big)$ with $\min_{\|\beta\|=1,\ \beta \in S_x} \Big( \sum_{s=1}^{t-1} \langle \beta,\ x(s) \rangle^2 \Big)$.

**Lemma 20.** *For all* $m \in [M]$ *and* $s \in [T]$, *any least square solution* $\hat{\theta}_m(t)$ *of* $\big( \sum_{s=1}^{t-1} x(s)x(s)^\top \big) \theta = \sum_{s=1}^{t-1} x(s)y_{a(s),m}(s)$ *satisfies*

$$\left\| \pi_S\big(\hat{\theta}_m(s)\big) - \pi_S\big(\theta_m^*\big) \right\|_2 \leq \frac{\| \sum_{s=1}^{t-1} x(s)\eta_{a(s),m}(s) \|_2}{\min_{\|\beta\|=1,\ \beta \in S_x} \Big( \sum_{s=1}^{t-1} \langle \beta,\ x(s) \rangle^2 \Big)}.$$

*Proof.* From the definition of $\hat{\theta}_m(t)$, we have

$$\Big( \sum_{s=1}^{t-1} x(s)x(s)^\top \Big) (\hat{\theta}_m(t) - \theta_m^*) = \sum_{s=1}^{t-1} x(s)\eta_{a(s),m}(s).$$

Since the row space of $\big( \sum_{s=1}^{t-1} x(s)x(s)^\top \big)$ is in $S$,

$$\left\| \sum_{s=1}^{t-1} x(s)\eta_{a(s),m}(s) \right\|_2 = \left\| \Big( \sum_{s=1}^{t-1} x(s)x(s)^\top \Big) \ \Big( \pi_S\big(\hat{\theta}_m(s)\big) - \pi_S\big(\theta_m^*\big) \Big) \right\|_2$$

$$\geq \min_{\|\beta\|=1,\ \beta \in S_x} \Big( \beta^\top \big( \sum_{s=1}^{t-1} x(s)x(s)^\top \big) \beta \Big) \left\| \pi_S\big(\hat{\theta}_m(s)\big) - \pi_S\big(\theta_m^*\big) \right\|_2.$$

The last inequality holds by Lemma 28. $\qquad\square$

With above two lemmas, we can obtain the same regret bound and objective fairness as in Theorem 1 and 2.

It is important to note that the same discussion applies to `MOG-R` and `MOG-WR`, indicating that our proposed near-greedy algorithms can perform well even when the feature vectors do not span $\mathbb{R}^d$, when there exist good arms for multiple objectives.

## J LOWER BOUND

**Theorem 5.** *Suppose Assumptions 1, 2, and 3 hold, and* $d \geq 2$ *and* $K \geq d^2$. *For any algorithm choosing action* $a(t)$ *at round* $t$, *there exists a worst-case problem instance such that the Pareto regret of the algorithm is lower bounded as*

$$\sup_{(\theta_1^*,\dots,\theta_M^*)} \mathcal{PR}(T) = \Omega(\sqrt{dT}).$$

**Discussion of Theorem 5.** The above theorem shows that the regret bound for our algorithm in Theorem 1 is optimal in terms of $d$ and $T$. This bound matches the lower bound of Chu et al. (2011a) in the single-objective setting. However, in their work, the $d$ term in the lower bound is obtained by partitioning the time horizon and carefully designing the features within each partition. As a result, their analysis requires the condition $T \geq d^2$, and their approach is not applicable in our fixed feature setting. Instead, we obtain the $d$ term by partitioning the set of $K$ arms, which leads to the requirement that $K \geq d^2$ in our analysis.

For our convenience, we define the following augmented parameter set, which will be used throughout the remainder of this section.

**Definition 15.** *The augmented parameter set* $\Theta$ *is a set of combinations of objective parameters, i.e.,* $\Theta := \big\{ \big(\theta_1^{(1)}, \theta_2^{(1)}, \dots, \theta_M^{(1)}\big), \big(\theta_1^{(2)}, \theta_2^{(2)}, \dots, \theta_M^{(2)}\big), \dots, \big(\theta_1^{(d)}, \theta_2^{(d)}, \dots, \theta_M^{(d)}\big) \big\}$, *where each* $\theta_m^{(j)} \in \mathbb{R}^d$ *for all* $j \in [d]$ *and for all* $m \in [M]$. *The set of the first objective parameters in each of instances in* $\Theta$ *is defined as* $\Theta_1 = \big\{ \theta_1^{(1)}, \theta_1^{(2)}, \dots, \theta_1^{(d)} \big\}$.

Note that $\Theta$ represents $d$ separate multi-objective problem instances, while $\Theta_1$ represents $d$ separate single-objective (objective 1) problem instances.

*Proof sketch.* We prove the theorem by constructing the augmented parameter set $\Theta$ and defining the feature set so that the feature vectors are aligned with the direction of objective parameters and has maximum length, thereby ensuring that Assumption 3 is satisfied. Then, we bound the Bayes Pareto regret $\mathbb{E}_{\theta^* \sim \text{UNIFORM}(\Theta)}[\mathcal{PR}(T)]$ for any action sequence $a(t), t \in [T]$. For this, we convert our problem to single objective problem, and then use Lemma 26. The proof of Theorem 5 is presented in Section J.2, and its supporting lemmas are presented in Section J.1

### J.1  TECHNICAL LEMMAS FOR THEOREM 5

We first construct the set of problem instances that can be converted to a single objective problem.

**Lemma 21.** *Suppose Assumptions 1, 2, and 3 hold. For all $d \geq 2$ and $K \geq d^2$, there exist an augmented parameter set $\Theta$ and a set of feature vectors such that for any action sequence $a(t) \in [K]$ for $t \in [T]$, there exists another action sequence $a'(t) \in [d]$ for $t \in [T]$ that satisfies*

$$\mathbb{E}_{\theta^* \sim \text{UNIFORM}(\Theta)} \left[ \sum_{t=1}^{T} \Delta_{a(t)} \right] \geq \mathbb{E}_{\theta_1^* \sim \text{UNIFORM}(\Theta_1)} \left[ \sum_{t=1}^{T} (\max_{i \in [d]} x_i^\top \theta_1^* - x_{a'(t)}^\top \theta_1^*) \right]. \quad (3)$$

*Proof.* It is enough to show when $M = d$ and $K = d^2$, because if $M > d$ (Assumption 2 guarantees $M \geq d$) or $K > d^2$, we can make the same argument by setting $\theta_m^{(j)} = \theta_d^{(j)}$ for all $d \leq m \leq M$ and $j \in [d]$ or $x_i = x_d$ for $d \leq i \leq K$. We first divide the cases by when $d = 2$ and $d \geq 3$.

**Case 1**. $d = M = 2$ and $K = 4$

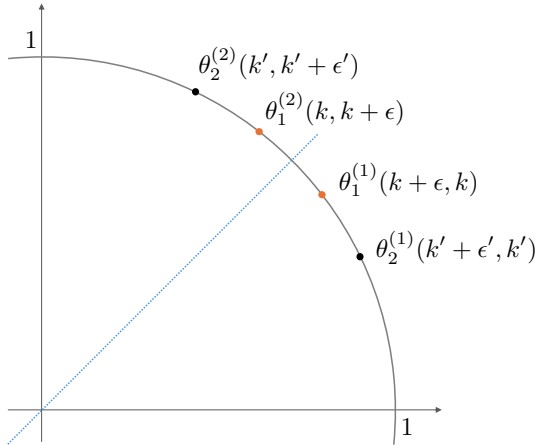

Figure 6: Problem space $\Theta$ construction when $d = 2$.

Fix $0 < \epsilon < 1$ and let $\theta_1^{(1)} = (k + \epsilon, k)$, $\theta_1^{(2)} = (k, k + \epsilon)$, $\theta_2^{(1)} = (k' + \epsilon', k')$, $\theta_2^{(2)} = (k', k' + \epsilon')$, where $\epsilon < \epsilon' < 1$ and $2k^2 + 2k\epsilon + \epsilon^2 = 2(k')^2 + 2(k')\epsilon + \epsilon^2 = 1$. Define the feature vectors $x_1 = \theta_1^{(1)}$, $x_2 = \theta_1^{(2)}$, $x_3 = \theta_2^{(1)}$, and $x_4 = \theta_2^{(2)}$. Then all feature vectors and objective parameters have $l_2$ norm 1, satisfying Assumption 1. Also, each $(\theta_1^{(1)}, \theta_2^{(1)})$ and $(\theta_1^{(2)}, \theta_2^{(2)})$ satisfies Assumption 2, and since $x_1, x_2, x_3, x_4$ are $\gamma$-good arms for $\theta_1^{(1)}, \theta_1^{(2)}, \theta_2^{(1)}, \theta_2^{(2)}$ for any $\gamma \leq 1$, the feature set satisfies Assumption 3. Now, we show that the good arms for objective 1 ($x_1$ and $x_2$) are always the better choice than other arms ($x_3$ and $x_4$) in both problem instances from the perspective of Pareto optimality.

If $(\theta_1^*, \theta_2^*) = (\theta_1^{(1)}, \theta_2^{(1)})$,

$$\mathbb{E}[\Delta_1] = 1 - \langle \theta_1^{(1)}, \theta_1^{(1)} \rangle \ (= 0)$$

$$\mathbb{E}[\Delta_2] = 1 - \langle \theta_1^{(2)}, \theta_1^{(1)} \rangle \ (= \epsilon^2)$$

$$\mathbb{E}[\Delta_3] = 0 = 1 - \langle \theta_1^{(1)}, \theta_1^{(1)} \rangle$$

$$\mathbb{E}[\Delta_4] = 1 - \langle \theta_2^{(2)}, \theta_1^{(1)} \rangle > \epsilon^2 = 1 - \langle \theta_1^{(2)}, \theta_1^{(1)} \rangle.$$

Otherwise, if $(\theta_1^*, \theta_2^*) = (\theta_1^{(2)}, \theta_2^{(2)})$,

$$\mathbb{E}[\Delta_1] = 1 - \langle \theta_1^{(1)}, \theta_1^{(2)} \rangle \ (= \epsilon^2)$$

$$\mathbb{E}[\Delta_2] = 1 - \langle \theta_1^{(2)}, \theta_1^{(2)} \rangle \ (= 0)$$

$$\mathbb{E}[\Delta_3] = 1 - \langle \theta_2^{(1)}, \theta_1^{(2)} \rangle > \epsilon^2 = 1 - \langle \theta_1^{(1)}, \theta_1^{(2)} \rangle$$

$$\mathbb{E}[\Delta_4] = 0 = 1 - \langle \theta_1^{(2)}, \theta_1^{(2)} \rangle.$$

Therefore, if we define $a'(t) = \begin{cases} 1 & \text{if } a(t) = 1 \text{ or } 3 \\ 2 & \text{if } a(t) = 2 \text{ or } 4 \end{cases}$, the statement of lemma is satisfied.

**Case 2.** $d = M \geq 3$ and $K = d^2$

Fix $0 < \epsilon < 1$ and define $\Theta$ and feature vectors $x_i$, $i \in [d^2]$ as

$$x_1 = \theta_1^{(1)} = (k + \epsilon, k, \dots, k),$$

$$x_2 = \theta_1^{(2)} = (k, k + \epsilon, k, \dots, k),$$

$$\dots$$

$$x_d = \theta_1^{(d)} = (k, \dots, k, k + \epsilon),$$

$$x_{d+1} = \theta_2^{(1)} = (k' + 2\epsilon, k' - \epsilon, k' \dots, k'),$$

$$x_{d+2} = \theta_2^{(2)} = (k', k' + 2\epsilon, k' - \epsilon, k' \dots, k'),$$

$$\dots$$

$$x_{2d} = \theta_2^{(d)} = (k' - \epsilon, k', \dots, k', k' + 2\epsilon)$$

$$x_{2d+1} = \theta_3^{(1)} = (k' + 2\epsilon, k', k' - \epsilon, k' \dots, k'),$$

$$x_{2d+2} = \theta_3^{(2)} = (k', k' + 2\epsilon, k', k' - \epsilon \dots, k'),$$

$$\dots$$

$$x_{d^2} = \theta_d^{(d)} = (k', k', \dots, k' - \epsilon, k' + 2\epsilon),$$

$$\text{where } dk^2 + 2k\epsilon + \epsilon^2 = d(k')^2 + 2(k')\epsilon + 5\epsilon^2 = 1.$$

It is obvious that $k' < k$ and $\|x_i\| = 1$ for all $i \in [d^2]$. Similar to the simple case when $d = 2$, for $j \in [d]$, each $(\theta_1^{(j)}, \dots, \theta_d^{(j)})$ satisfies Assumption 2, and the feature set satisfies Assumption 3.

To prove the lemma, similar to the simple $d = 2$ case, we will show that $\theta_1^{(j)}$ is always better than $\theta_m^{(j)}$ ($m \neq 1$) for all $j \in [d]$. For any feature vector $x_i$, we denote by $\Delta(x_i)$ the sub-optimality gap of the feature vector, i.e. $\Delta(x_i) := \Delta_i$. Then, it is enough to show that for any $m, j \in [d]$ and $\theta^* \in \Theta$, $\mathbb{E}_{\theta^*}[\Delta(x_{(m-1)d+j})] = \mathbb{E}_{\theta^*}[\Delta(\theta_m^{(j)})] \geq \max_{j' \in [d]} (\theta_1^{(j')})^\top \theta_1^* - (\theta_1^{(j)})^\top \theta_1^*$ holds.

Let $\theta^*$ be the objective parameters for $(j_*)$-th instance, i.e. $\theta^* = (\theta_1^{(j_*)}, \theta_2^{(j_*)}, \dots, \theta_d^{(j_*)}) \in \Theta$. If $j_* = j$, then $\mathbb{E}_{\theta^*}[\Delta(\theta_m^{(j)})] = 0 = 1 - (\theta_1^{(j)})^\top \theta_1^{(j)}$.

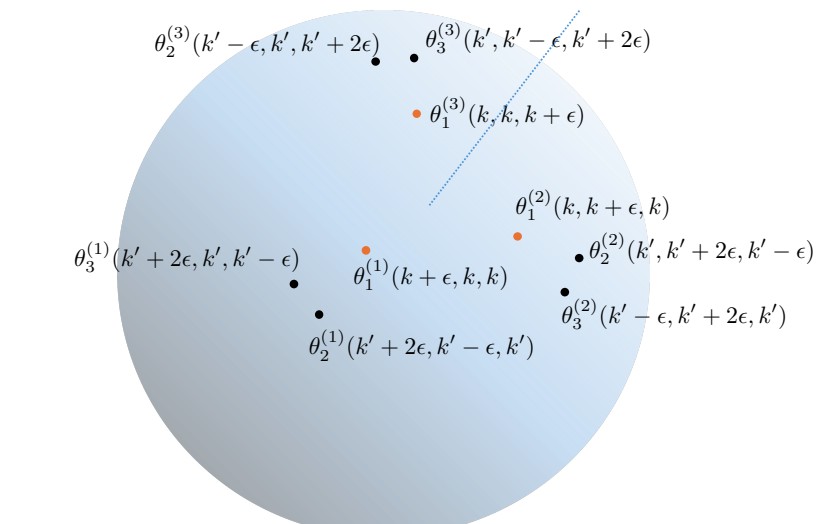

Figure 7: Problem space $\Theta$ construction when $d = 3$. The blue line represents the direction of $(1, 1, 1)$, and the sphere has the radius of 1.

Suppose $j_* \neq j$. For all $m' \in [d]$, since

$$\langle \theta_1^{(j)}, \theta_1^{(j_*)} \rangle = dk^2 + 2k\epsilon = 1 - \epsilon^2,$$

$$\langle \theta_1^{(j)}, \theta_{m'}^{(j_*)} \rangle \leq (k+\epsilon)(k') + k(k'+2\epsilon) + k(k'-\epsilon) + (d-3)kk'$$

$$= dkk' + k\epsilon + k'\epsilon$$

$$< \frac{d}{2}k^2 + \frac{d}{2}(k')^2 + k\epsilon + k'\epsilon$$

$$= \left(\frac{1}{2} - \frac{\epsilon^2}{2}\right) + \left(\frac{1}{2} - \frac{5\epsilon^2}{2}\right)$$

$$= 1 - 3\epsilon^2,$$

it holds that $\mathbb{E}_{\theta^*}[\Delta(\theta_1^{(j)})] = \epsilon^2 = 1 - \left(\theta_1^{(j)}\right)^\top \theta_1^*$.

For $m \neq 1$ and $m' \neq 1$, we have

$$\langle \theta_m^{(j)}, \theta_1^{(j_*)} \rangle \leq \max_{m' \neq 1} \langle \theta_1^{(j)}, \theta_{m'}^{(j_*)} \rangle < 1 - 3\epsilon^2,$$

$$\langle \theta_m^{(j)}, \theta_{m'}^{(j_*)} \rangle \leq 2(k')(k'+2\epsilon) + (k'-\epsilon)^2 + (d-3)(k')^2$$

$$= d(k')^2 + 2k'\epsilon + \epsilon^2$$

$$= 1 - 4\epsilon^2,$$

and hence $\mathbb{E}_{\theta^*}[\Delta(x_{(m-1)d+j})] = \mathbb{E}_{\theta^*}[\Delta(\theta_m^{(j)})] > 3\epsilon^2 > \epsilon^2 = 1 - \left(\theta_1^{(j)}\right)^\top \theta_1^*$. Therefore, if we define $a'(t) = a(t) \bmod m$ (if $a(t)/m \in \mathbb{N}$, then $a'(t) = m$), then the lemma holds. $\qquad \square$

**Lemma 22.** *Suppose Assumptions 1, 2, and 3 hold. For all $0 < \epsilon < 1$, $d \geq 2$, $K \geq d^2$, and any action sequence $a(t)$ for $t \in [T]$, there exists a augmented parameter set $\Theta$ and a set of features satisfying Equation (3), where for all $j \in [d]$, the expected reward of arm $j$ for objective 1 is equal to 1 in $j$-th problem instance and is $1 - \epsilon^2$ in other instances $j' \in [d] - \{j\}$.*

*Proof.* The parameter set $\Theta$ and the feature set $\{x_1 = \theta_1^{(1)}, x_2 = \theta_1^{(2)}, \ldots, x_{d^2} = \theta_1^{(d)}(= x_{d^2+1} = \ldots = x_K)\}$ constructed in the proof of Lemma 21 satisfy the properties required in the latter part of this lemma. For each $j \in [d]$, the feature vector of arm $j$ is given by $\theta_1^{(j)}$, so in the $j$-th instance,

the expected reward for objective 1 is $\langle \theta_1^{(j)}, \theta_1^{(j)} \rangle = 1$. For any other instance $j' \in [d] \setminus j$, we have $\langle \theta_1^{(j)}, \theta_1^{(j')} \rangle = 1 - \epsilon^2$. $\qquad\square$

The above lemma reduces the problem of bounding multi-objective regret to that of deriving a lower bound for the single-objective case. In particular, for each of $d$ instances, one arm among the $d$ arms has a single-objective expected reward larger than the others by $\epsilon^2$, which makes it possible to apply Lemma 26.

## J.2 PROOF OF THEOREM 5

*Proof.* By Lemma 22, it is enough to bound the single objective regret $\mathbb{E}_{\theta_1^* \sim \text{UNIFORM}(\Theta_1)}[\sum_{t=1}^{T}(\max_{i\in[d]} x_i^\top \theta_1^* - x_{a'(t)}^\top \theta_1^*)]$, where for each $\theta_1^{(j)} \in \Theta_1$, the expected reward of the arm $j$ is equal to $1$, while the other arms $j' \in [d] - \{j\}$ have the expected reward $1 - \epsilon^2$. If we set $\epsilon = \sqrt{1 - \frac{1}{1 + \frac{1}{2}\sqrt{\frac{d}{T}}}}$, then the expected reward of arm $j$ is $\frac{1}{1 + \frac{1}{2}\sqrt{\frac{d}{T}}}$ for $j'(\neq j)$-th instances. Scaling by $\frac{1}{2} + \frac{1}{4}\sqrt{\frac{d}{T}} > \frac{1}{2}$, we have that the expected reward of arm $j \in [d]$ is $\frac{1}{2} + \frac{1}{4}\sqrt{\frac{d}{T}}$ for $j$-th instance, while it is $\frac{1}{2}$ for other instances. Applying Lemma 26, we have

$$\mathbb{E}_{\theta_1^* \sim \text{UNIFORM}(\Theta_1)}[\sum_{t=1}^{T}(\max_{i\in[d]} x_i^\top \theta_1^* - x_{a'(t)}^\top \theta_1^*)] = \Omega(\sqrt{dT}).$$

Therefore, by Lemma 22,

$$\sup_{(\theta_1^*,\dots,\theta_M^*)} \left[\sum_{t=1}^{T} \mathbb{E}[\Delta_{a(t)}]\right] \geq \mathbb{E}_{\theta^* \sim \text{UNIFORM}(\Theta)} \left[\sum_{t=1}^{T} \Delta_{a(t)}\right]$$

$$\geq \mathbb{E}_{\theta_1^* \sim \text{UNIFORM}(\Theta_1)} \left[\sum_{t=1}^{T}(\max_{i\in[d]} x_i^\top \theta_1^* - x_{a'(t)}^\top \theta_1^*)\right]$$

$$= \Omega(\sqrt{dT}).$$

$\qquad\square$

# K EXPERIMENT

In this section, we present the experimental settings and results for our proposed algorithm. In summary, our algorithm achieves excellent empirical performance and exhibits stability across different parameter settings. Detailed descriptions of the experimental setup can be found in Section K.1.

We evaluated the performance of each algorithm in both cases of fixed arms and stochastic arms. When playing with stochastic arms, only contextual algorithms are compared. We evaluate the empirical performance of multi-objective bandit algorithms from three perspectives: cumulative Pareto regret, Pareto front approximation, and objective fairness. The results are presented in Section K.2

Additionally, we conducted experiments to examine how the performance of our proposed algorithms varies with different parameter settings. Specifically, we altered the parameter $B$ and the initial objective parameters $\beta_1, \dots, \beta_M$ of the MOG algorithm and measured the cumulative Pareto regret and the objective fairness index. The results are presented in Section K.3

To indirectly evaluate whether our algorithm performs well in real-world scenarios, we conducted a bandit experiment based on offline real-world data. A detailed explanation and the corresponding results are provided in Section K.4.

## K.1 SETTINGS

We validate the empirical performance of MOG, MOG-R, and MOG-WR in a linear bandit setting, comparing them with other multi-objective algorithms. Specifically, we experiment with a linear

bandit where $y_m(t) = \mathcal{N}(x_i^T \theta_m^*, 0.1^2)$ for all $i \in [K]$ and $m \in [M]$. For each problem instance, $M$ objective parameters are sampled uniformly at random from the positive part of $\mathbb{S}^{d-1}$. Then, $K$ feature vectors ($K > 2M$) are generated by drawing samples from $\mathbb{B}^d$. In the fixed arms setting, the first $M$ feature vectors are sampled from a multivariate normal distribution with the true objective parameter as the mean and a covariance matrix of $0.1I_d$. These vectors are then scaled to ensure their magnitudes lie within the range $(3/4, 1)$. The remaining $K - M$ feature vectors are sampled uniformly at random from $\mathbb{B}^d$, with $M$ of these scaled to have magnitudes greater than $3/4$ and the rest scaled to have magnitudes less than $3/4$. Limiting the magnitudes of the feature vectors ensures that excessively large Pareto fronts, which could lead to meaningless results, are avoided. For the varying arms setting, contexts are drawn uniformly from $\mathbb{B}^d$. The results are averaged over 10 independent problem instances for each $(d, K, M)$ combination, with each problem instance being repeated 10 times to compute the final statistics (repeated 5 times for problem instances with $(d, K, M) = (20, 400, 20)$).

We conduct experiments on our proposed near-greedy algorithms and the three baselines, `ParetoUCB` (Drugan & Nowe, 2013), `MOGLM-UCB` (Lu et al., 2019), and `PFIwR` (Kim et al., 2023) with tuned parameters for confidence width (The algorithms proposed by Cheng et al. (2024) are excluded from the experiments as they are specifically designed for problems with hierarchical objective structures.) The experiments are run on Xeon(R) Gold 6226R CPU @ 2.90GHz (16 cores). When tuning existing algorithms, we selected the parameter settings that yielded the best regret performance within the range specified in their respective papers. For `PFIwR`, we set $\delta = 0.1$ and $\epsilon = 0.18$. For `MOGLM-UCB`, the confidence width is defined as $\gamma_t = c \log \frac{\det(Z_t)}{\det(Z_1)}$, where $Z_t = I_d + \frac{1}{2} \sum_{s=1}^{t} x(s)x(s)^\top$, with the tuned parameter $c = 0.1$. Additionally, we use $B = 0.01$ for our proposed algorithms, `MOG`, `MOG-R`, and `MOG-WR`. In terms of random variables in the `MOG-R` algorithm, we use uniform distribution $(\frac{1}{M}, \ldots, \frac{1}{M})$ for choosing the target objective. For the `MOG-WR` algorithm, we use Dirichlet$(1, \ldots, 1)$ for generating the weight vectors.

### K.2 MULTI-OBJECTIVE BANDIT ALGORITHM COMPARISON

#### K.2.1 CUMULATIVE PARETO REGRET

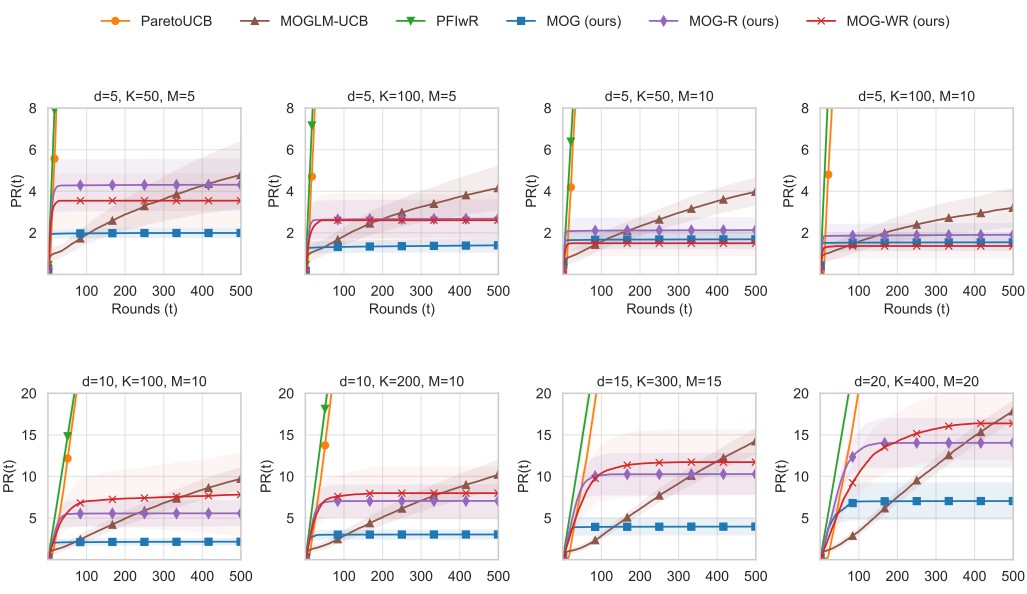

Figure 8: Cumulative Pareto regret of multi-objective bandit algorithms with fixed arms across various $(d, K, M)$ combinations. The shaded areas represent $\pm$ half the standard deviation for each algorithm.

The following summarizes the empirical results illustrated in Figure 8 and Figure 9, which plot the cumulative regret of algorithms for the fixed context case and stochastic context case, respec-

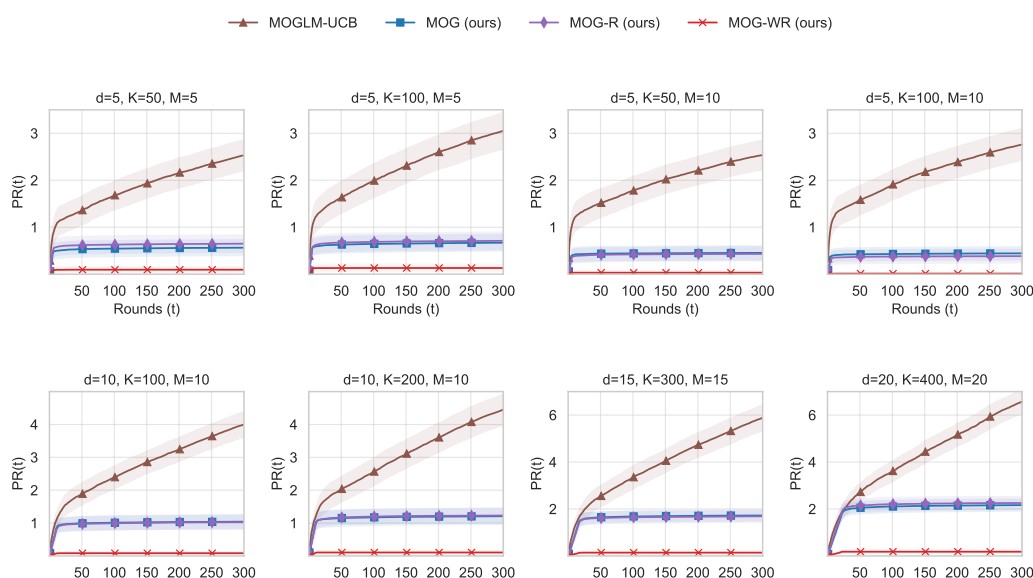

Figure 9: Cumulative Pareto regret of multi-objective bandit algorithms with stochastic contexts across various $(d, K, M)$ combinations. The shaded areas represent $\pm$ half the standard deviation for each algorithm.

tively. As observed in these figures, our simple near-greedy algorithms, `MOG`, `MOG-R`, and `MOG-WR`, demonstrate superior empirical performance compared to existing algorithms. In both fixed and stochastic experimental setups, our proposed algorithms exhibit almost no regret after the exploration phase, whereas other algorithms maintain a sublinear regret trend due to additional exploration terms even after the initial rounds. Specifically, `ParetoUCB` and `PFIwR` rely on conservative equations within the algorithm, resulting in relatively weak empirical performance. `MOGLM-UCB`, by using a tunable confidence width, achieved better empirical performance than the previous two algorithms but still lagged behind our proposed near-greedy algorithms. These results support our claim that in multi-objective settings, where multiple objectives generate many good arms, a brief exploration during the initial rounds is sufficient, after which exploitation alone can effectively address the problem.

Among proposed algorithms, in the fixed feature setup, `MOG` consistently achieved the best performance in most combinations of $(d, K, M)$. This is likely due to its deterministic selection of diverse arms, allowing it to efficiently and reliably complete the initial rounds. Among the randomized algorithms, `MOG-WR` was better in experiments with relatively small $d$, whereas `MOG-R` outperformed in larger experimental setups.

In the stochastic context setup, our near-greedy algorithms demonstrated exceptionally strong performance. This aligns with findings from single-objective studies, which have shown similar results, and extends naturally to the multi-objective setting. However, a surprising observation is that `MOG-WR` performed remarkably well, achieving near-zero regret in most scenarios. Notably, in the fixed-arm case, arms were selected to ensure that each objective direction contained some good arms. In contrast, in the stochastic setup, arms were drawn independently in each round from a unit ball uniform distribution without such constraints. Under this setup, our experiments revealed that `MOG` and `MOG-R` no longer had a performance advantage over `MOG-WR`. Additionally, `MOG` showed very little difference between its deterministic and randomized versions in stochastic settings.

This remarkable performance of `MOG-WR` is likely due to its greedy selection of arms in intermediate directions of the objectives. In multi-objective bandits, let us define the *objective region* as the region formed by the weighted sums of all true objective vectors. Under this definition, any optimal arm corresponding to a direction within the objective region belongs to the Pareto front. If some prior knowledge about each objective is available, the probability that the intermediate directions of the initial objective parameters fall within the objective region is higher than that of the initial objective

directions themselves. Specifically, in this experiment, the true objective parameters were all drawn from $\mathbb{R}_+^d$, and we used the standard basis vectors of $\mathbb{R}^d$ (along with additional vectors if necessary) as the initial objective parameters. In this case, the probability that intermediate directions between these standard basis vectors belong to the objective region is trivially higher than that of each $e_i$ direction. This observation provides a key explanation for why `MOG-WR` experiences almost no regret during the initial phase. Moreover, in real-world scenarios, prior knowledge about the true objective parameters is often available, which can further enhance the performance of `MOG-WR`.

### K.2.2 PARETO FRONT APPROXIMATION

As mentioned earlier, our proposed algorithms do not compute the empirical Pareto front at every round but can approximate the Pareto front when necessary. In this section, we empirically demonstrate how effectively `MOG`, `MOG-R`, and `MOG-WR` can approximate the Pareto front. Since our algorithms are near-greedy and do not include additional exploration terms after the initial rounds, we use $\hat{\theta}_m(t)$ as described in Lemma 1 to estimate the empirical Pareto front. We compare the estimated empirical Pareto front from our algorithms with those used by existing algorithms to evaluate how accurately they identify the true Pareto front. As a comparison metric, we use accuracy, defined as the proportion of arms correctly identified as belonging to the true Pareto front.

Figure 10 demonstrates that our algorithms effectively identify the Pareto front. Notably, when $d, M \leq 10$, `MOG` and `MOG-R` quickly identified the Pareto front, achieving an accuracy exceeding 0.98 within the first 100 rounds on average. The deterministic version `MOG` achieves the fastest, most accurate, and most stable Pareto front approximation across all experimental settings. Specifically, even in experiments with high dimensionality, a large number of arms, and multiple objectives, `MOG` estimates the Pareto front with accuracy exceeding 0.95 within the first 100 rounds. Next, `MOG-R` performed well in most cases, except for larger parameter experiments where $d, M \geq 15$ and $K \geq 300$. Similarly, `MOGLM-UCB` consistently showed strong performance across all settings.

In contrast, `MOG-WR`, while outperforming `ParetoUCB` and `PFIwR` after 300 rounds, exhibited inferior Pareto front approximation performance compared to `MOG` and `MOG-R`. This is an expected result, as efficient Pareto front approximation requires $\hat{\theta}_m(t)$ to converge quickly to $\theta_m^*$ for each objective $m$. Consequently, algorithms like `MOG` and `MOG-R`, which select more diverse arms, are better suited for this task than `MOG-WR`. Nevertheless, by the end of 500 rounds, `MOG-WR` also achieved higher Pareto front approximation accuracy compared to other existing algorithms.

### K.2.3 OBJECTIVE FAIRNESS

As shown in Figures 11 and 12, we experimentally verified that our proposed algorithms, `MOG` and `MOG-R`, satisfy objective fairness. In the fixed feature setup, the objective fairness index ($\text{OFI}_{0.05,T}$) of both `MOG` and `MOG-R` was observed to converge approximately to $\frac{1}{M}$ regardless of the number of arms $K$, which is consistent with Theorems 2 and 4. This indicates that `MOG` and its randomized version consistently select the optimal arms for all objectives, ensuring that no objective is ignored.

In contrast, for the `MOG-WR` algorithm, even considering that it is based on generalized objective fairness, the $\text{OFI}_{0.05,500}$ decreased significantly as the dimension $d$ increased. This phenomenon arises because the difficulty of obtaining weighted vectors in directions close to specific objectives increases with higher dimensions. Specifically, we set $\mathcal{D}$ to dirichlet$(1, \ldots, 1)$, resulting in weight vectors being sampled uniformly from $\Delta^M$. When the distribution was adjusted to favor sampling near the vertices of $\Delta^M$ (e.g., dirichlet$(0.5, \ldots, 0.5)$), the objective fairness index can be improved.

In the contextual setup, the $\text{OFI}_{0.05,500}$ values were generally much higher than those observed in the fixed setup. This can be attributed to the experimental setting, where arms were uniformly generated in $\mathbb{B}^d$, causing near-optimal arms for each objective to overlap more frequently. In other words, a single arm often became the near-optimal arm for multiple objectives, resulting in a higher $\text{OFI}_{0.05,500}$. Interestingly, contrary to our intuition, `MOG-WR` exhibited higher $\text{OFI}_{0.05,500}$ values than `MOG` and `MOG-R` in this setup. This phenomenon occurs because selecting optimal arms in weighted objective directions becomes more advantageous than selecting optimal arms for each individual objective.

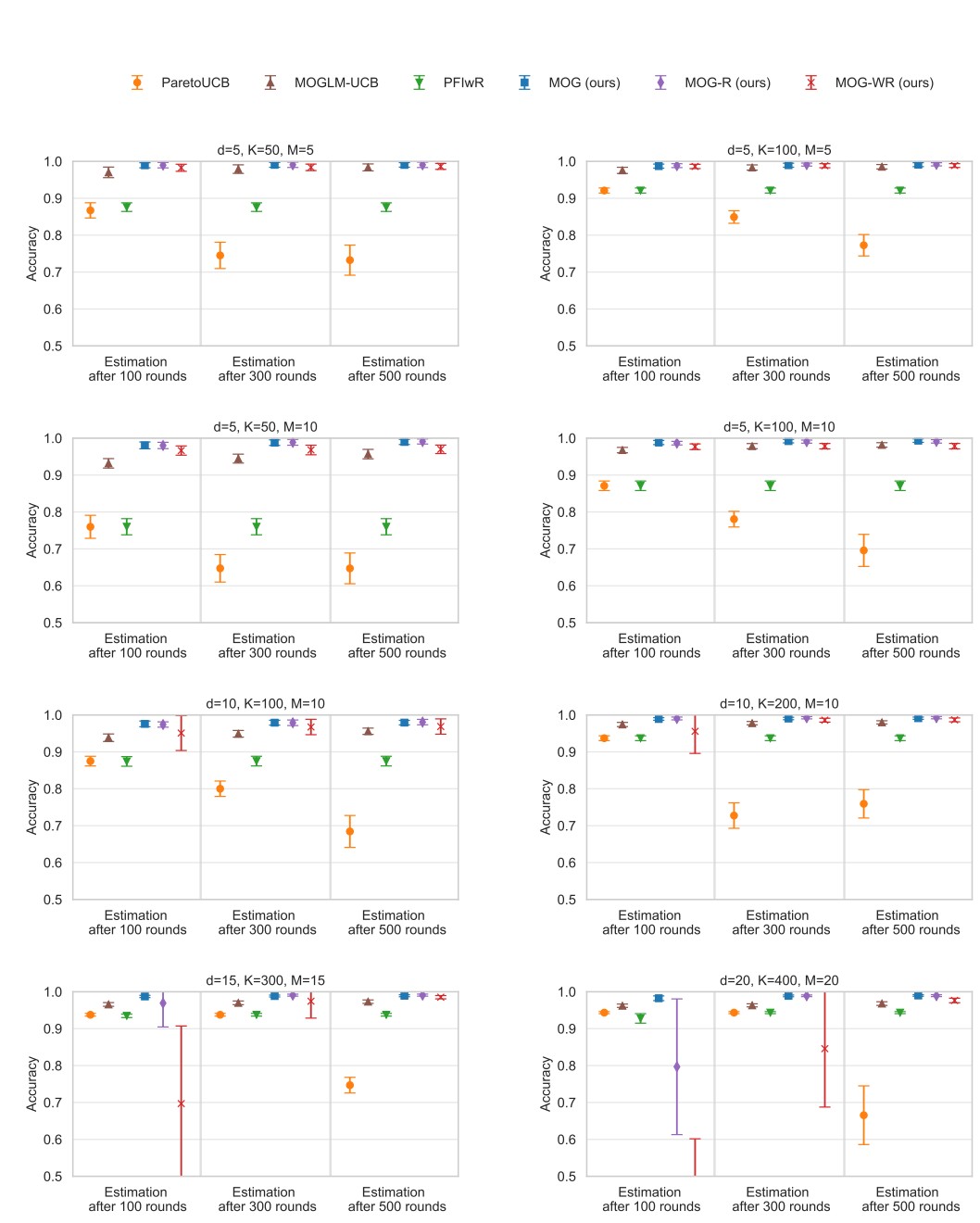

Figure 10: Pareto front estimation accuracy of multi-objective bandit algorithms across various $(d, K, M)$ combinations. For each algorithm, the error bars represent $\pm$ the standard deviation.

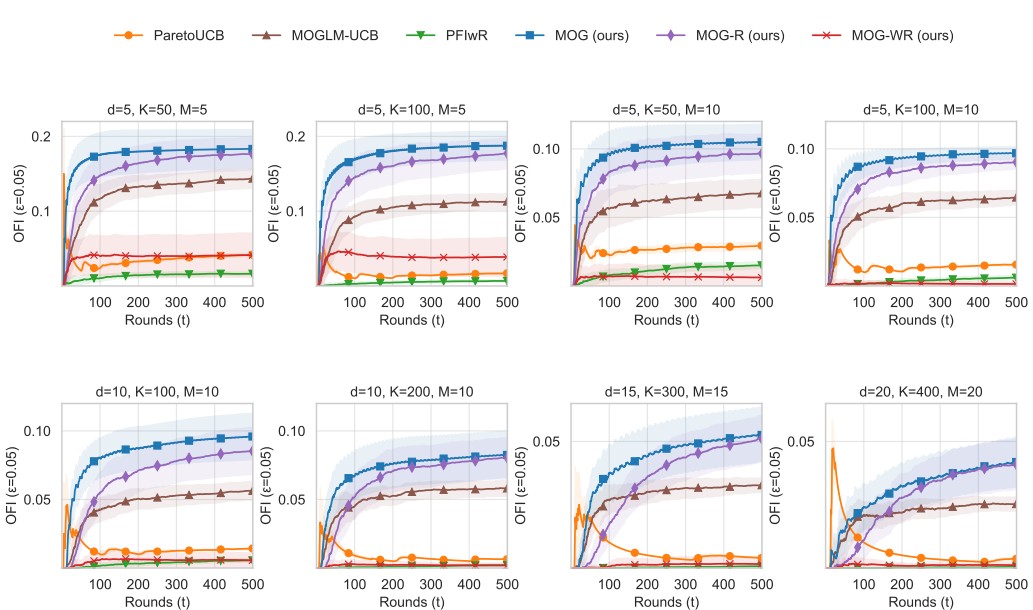

Figure 11: Objective fairness index ($\epsilon = 0.05$) of multi-objective bandit algorithms with fixed arms across various $(d, K, M)$ combinations. The shaded areas represent $\pm$ half the standard deviation for each algorithm.

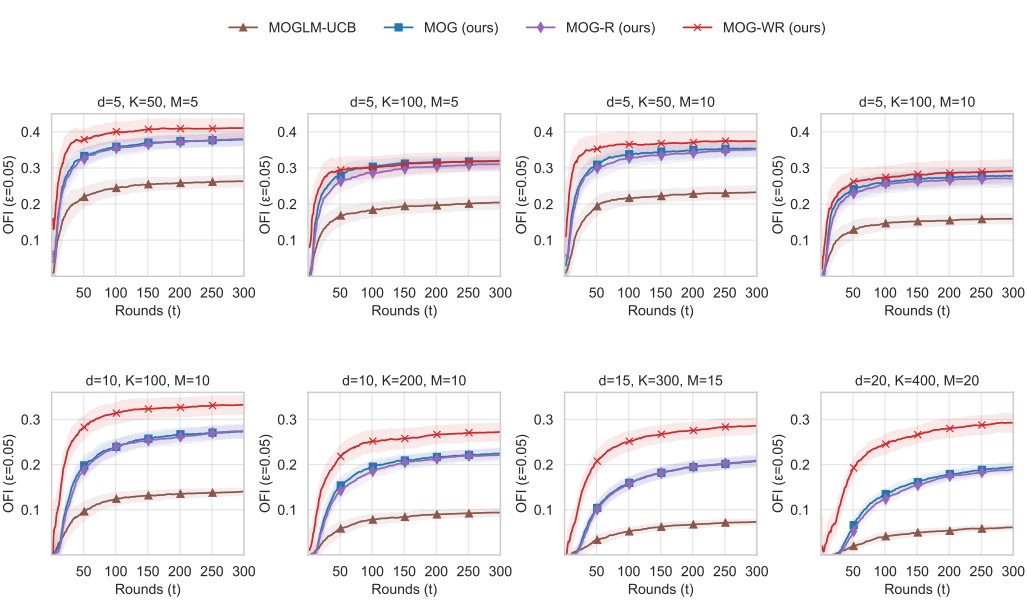

Figure 12: Objective fairness index ($\epsilon = 0.05$) of multi-objective bandit algorithms with stochastic contexts across various $(d, K, M)$ combinations. The shaded areas represent $\pm$ half the standard deviation for each algorithm.

### K.3 EFFECT OF PARAMETER SETTINGS ON ALGORITHM PERFORMANCE

#### K.3.1 EFFECT OF $B$

We conduct experiments to demonstrate that our algorithm is not particularly sensitive to the choice of $B$, and that free exploration still occurs effectively even when $B$ is set to a relatively small value. The experimental setup is identical to that of the previous experiments, and the results are averaged over 10 repetitions for each of 10 independent problem instances per $(d, K, M)$ combination.

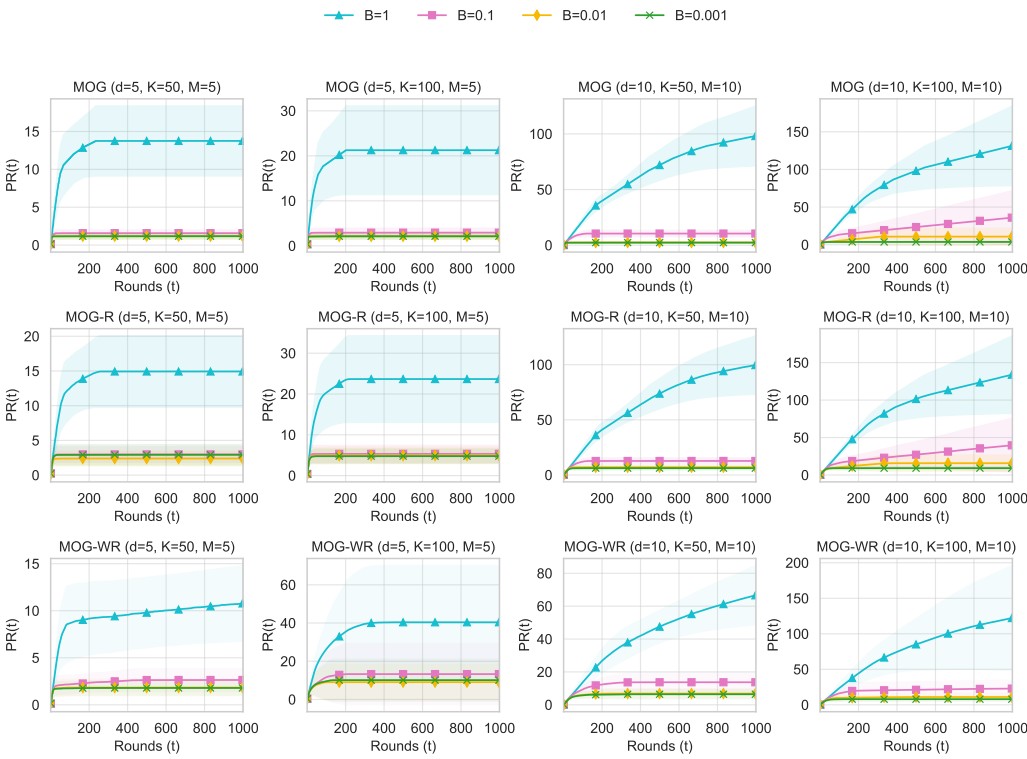

Figure 13: Cumulative Pareto regret of the `MOG`, `MOG-R`, and `MOG-WR` with fixed features across various $B$ values.

As shown in Figure 13 and Figure 15, free exploration emerges robustly for even very small values of $B$ in both fixed feature and stochastic context settings. This is because multiple objectives naturally induce sufficient diversity among the selected arms, thereby reducing the need for dedicated initial rounds. Furthermore, objective fairness is consistently satisfied across all cases (Figure 14, Figure 16).

#### K.3.2 EFFECT OF INITIAL OBJECTIVE PARAMETERS

We examined the influence of the parameter $B$ and the configurations of $\beta_1, \ldots, \beta_M$ on the performance of the `MOG` algorithm. Specifically, we considered three distinct combinations with varying degrees of diversity, as summarized in Table 1. For each parameter setting, we assessed the cumulative Pareto regret of `MOG`. The experiments were conducted under a stochastic context across various $(d, K, M)$ combinations. We evaluated the performance of the algorithm under three cases, $B = 1$, $0.1$, and $0.01$, and examined how the degree of diversity in the initial objective parameters affects learning.

Figure 17 illustrates the differences in cumulative Pareto regret for each parameter setting when $(d, K, M) = (5, 50, 5)$ and $(10, 100, 10)$. Our `MOG` algorithm demonstrated stable performance across all initial objective parameter combinations proposed in the stochastic context setup. Observing the inflection points in the graphs, we found that using highly diverse initial objective parameter combinations allowed the algorithm to complete the initial exploration phase the fastest. However, regret was lower when using less diverse combinations. As explained in Appendix K.2, this result

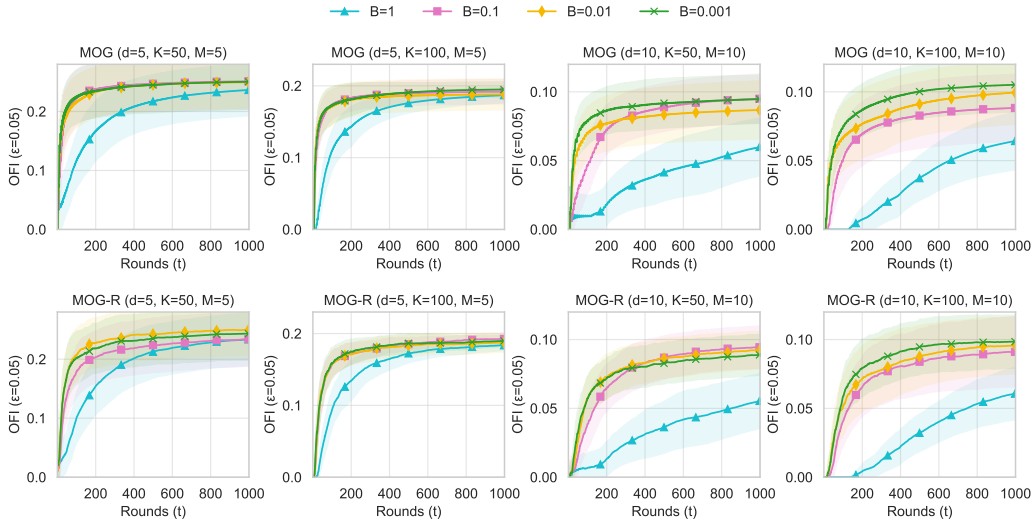

Figure 14: Objective fairness index of the `MOG` and `MOG-R` algorithms with fixed features across various $B$ values.

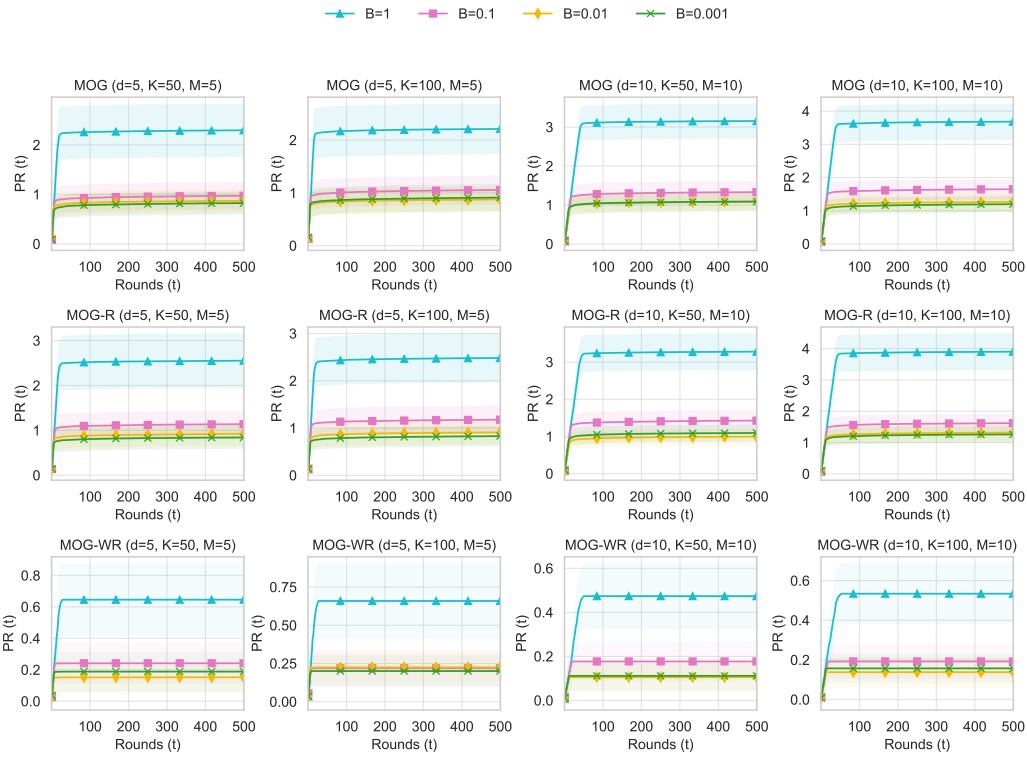

Figure 15: Cumulative Pareto regret of the `MOG`, `MOG-R` and `MOG-WR` algorithms with stochastic contexts across various $B$ values.

is related to the probability that the initial objective parameters lie within the region formed by the weighted sum of Pareto optimal arms. In cases where there is some prior knowledge of the objective parameters, initial objective parameters in intermediate directions are less likely to generate regret. This outcome is also possible in our experimental setup because the objective parameters were sampled from the positive part of $\mathbb{B}^d$. Therefore, in the absence of any prior knowledge, lower diversity

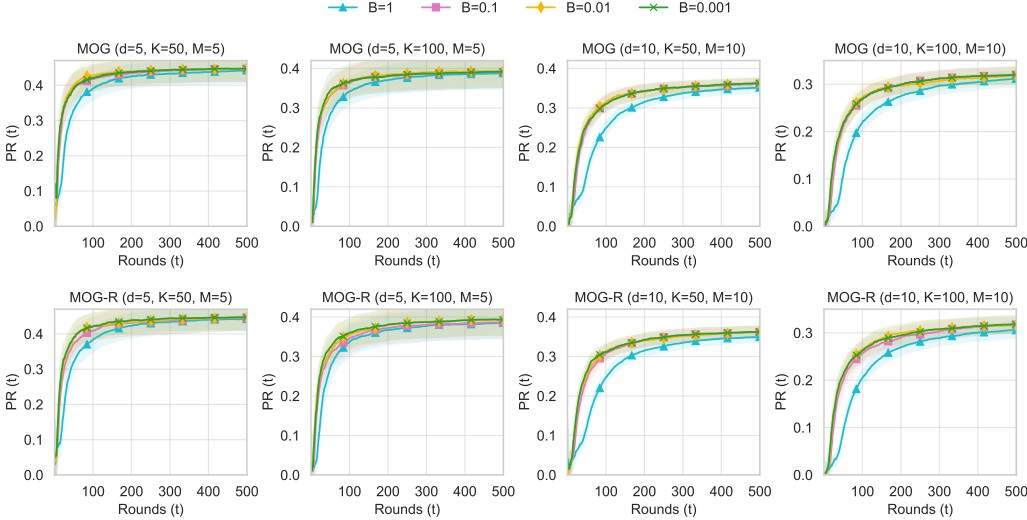

Figure 16: Objective fairness index of the `MOG` and `MOG-R` algorithms with stochastic contexts across various $B$ values.

Table 1: Combinations of initial objective parameters used in the experiments

| $M$ | Diversity | $\beta_1, \ldots, \beta_M$ |
|---|---|---|
| 5 | high | $e_1^{(5)}, \ldots, e_5^{(5)}$ |
| | moderate | $\frac{1}{\sqrt{2}}\left(e_1^{(5)} + e_2^{(5)}\right), \frac{1}{\sqrt{2}}\left(e_2^{(5)} + e_3^{(5)}\right), \ldots, \frac{1}{\sqrt{2}}\left(e_5^{(5)} + e_1^{(5)}\right)$ |
| | low | $\frac{1}{\sqrt{3}}\left(e_1^{(5)} + e_2^{(5)} + e_3^{(5)}\right), \frac{1}{\sqrt{3}}\left(e_2^{(5)} + e_3^{(5)} + e_4^{(5)}\right), \ldots, \frac{1}{\sqrt{3}}\left(e_5^{(5)} + e_1^{(5)} + e_2^{(5)}\right)$ |
| 10 | high | $e_1^{(10)}, \ldots, e_{10}^{(10)}$ |
| | moderate | $\frac{1}{\sqrt{3}}\left(e_1^{(10)} + e_2^{(10)} + e_3^{(10)}\right), \frac{1}{\sqrt{3}}\left(e_2^{(19)} + e_3^{(10)} + e_4^{(10)}\right), \ldots, \frac{1}{\sqrt{3}}\left(e_{10}^{(10)} + e_1^{(10)} + e_2^{(10)}\right)$ |
| | low | $\frac{1}{\sqrt{6}}\left(\sum_{m=1}^{6} e_m^{(10)}\right), \frac{1}{\sqrt{6}}\left(\sum_{m=2}^{7} e_m^{(10)}\right), \ldots, \frac{1}{\sqrt{6}}\left(e_{10}^{(10)} + \sum_{m=1}^{5} e_m^{(10)}\right),$ |

in the initial objective parameters may not be advantageous. In such cases, it is recommended to use diverse $\beta_1, \ldots, \beta_M$ to facilitate rapid initial exploration.

Consistent with the results in the previous section, the algorithm demonstrates strong regret performance across all values of $B$ ($B = 1$, 0.1, 0.01), with smaller $B$ values completing the initial exploration phase more rapidly. The best performance was achieved with $B = 0.01$, indicating that a brief initial exploration was sufficient. This finding is consistent not only in the stochastic context setup but also in the fixed feature setup, as shown in Figure 8.

### K.4 EXPERIMENT BASED ON REAL-WORLD WINE DATA (UCI MACHINE LEARNING REPOSITORY)

#### K.4.1 SETTINGS

We conducted experiments using the wine dataset from the UCI Machine Learning Repository to evaluate the performance of our algorithm in a bandit setting. The dataset contains 13 numerical attributes for each wine (Table 2); among these, we used alcohol, quality, and red as reward objectives, while the remaining 10 attributes were used as features. Figure 18 illustrates how the offline data was adapted for the bandit experimental setup. For each reward objective, we first performed linear regression on the normalized features. Then, in each round, rewards were generated by adding noise to the predicted value based on the regression model. The noise was sampled from $\mathcal{N}(0, 1)$ to

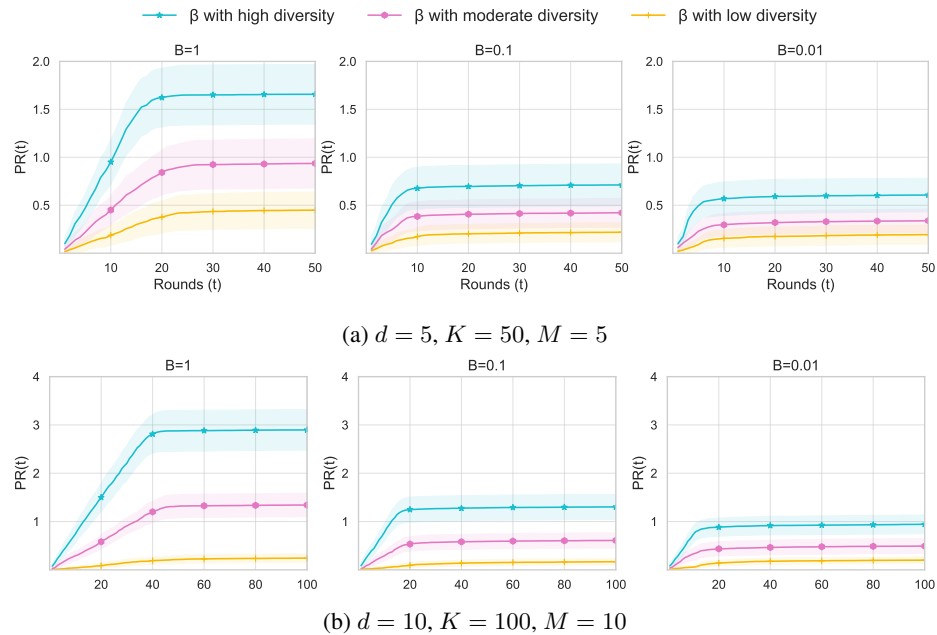

(a) $d = 5$, $K = 50$, $M = 5$

(b) $d = 10$, $K = 100$, $M = 10$

Figure 17: Cumulative Pareto regret of the `MOG` algorithm across various parameter combinations

mimic the variability observed in the original dataset. Experiments were conducted under two settings, $K = 50$ and $K = 100$, with 100 episodes of 500 rounds each being generated for evaluation.

Table 2: 3-objective bandit problem construction using off-line wine dataset.

| **Features** | fixed acidity | volatile acidity | citric acid | residual sugar | chlorides |
| | free sulfur dioxide | total sulfur dioxide | density | pH | sulphates |
| **Reward** | alcohol | quality | red | | |

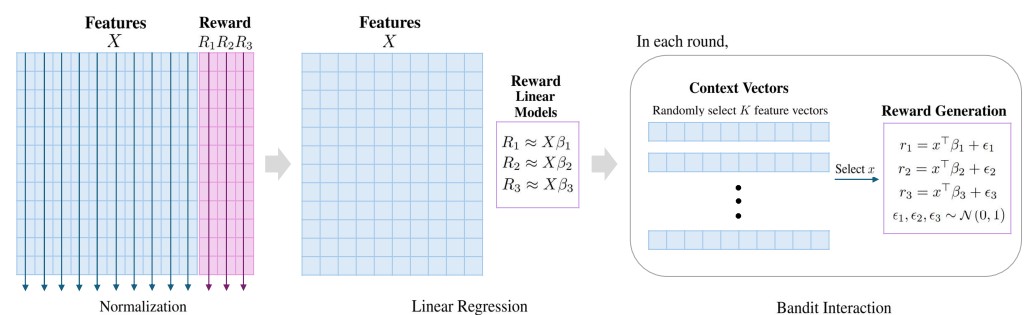

Figure 18: Description of how to use real-world off-line data for 3-objective bandit experiment.

In this real-world-inspired experiment, we measured the cumulative reward to compare the performance of our algorithms with that of the existing contextual multi-objective bandit algorithm, `MOGLM-UCB`. The parameter for `MOGLM-UCB` was set to $c = 1$ or $0.1$ as in Lu et al. (2019), and the experimental configuration for `MOG` is provided in Table 3.

Table 3: Parameter settings of `MOG` and its variants.

| Algorithm | Settings |
|---|---|
| `MOG` | $B = 1$ and $0.1$ |
| `MOG-R` (ours) | $B = 1$ and Uniform distribution for target objective selection |
| `MOG-WR` (ours) | $B = 1$ and Dirichlet($\alpha$) distribution for weight vector generation where $\alpha = (1, 1, 1), (2, 1, 1), (1, 2, 1), (1, 1, 1.5), (1, 1, 2)$ |

### K.4.2 RESULTS

Table 4 and Table 5 report the average cumulative rewards obtained by each algorithm under the two settings, $K = 50$ and $K = 100$, respectively. The `MOG-WR` algorithm demonstrates Pareto-optimal performance with respect to the cumulative rewards over the three objectives. Notably, `MOG-WR` uses weights sampled from a Dirichlet distribution, and the choice of its parameters affects which objective the algorithm tends to prioritize among the three objectives.

Table 4: Performance results of each algorithm on three objectives when $K = 50$. Algorithms marked with † achieved Pareto optimal reward performance. Boldfaced values indicate the highest reward achieved for each individual objective. The results are averaged over 100 generated episodes.

| Algorithm | Alcohol | Quality | Red |
|---|---|---|---|
| `MOGLM-UCB` ($c = 1$) | 251.79 | 93.52 | 414.66 |
| `MOGLM-UCB` ($c = 0.1$) | 314.24 | 127.58 | 359.81 |
| `MOG` ($B = 1$) | 499.25 | 187.29 | 249.38 |
| `MOG` ($B = 0.1$) | 505.23 | 195.46 | 237.53 |
| `MOG-R` ($B = 1$) | 493.70 | 188.29 | 251.33 |
| `MOG-WR` ($B = 1, \alpha = (1, 1, 1))^\dagger$ | 558.66 | 237.63 | 386.89 |
| `MOG-WR` ($B = 1, \alpha = (2, 1, 1))^\dagger$ | **709.53** | 297.25 | 223.71 |
| `MOG-WR` ($B = 1, \alpha = (1, 2, 1))^\dagger$ | 624.95 | **310.85** | 259.50 |
| `MOG-WR` ($B = 1, \alpha = (1, 1, 1.5))^\dagger$ | 441.11 | 161.83 | 566.18 |
| `MOG-WR` ($B = 1, \alpha = (1, 1, 2))^\dagger$ | 346.12 | 102.46 | **675.28** |

Table 5: Performance results of each algorithm on three objectives when $K = 100$. Algorithms marked with † achieved Pareto optimal reward performance. Boldfaced values indicate the highest reward achieved for each individual objective. The results are averaged over 100 generated episodes.

| Algorithm | Alcohol | Quality | Red |
|---|---|---|---|
| `MOGLM-UCB` ($c = 1$) | 290.26 | 109.80 | 474.66 |
| `MOGLM-UCB` ($c = 0.1$) | 348.17 | 142.29 | 418.70 |
| `MOG` ($B = 1$) | 535.11 | 203.75 | 287.56 |
| `MOG` ($B = 0.1$) | 541.99 | 209.11 | 277.22 |
| `MOG-R` ($B = 1$) | 541.72 | 203.49 | 285.69 |
| `MOG-WR` ($B = 1, \alpha = (1, 1, 1))^\dagger$ | 610.48 | 251.21 | 433.40 |
| `MOG-WR` ($B = 1, \alpha = (2, 1, 1))^\dagger$ | **765.97** | 317.67 | 266.33 |
| `MOG-WR` ($B = 1, \alpha = (1, 2, 1))^\dagger$ | 663.21 | **336.77** | 305.54 |
| `MOG-WR` ($B = 1, \alpha = (1, 1, 1.5))^\dagger$ | 480.48 | 178.30 | 612.76 |
| `MOG-WR` ($B = 1, \alpha = (1, 1, 2))^\dagger$ | 393.74 | 112.88 | **727.21** |

Figure 19 and Figure 20 show plots of cumulative rewards over time for each objective, as well as the final cumulative rewards for two of the objectives achieved by each algorithm. In particular, when the Dirichlet distribution was set to Dirichlet$(1, 1, 1.5)$ (olive point), the `MOG-WR` algorithm

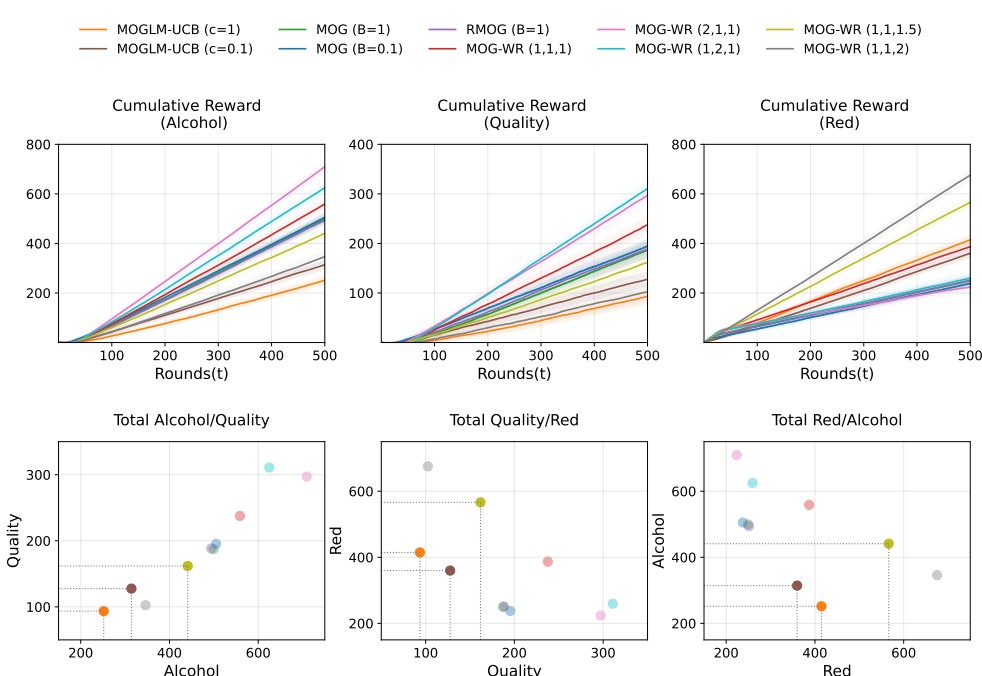

Figure 19: Cumulative reward for 3-objectives, alcohol, quality, and red, when $K = 50$.

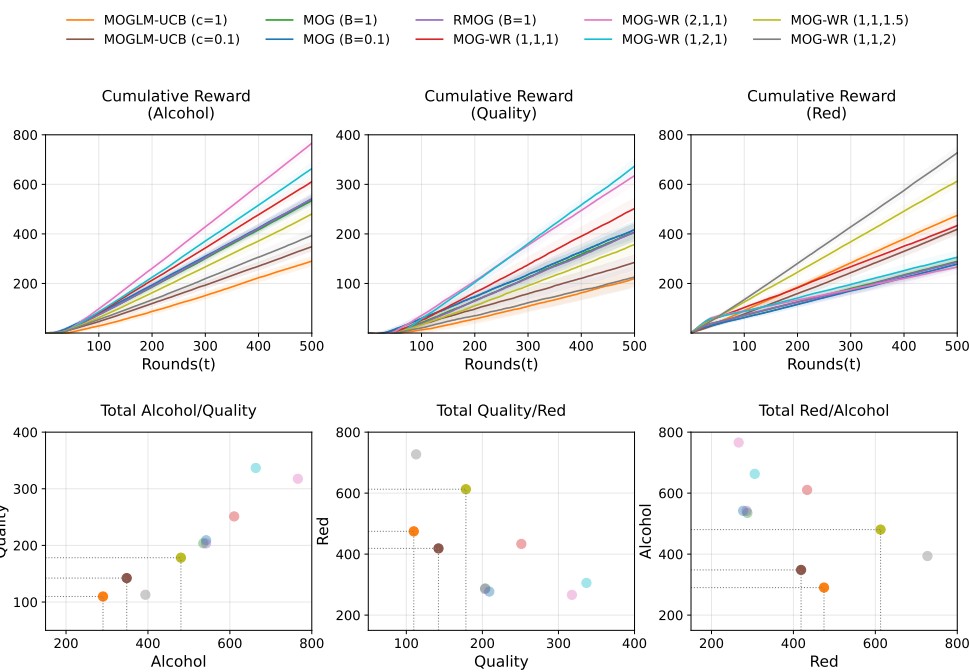

Figure 20: Cumulative reward for 3-objectives, alcohol, quality, and red, when $K = 100$.

was observed to dominate the `MOGLM-UCB` algorithm (orange and brown points) across all three objectives.

# L    AUXILIARY LEMMAS

**Lemma 23** (Lemma A.1. of Kannan et al. (2018)). *Let $\eta_1, \ldots, \eta_t$ be independent $\sigma^2$-subgaussian random variables. Let $x_1, \ldots, x_t$ be vectors in $\mathbb{R}^d$ with each $x_s$ chosen arbitrarily as a function of $(x_1, \eta_1), \ldots, (x_{s-1}, \eta_{t'-1})$ subject to $\|x_s\| \leq x_{\max}$. Then with probability at least $1 - \delta$,*

$$\left\| \sum_{s=1}^{t} \eta_s x(s) \right\| \leq \sigma \sqrt{2x_{\max} dt \log(dt/\delta)}.$$

Note that, the above lemma holds even when $\eta_1, \ldots, \eta_t$ be conditionally $\sigma^2$-subgaussian random variables, because it was driven by using $\sigma^2$-subgaussian martingale.

**Lemma 24** (Lemma 8 of Li et al. (2017)). *Given $\|x_i\| \leq 1$ for all $i \in [K]$, suppose there is an integer $m$ such that $\lambda_{\min}(V_m) \geq 1$, then for any $\delta > 0$, with probability at least $1 - \delta$, for all $t \geq m + 1$,*

$$\|S_t\|_{V_t^{-1}}^2 \leq 4\sigma^2 \left( \frac{d}{2} \log(1 + \frac{2t(x_{\max})^2}{d}) + \log(\frac{1}{\delta}) \right).$$

**Lemma 25** (Theorem 3.1 of Tropp (2011)). *Let $\mathcal{H}_1 \subset \mathcal{H}_2 \cdots$ be a filtration and consider a finite sequence $\{X_k\}$ of positive semi-definite matrices with dimension $d$ adapted to this filtration. Suppose that $\lambda_{\max}(X_k) \leq R$ almost surely. Define the series $Y \equiv \sum_k X_k$ and $W \equiv \sum_k \mathbb{E}[X_k | \mathcal{H}_{k-1}]$. Then for all $\mu \geq 0$, $\gamma \in [0, 1)$ we have*

$$\mathbb{P}[\lambda_{\min}(Y) \leq (1 - \gamma)\mu \ \text{and} \ \lambda_{\min}(W) \geq \mu] \leq d\left( \frac{e^{-\gamma}}{(1-\gamma)^{1-\gamma}} \right)^{\mu/R}.$$

**Lemma 26** (Theorem 5.1 of Auer et al. (2002)). *For any $T \geq K \geq 2$, consider the multi-armed bandit problem such that the probability slot machine pays $1$ is set to $\frac{1}{2} + \frac{1}{4}\sqrt{\frac{K}{T}}$ for one uniformly chosen arm and $\frac{1}{2}$ for the rest of $K - 1$ arms. Then, there exists $\gamma$ such that for any (multi-armed) bandit algorithm choosing action $a_t$ at time $t$, the expected regret is lower bounded by*

$$\mathbb{E}\left( p_i T - \sum_{t=1}^{T} r_{t,a_t} \right) = \Omega(\sqrt{KT}).$$

**Lemma 27.** *For any random variable vector $X \sim D$, $\mathbb{E}[XX^\top] \succeq \mathbb{E}[X]\mathbb{E}[X]^\top$*

*Proof of Lemma 27.* For any $u \in \mathbb{S}^{d-1}$, $u^\top \mathbb{E}[XX^\top]u = \mathbb{E}[u^\top XX^\top u] = \mathbb{E}[\langle u, X \rangle^2] \geq (\mathbb{E}[\langle u, X \rangle])^2 = u^\top \mathbb{E}[X]\mathbb{E}[X]^\top u.$

**Lemma 28.** *Let $v$ be a vector in $S \subset \mathbb{R}^d$ and $A$ be a $d \times d$ matrix. Then $\|Av\|_2 \geq (\min_{u \in S} u^\top A u) \|v\|_2$.*

*Proof of Lemma 28.*

$$\frac{\|Av\|_2}{\|v\|_2} = \left\| A \frac{v}{\|v\|_2} \right\|_2 \geq \min_{u \in S} \|Au\|_2 \geq \min_{u \in S} u^\top A u.$$