# OpenReview forum: "Blessings of Many Good Arms in Multi-Objective Linear Bandits"
_ICLR.cc/2026/Conference — ICLR 2026 Conference Withdrawn Submission_

### Official Review · Reviewer_JpnB · 2025-10-27

**Soundness:** 2
**Presentation:** 3
**Contribution:** 1
**Rating:** 4
**Confidence:** 4

**Summary:**

This paper investigates the multi-objective linear bandit (MOLB) problem, challenging the notion that multiple objectives necessarily lead to complex algorithms. The authors introduce a "goodness of arms" condition, which assumes that for each objective, there exists at least one arm that performs sufficiently well. Under this core assumption, the paper proposes a simple near-greedy algorithm, MOG, which cycles through objectives and selects the best arm for the current target objective. The authors provide theoretical analysis showing that MOG achieves sub-linear $\tilde{\mathcal{O}}(\sqrt{T})$ Pareto regret, notably without relying on context diversity assumptions. Furthermore, the paper introduces a new "objective fairness" metric to ensure that the algorithm does not neglect any single objective, and proves MOG satisfies this criterion.

**Strengths:**

- The paper is well-motivated, addressing the high computational complexity of many existing multi-objective bandit algorithms that require, for example, repeatedly updating an empirical Pareto front. The goal of finding a simpler, more efficient algorithm is highly relevant.

**Weaknesses:**

- The proposed MOG algorithm, which greedily optimizes one objective at a time in a round-robin fashion, is fundamentally unsuited for multi-objective optimization. The very purpose of MOLB is to find arms on the Pareto front that represent a compromise between conflicting objectives. The MOG algorithm is structurally incapable of this. For instance, in a simple 2-objective problem with optimal arms [1, 0], [0, 1], and [0.8, 0.8], MOG will only pull [1, 0] (when $m=1$) and [0, 1] (when $m=2$). It will never identify or pull the [0.8, 0.8] arm, which is arguably the most satisfactory and practical solution. The algorithm is not "rational" from a multi-objective standpoint; it is simply $M$ non-communicating single-objective algorithms.
- The paper's entire analysis, and the only reason the flawed MOG algorithm "works" (i.e., achieves low regret), hinges on the "Goodness of Arms" (GoA) assumption (Definition 5). This assumption essentially posits that for each objective $m$, there is an arm $i$ that is already "good enough" for it. This is problematic as it bypasses the core challenge of finding trade-offs by assuming "good" solutions for each objective already exist. Furthermore, it replaces the standard (and strong) "context diversity" assumption with an equally strong, if not stronger, "problem structure" assumption. The paper's claim to be the first to work "without any diversity assumptions" is misleading, as it has simply swapped one restrictive assumption for another.
- The adaptation of "Objective Fairness" is not a strength but a further weakness. This metric merely verifies that the algorithm pays attention to each objective, which MOG does by design (due to its round-robin nature). However, as the [0.8, 0.8] arm example shows, an algorithm can satisfy this fairness criterion while completely failing to find the most rational multi-objective compromise. The metric is designed to make this irrational algorithm look successful, but it does not measure what actually matters in this problem.
- The algorithm itself (greedy round-robin with OLS) is not novel. The paper's contribution is its analysis. But since this analysis only holds under the restrictive GoA assumption, which in turn only justifies a conceptually flawed algorithm, the contribution is negligible. The paper does not show that "a simple algorithm can solve MOLB"; it shows that "if the MOLB problem is structured to be trivially easy (per GoA), a simple (and flawed) algorithm works." This is not a significant result for ICLR.

**Questions:**

- The entire analysis hinges on the $\gamma$-goodness assumption. How does the algorithm's Pareto regret degrade as $\gamma \rightarrow 0$? The practical utility of this algorithm seems entirely dependent on this problem-specific parameter, which is not explored.
- The paper states Assumption 2 (that $\theta_m^*$ span $\mathbb{R}^d$) is without loss of generality 18, relaxable to spanning the feature space19. What happens if this assumption is not met, which is a very practical scenario?

---

> ### Author Response · Authors · 2025-11-22
> **Rebuttal by Authors [1/2]**
>
> Thank you for taking the time to review our paper and for providing constructive feedback.
>
> ---
>
> #### **Selection Strategy**
>
> First, we would like to emphasize that our $\texttt{MOG}$ algorithms refer to a **class** of multi-objectvive (near) greedy algorithms, and that **multiple selection strategies (other than standard round-robin version) are possible** (lines 251–256). For example, in the appendix, we introduce $\texttt{MOG-WR}$, which selects a set of arms along the positive side of the Pareto front (Figure 2), rather than only optimal arms for individual objectives. Although many other variants are certainly possible, we chose to present the round-robin–based $\texttt{MOG}$ as the standard version because it most transparently illustrates *how the multi-objective structure naturally leads to free exploration*.
>
>
> Moreover, we would like to clarify that *identifying* the Pareto front and *selecting* all Pareto-optimal arms are distinct objectives. If the intended meaning of "finding the Pareto front" is identifying it, then **$\texttt{MOG}$ can approximate the Pareto front** on demand (and, in fact, can do so more efficiently than several existing MOLB algorithms), even though it selects only a subset of the Pareto-optimal arms. In addition, which Pareto-optimal arm is preferable can vary depending on the problem setting. For example, in settings where a dominant objective exists in each round, selecting only $[1,0]$ or $[0,1]$ rather than $[0.8, 0.8]$ is entirely reasonable, and in such cases, $\texttt{MOG}$ and $\texttt{MOG-R}$ are appropriate choices.
>
> ---
>
> #### **Goodness of arms**
>
>
> Importantly, assuming the existence of “good" solutions for each objective does not imply that the algorithm can automatically find those solutions, and therefore it does **not make the problem trivially easy**.  If the intended meaning behind the reviewer's use of the term "finding trade-offs" is “finding Pareto front", then our algorithms do **not avoid this challenge** but is fully capable of addressing it. As we state in the previous paragraph, our algrotihms can estimate the Pareto front on demand.
>
> Furthermore, our claim is **not** that the goodness-of-arm condition is *weaker or stronger* than previously studied assumptions in the free-exploration literature. Rather, our point is that it represents a fundamentally **different** condition, and that our work is the first to reveal a *novel type of free exploration* that emerges specifically from this condition. Moreover, this condition naturally arises in many real-world applications—such as recommendation systems—where each objective typically has at least one arm (or item) that performs reasonably well. For reference, Bayati et al. (2020) assume a condition similar to (or slightly stronger than) ours in the MAB setting (Appendix C.2) and establish statistical guarantees for a greedy algorithm.

---

> ### Author Response · Authors · 2025-11-22
> **Rebuttal by Authors [2/2]**
>
> #### **Fairness**
>
> The **goal** of a multi-objective algorithm varies **depending on the problem setting**, and this goal can be expressed through **different notions of fairness** within the regret-minimization framework. The choice of fairness criterion naturally leads to **different arm-selection strategies**. For example:
>
> 1. Nash Welfare
> Nash welfare aims to select an arm that guarantees reasonably high rewards across all objectives. From this perspective, $[0.8, 0.8]$ is *always* better than $[1,0]$ or $[0,1]$. Since this criterion optimizes a scalarized reward given by the product of objective-wise utilities, it necessarily leads to a unique optimal arm.
>
> 2. Fairness of Drugan & Nowé (2013).
> Much of the MO bandit literature adopts this fairness notion, which treats all Pareto-front arms equivalently. Algorithms designed under this criterion typically estimate the Pareto front and then select arms uniformly (or near-uniformly) from that set.
>
> 3. Objective Fairness (ours).
> Our notion of objective fairness requires that each objective’s near-optimal arm be selected continuously over time. Algorithms such as $\texttt{MOG}$ and $\texttt{MOG-R}$ aim to achieve objective fairness and are well suited for problem settings where a dominant objective exists in each round.
>
> 4. Generalized Objective Fairness (ours).
> This stronger notion requires that, for every direction defined by a weight-sum scalarization of the objectives, a near-optimal arm be selected continuously. The $\texttt{MOG-WR}$ algorithm is designed to achieve this by selecting arms with respect to randomized combinations of the objectives, thereby exploring the positive side of the Pareto front.
>
>
> ---
>
> #### **What if assumption 2 or 3 does not hold?**
>
> If $\gamma \rightarrow 0$, then for some objectives no sufficiently good arm exists, and the optimal arm for one objective may coincide with the optimal arm for another. In such cases, the Pareto front becomes insufficiently diverse, and free exploration no longer arises. Similarly, if the objective vectors do not span the feature space, the resulting Pareto-optimal arms may also fail to span the feature space, in which case free exploration does not occur without additional assumptions. Consequently, in both scenarios, our algorithm cannot guarantee sublinear regret. However, in practice, our near greedy algorithm performs well in a much broader range of settings, even when these assumptions are not strictly satisfied.
>
> ---
>
>
> #### **Why important?**
>
> Our work provides a theoretical justification for the intuition that, **in real-world decision-making problems where good arms are abundant, having multiple objectives can eliminate the need for additional explicit exploration**. We believe this result serves as an important foundation for the practical deployment of multi-objective algorithms. With all due respect, we are afraid that our contributions are not fairly evaluated based on the review comment.

---

### Official Review · Reviewer_y4Tp · 2025-10-31

**Soundness:** 3
**Presentation:** 3
**Contribution:** 2
**Rating:** 4
**Confidence:** 3

**Summary:**

This paper investigates multi-objective stochastic linear bandits with finite arms, contributing a novel approach by introducing objective fairness for multi-objective bandits.

**Strengths:**

The paper focuses on a central issue in multi-objective optimization. By enhancing decision diversity, it advances the resolution of multi-objective optimization problems.

**Weaknesses:**

1. How to initialize $\{\beta_1, \dots, \beta_M\}$?
2. The assumption $||\theta_m^\*|| = 1$ or  $\ell\leq ||\theta_m^*||$ is not common in single-objective or multi-objective bandit literature.
3. The assumption that $\theta_1^\*, \ldots, \theta_M^\*$ span $\mathbb{R}^d$ is not common.
4. Since maximizing a single objective can make the Pareto suboptimality gap vanish, alternating the maximization across objectives appears to be a natural approach under the proposed fairness definition. This raises the impression that the fairness metric might have been designed to align with the proposed algorithm. Could the authors clarify the design order and motivation behind the fairness definition?
5. The paper claims that multiple objectives can facilitate learning rather than hinder it. However, it is unclear where this viewpoint is theoretically or empirically demonstrated. Please point to the specific results.

**Questions:**

see the weaknesses.

---

> ### Author Response · Authors · 2025-11-22
> **Rebuttal by Authors**
>
> #### **$\beta_1, \ldots, \beta_M$ initialization**
>
> Although Algorithm 1 is stated so that it can be applied to both the fixed-feature setting and the time-varying context setting, in the fixed setting it is often simpler to select an **arm set that spans the feature space** in a round-robin fashion, rather than explicitly specifying $\beta_1, \ldots, \beta_M$. In stochastic-context settings, the simplest way to choose $\beta_1, \ldots, \beta_M$ is to set them as the **basis vectors** of the feature space. Such forced sampling immediately yields a $\log T$ guarantee for $T_0$. Moreover, we have empirically confirmed that our algorithm performs robustly across a wide range of choices for $\beta_1, \ldots, \beta_M$ (Appendix K.3.2).
> Thank you for your question, and we will provide a more concrete description of the initial exploration phase in the algorithm in the revised version.
>
> ---
>
> #### **Assumptions**
>
> First, we want to emphasize that research on free exploration can be viewed as identifying conditions under which (near)-greedy algorithms achieve statistical efficiency (Kannan et al., 2018; Raghavan et al., 2018;  Bayati et al., 2020; Bastani et al., 2021; Kim and Oh, 2025). In this context, the primary contribution of our work is to identify **a new condition under which (near-)greedy algorithms can achieve sublinear regret**.
>
> In the main text (Assumption 1), we assume $\\|\theta_m^\*\\|_2 = 1$ for a clear presentation of the proof. However, in Appendix H, we relax this assumption to $l \le \\|\theta_m^\*\\|_2 \le L$. For a multi-objective problem to be meaningful, the relative importance (or scale) of the objectives must be at least **comparable**; if one objective parameter is significantly smaller in magnitude than the others, that objective becomes effectively negligible. From this perspective, our boundedness assumption ensures that the objectives do not differ too drastically in scale.
>
> Furthermore, for free exploration to arise, the arms chosen greedily with respect to each objective must collectively span the feature space. This requires, in turn, that the objective parameters themselves span the feature space. Accordingly, our work focuses on settings where the number of objectives is comparable to the feature-space dimension.
>
> ---
>
> #### **Fairness \& Algorithm Design**
>
> Thank you for a good question. As you noted, these two aspects are closely related. The goal of a multi-objective algorithm varies depending on the problem setting, and this goal can be expressed through different notions of fairness within the regret-minimization framework. The key point is that **the choice of fairness criterion naturally leads to different arm-selection strategies**. For example:
>
>
> 1. Fairness of Drugan & Nowé (2013) :
> Much of the MO bandit literature adopts this fairness notion, which treats all Pareto-front arms equivalently. Algorithms designed under this criterion typically estimate the Pareto front and then select arms uniformly (or near-uniformly) from that set.
>
> 2. Objective Fairness (ours) :
> Our notion of objective fairness requires that each objective’s near-optimal arm be selected continuously over time. Algorithms such as $\texttt{MOG}$ and $\texttt{MOG-R}$ aim to achieve objective fairness and are well suited for problem settings where a dominant objective exists in each round, while also being applicable to settings where any Pareto-front arm is acceptable.
>
> 4. Generalized Objective Fairness (ours) :
> This stronger notion requires that, for every direction defined by a weight-sum scalarization of the objectives, a near-optimal arm be selected continuously. The $\texttt{MOG-WR}$ algorithm is designed to achieve this by selecting arms with respect to randomized combinations of the objectives, thereby exploring the positive side of the Pareto front.
>
>
> ---
>
> #### **Multiple objectives can facilitate learning rather than hinder it**
>
> First, the key point we aimed to convey in the introduction is the observation that multiple objectives do not inherently complicate the learning process. Rather, with the goodness condition, the presence of multiple objectives introduces a new benefit—namely, a form of free exploration. We propose and analyze this new type of free exploration that arises specifically from the existence of multiple objectives, both theoretically and empirically.
>
> To the best of our knowledge, this is the first attempt to shed light on the **phenomenon of implicit exploration** that emerges uniquely in the multi-objective setting—an aspect that has been largely overlooked in the existing literature. We view our work as a first step toward understanding the additional exploration benefits brought by multiple objectives, and we believe it holds strong potential for extension to other models and algorithmic frameworks.
>
> ---
>
> Thank you for taking the time to review our paper and for providing thoughtful feedback. We hope that our responses have adequately addressed the concerns you raised.

---

> > ### Comment · Reviewer_y4Tp · 2025-11-26
> >
> > Thank you for the clarification. However, the current version still does not convincingly demonstrate the claimed advantage of the multi-objective setting.Therefore, I will keep my score unchanged.

---

### Official Review · Reviewer_adWY · 2025-10-31

**Soundness:** 3
**Presentation:** 3
**Contribution:** 2
**Rating:** 6
**Confidence:** 3

**Summary:**

This work considers the multi-objective goal for the multi-armed bandit problem. There are $k$ arms and $m$ objectives, with each objective having a different mean reward for a given arm. In such settings, there is no clear objective one should aim to maximize; instead there are various metrics that are better fitted for various settings/contexts. This paper proposes a new metric, defined as the objective fairness index. Essentially it looks to ensure that for each objective, the optimal arms for that objective is selected a constant fraction of the time. The work gives algorithms for this objective, analyzes the corresponding regret bounds, and connects it with the literature on pareto front objectives.

**Strengths:** I think the paper is relatively well written and well motivated. I have some mixed opinions on the proposed objective (see below), but I think if one takes this as a given, the suite of results presented is fairly complete. Upper and lower bounds on the regret and fairness objective achieved by the proposed algorithm are presented. The $\sqrt{T}\log(T)$ upper bound is essentially enough for algorithm agnostic regret lower bound. While the algorithm is simple, the technical analysis does require some work.

**Weakness:** The proposed objective is essentially an egalitarian objective -- we care about the welfare of the worst off arm. But I'm not note why the authors don't directly use the egalitarian objective as opposed to strange binary metric of whether an arms was sufficiently close the best arms of the objective? This can exacerbate the issues of egalitarian fairness more - namely, the *best* arm for any agent may be extremely bad for all the other agents. It thus makes sense to choose arms that are more reasonable to more participants. Given the nature of objective, as defined, the chosen algorithm is thus not unsurprising. Choose the best arm for agent in round robin fashion. Can you please justify this?

I also think the work is missing some important connections to past literature. Multi-objective bandits have matured beyond standard Pareto front goals. I especially think that a discussion on Nash Welfare as an objective (which balances utilitarian and egalitarian objectives) is richly warranted. This has been studied in multi-objective bandits; see [1,2,3].

[1]: Hossain, Safwan, Evi Micha, and Nisarg Shah. "Fair algorithms for multi-agent multi-armed bandits." Advances in Neural Information Processing Systems 34 (2021): 24005-24017

[2]: Barman, Siddharth, et al. "Fairness and welfare quantification for regret in multi-armed bandits." Proceedings of the AAAI Conference on Artificial Intelligence. Vol. 37. No. 6. 2023.

[3]: Zhang, Mengxiao, Ramiro Deo-Campo Vuong, and Haipeng Luo. "No-regret learning for fair multi-agent social welfare optimization." Advances in Neural Information Processing Systems 37 (2024): 57671-57700.

**Strengths:**

See above

**Weaknesses:**

See above

**Questions:**

See above

---

> ### Author Response · Authors · 2025-11-22
> **Rebuttal by Authors**
>
> Thank you for taking the time to review our paper and for providing thoughtful and constructive feedback. We sincerely appreciate your recognition of our contributions and the valuable comments you have shared.
>
> First, we would like to emphasize that our $\texttt{MOG}$ algorithms refer to a **class** of multi-objectvive (near) greedy algorithms, and that **multiple selection strategies (other than standard round-robin version) are possible** (lines 251–256). For example, in the appendix, we introduce $\texttt{MOG-WR}$, which selects a set of arms along the positive side of the Pareto front (Figure 2). The issue of egalitarian fairness you raised **does not pose a significant concern** for $\texttt{MOG-WR}$. Although many other variants are certainly possible, we chose to present the round-robin–based $\texttt{MOG}$ as the standard version because it most transparently illustrates *how the multi-objective structure naturally leads to free exploration*.
>
> The **goal** of a multi-objective algorithm varies **depending on the problem setting**, and this goal can be expressed through **different notions of fairness** within the regret-minimization framework. The choice of fairness criterion naturally leads to **different arm-selection strategies**. For example:
>
> 1. **Nash Welfare** :
> Nash welfare is typically studied in the **multi-agent setting** rather than the multi-objective setting and the references you cited also consider the multi-agent case. Nash welfare aims to select an arm that guarantees reasonably high rewards across all agents, effectively optimizing a **scalarized reward** given by the product of agent-wise utilities. This leads to a **unique optimal arm**.
> However, in the **multi-objective** literature, it is far more common to adopt approaches that identify and utilize a **diverse set of Pareto-optimal arms**, rather than selecting a single optimal arm (Drugan and Nowé, 2013; Turgay et al., 2018; Tekin and Turgay, 2018; Lu et al., 2019; Xu and Klabjan, 2023; Kim et al., 2023; Cheng et al., 2024; Crepon et al., 2024). In this context, our work also focuses on the exploration-free behavior that arises from selecting diverse Pareto-optimal arms; hence, we do not pursue Nash-welfare-style optimization.
>
> 2. **Fairness of Drugan & Nowé (2013)** :
> Much of the MO bandit literature adopts this fairness notion, which treats all Pareto-front arms equivalently. Algorithms designed under this criterion typically estimate the Pareto front and then select arms uniformly from that set.
>
> 3. **Objective Fairness (ours)** :
> Our notion of objective fairness requires that each objective’s near-optimal arm be selected continuously over time. Algorithms such as $\texttt{MOG}$ and $\texttt{MOG-R}$ aim to achieve objective fairness and are well suited for problem settings where a dominant objective exists in each round, while also being applicable to settings where any Pareto-front arm is acceptable.
>
> 4. **Generalized Objective Fairness (ours)** :
> This stronger notion requires that, for every direction defined by a weight-sum scalarization of the objectives, a near-optimal arm be selected continuously. The $\texttt{MOG-WR}$ algorithm is designed to achieve this by selecting arms with respect to randomized combinations of the objectives, thereby exploring the positive side of the Pareto front.
>
> Thank you again for your engagement.

---

### Official Review · Reviewer_rDEZ · 2025-10-31

**Soundness:** 1
**Presentation:** 3
**Contribution:** 2
**Rating:** 2
**Confidence:** 4

**Summary:**

The paper studies multi-objective linear bandits with $M$ objectives and proposes $\mathrm{MOG}$, a near-greedy algorithm that cycles objectives: before an OLS “estimation” phase it pushes the Gram matrix’s minimum eigenvalue above a threshold $B$ (i.e., until $\lambda_{\min}(V_{t-1}) \ge B$), then plays greedy per objective in round-robin; two variants (randomized $\mathrm{MOG}$-$\mathrm{R}$ and weighted-randomized $\mathrm{MOG}$-$\mathrm{WR}$) are also introduced. The central structural assumption is the existence of *$\gamma$-good arms* near each objective direction (Assumption 3), which implies linear growth of $\lambda_{\min}$ of the Gram matrix once residuals are small. The paper claims $\widetilde{O}(\sqrt{dT})$ Pareto regret for $\mathrm{MOG}$ (and variants) and an objective-fairness guarantee showing each objective is served at asymptotic frequency $1/M$ (or $p_m$ for $\mathrm{MOG}$-$\mathrm{R}$), with matching $\Omega(\sqrt{dT})$ lower bound.

**Strengths:**

- **A nice observation**: if each objective has many $\gamma$-good arms, the per-objective greedy steps jointly grow the design’s $\lambda_{\min}$ linearly; Lemma 2 formalizes this and enables $\widetilde{O}(\sqrt{dT})$ regret.
- **A new fairness index**: the authors propose a new fairness index that does not necessitate computing the empirical Pareto front in each iteration; the objective fairness index bound shows $\lim_{T\to\infty}\mathrm{OFI}_{\varepsilon,T}=1/M$ (or $\min_m p_m$ for $\mathrm{MOG}$-$\mathrm{R}$).

**Weaknesses:**

- **“Free exploration’’ vs explicit gate $B$**: the algorithm enforces an initial phase until $\lambda_{\min}(V_{t-1}) \ge B$, which is a horizon-dependent exploration cost; the “no/low exploration” narrative is misleading without a horizon-free tuning rule for $B$--the authors are **forcing** exploration at the beginning!
- **Assumption 3**: First, a lower bound on the parameters is an unusual assumption to have. Second, in Assumption 3, for intuition, say $\alpha=0$. This automatically ensures a lower bound on parameter norms. This seems strong.
- **Underspecified $T_0$**: $T_0$ (rounds to reach $\lambda_{\min}\ge B$) is not algorithmically guaranteed to be sublinear; without a diversity mechanism, the pre-$B$ phase may fail to span $\mathbb{R}^d$, making $T_0$ large (even $\Theta(T)$).
- **Hidden $\alpha$-dependence**: Theorem 1’s displayed regret omits explicit $\alpha$-dependence, yet $B=B(T,\alpha,\sigma,d)$ and the additive $4T_0$ inherit it; please state this dependence or bound $T_0(\alpha,\gamma,\lambda)$ explicitly.
- **Lemma 2**: First, the bound on estimation error should be stochastic. Second, the assumption $||\hat\theta_m-\theta_m^\ast||\le \alpha$ after $T_0$ is not ensured by the stated selection rule in the pre-$B$ phase. Third, the statement says that the estimation error should be less than $\alpha$, but the stated result does not depend on $\alpha$. The way the result is stated lacks rigorous and clarity.
- **Corollary 1 premise external to algorithm**: it assumes the initial feature set spans $\mathbb{R}^d$, but the algorithm does not enforce this; the resulting $O(\log T)$ claim for $T_0$ is conditional rather than guaranteed.
- **Ambiguity in “learning’’ / “simpler solutions’’**: early claims (in the introduction) are qualitative and somewhat opaque--necessitating a clearer exposition.
- **Role of $\alpha$ in “goodness’’**: unclear whether $\alpha$ is an exogenous geometric margin (property of the arm set) or an endogenous estimation tolerance.
- **Motivation for $\mathrm{MOG}$-$\mathrm{R}$ unclear**: the concrete drawback addressed (non-uniform shares, stochastic-context stability, or cycling artifacts) is not crisply articulated.
- **Horizon coupling of $B$**: $B$ is specified in theorems as a function of $T$ but presented in algorithms as a given threshold; include the explicit selection rule (or a doubling scheme) in the algorithmic description.

**Questions:**

- **Pre-$B$ phase diversity**: Can you provide a constructive rule (and guarantee) ensuring that the set $S$ collected while $\lambda_{\min}(V_{t-1})<B$ spans $\mathbb{R}^d$ with a quantitative lower bound on $\lambda_{\min}$? This would make Corollary 1 a consequence of the algorithm rather than an external condition.
- **Horizon-free $B$ (minor)**: How would you tune $B$ without knowing $T$? Is a doubling-trick analysis feasible without worsening the leading $\widetilde{O}(\sqrt{dT})$ term?
- **$\alpha$ as assumption vs target**: Is $\alpha$ part of the arm-set geometry (i.e., an exogenous margin in the definition of “goodness’’) or an endogenous estimation tolerance achieved once $\lambda_{\min}\ge B$? Where exactly does $\psi(\lambda,\gamma)$ enter operationally?
- **Role of $\mathrm{MOG}$-$\mathrm{R}$**: What concrete failure mode of $\mathrm{MOG}$ does $\mathrm{MOG}$-$\mathrm{R}$ address (e.g., non-uniform target shares $p_m$, stabilization under stochastic contexts, mitigation of cycling artifacts)? A small illustrative example would help.
- **“Free exploration’’ claim**: In what formal sense is exploration “free’’ if the algorithm requires an initial gate $\lambda_{\min}\ge B$ with $B=B(T,\alpha,\sigma,d)$?

---

> ### Author Response · Authors · 2025-11-22
> **Rebuttal by Authors [1/2]**
>
> Thank you for taking the time to review our paper and for providing constructive feedback. Your comments provide valuable insights that will help us further improve the rigor and clarity of our work. At the same time, with all due respect, several points appear to stem from fundamental misunderstandings of our main arguments/results, which we sincerely would like to clarify. Below, we provide our responses to each point, organized by topic.
>
> ---
>
> ### **"Free exploration" claim**
> We use the term *"free exploration"* to indicate that,
>
> *"natural exploration arises even when the algorithm primarily takes exploitation actions"*.
>
> We proved that this phenomenon indeed **occurs** after the initial $O(\log T)$ exploration rounds, when our proposed (near-)greedy algorithms are executed. In other words, our use of the term does **not** imply the absence of exploration or the use of only a constant amount of it; rather, it emphasizes the phenomenon that natural exploration occurs after forced sampling phase. That said, if this terminology feels imprecise, we are open to revising it to convey our intent more clearly.
>
> ---
>
> ### **Assumption 3**
> Assumption 3 does **not** impose any bound on the norm of the parameter. While the original assumption pertains only to the existence of a positive $\alpha$, as you suggested, let us also consider the case where $\alpha = 0$. In this case, $\beta$ must serve as one of the objective parameters, and the condition “there exist $k \in [K]$ such that $x_k^\top {\beta \over \\|\beta\\|_2} \ge \gamma$” simply means that *there exists at least one arm whose reward is high in the direction of that objective parameter*. This condition is unrelated to the magnitude of the parameter norm, since the left-hand side of the inequality involves the normalized form of $\beta$.
>
> ---
>
> ### **Initial Exploration (Pre-$B$ phase)**
> First, we sincerely appreciate your detailed comments regarding the initial exploration (pre-$B$) phase. Following your suggestion, we will explicitly incorporate a more detailed **description of the arm-selection procedure** during this phase into the algorithm in the revised version. As stated in Corollary 1, if the algorithm selects arms in a round-robin fashion within a feature set that spans $\mathbb{R}^d$, then the bound $T_0 = O(\log T)$ is achievable. In the fixed setting of the main paper, Assumptions 2 and 3 ensure that the arms span $\mathbb{R}^d$, which makes such forced sampling feasible. In addition, we will carefully consider your suggestion to include a brief explanation in the algorithmic description regarding how the theoretical value of $B$ can be specified.
>
> ---
>
> ### **Meaning of $\alpha$**
>
> The meaning of $\alpha$ is originally that of a **geometric margin**, since $\alpha$ is defined as the radius within which a good arm exists relative to the objective parameter (Assumption 3). To guarantee the existence of a good arm with respect to $\hat{\theta}_m$ after the initial exploration rounds, we set the theoretical value of $B$ so that the estimator $\hat{\theta}_m$ becomes $\alpha$-close to the true parameter $\theta_m^*$ with high probability. In this sense, $\alpha$ also plays the role of an endogenous estimation tolerance; however, this is better understood as a consequence of how $B$ is chosen, rather than as an inherent meaning of $\alpha$ itself.

---

> ### Author Response · Authors · 2025-11-22
> **Rebuttal by Authors [2/2]**
>
> ### **Lemma 2**
> Regarding the first two points (stochastic nature of this event or whether it holds), note that the condition "$\\|\hat{\theta}_m - \theta_m^*\\|_2 \le \alpha$ after $T_0$" appears in the **if-statement** of the Lemma. Therefore, questions about the stochastic nature of this event or whether it holds are **not** within the scope of what the Lemma is intended to establish.
>
> Moreover, the result of Lemma 2 does **not** lack rigor or clarity. The reason $\alpha$ does not explicitly appear in the theorem statement is due to the intricate dependency between $\alpha$ and $\lambda$. (Specifically, we set $\alpha$ so that it remains below the value $\psi(\lambda, \gamma)$.) The detailed explanation can be found in Appendix D (lines 1012–1020). Nevertheless, we are considering adding a brief clarification of this point in the main text as well.
>
> ---
>
> ### **$\texttt{MOG-R}$ Algorithm**
>
> First, we introduced $\texttt{MOG-R}$ as **the simplest randomized variant** of the standard round-robin version, rather than to address a specific drawback of the standard version of $\texttt{MOG}$. The $\texttt{MOG-R}$ algorithm  requires stochastic arguments to establish its regret bound, and it serves an expository purpose in the paper, providing a natural bridge before we introduce the subsequent $\texttt{MOG-WR}$ algorithm.
>
> Moreover, the $\texttt{MOG-R}$ algorithm can be applied in settings where the dominant objective follows a stochastic distribution. For example, in scenarios where the user changes in every round and the users’ prioritized objectives follow some underlying distribution, $\texttt{MOG-R}$ can be used by greedily selecting the arm that is best with respect to each user’s most prioritized objective.
>
> ---
>
>
> We also sincerely appreciate your additional comments regarding the presentation, and we will carefully take them into consideration when preparing our revision. Once again, thank you for your detailed feedback, and we hope that our responses have adequately addressed the concerns you raised.

---

> ### Comment · Reviewer_rDEZ · 2025-11-26
>
> **Free exploration:** I feel that the terminology is misleading. There is "forced / explicit exploration" in early rounds, and yes, $O(log T)$ rate suffices to have decent estimates. This seems natural. I do not prefer to call results stemming from an initial explicit exploration "free exploration".
>
> **Assumption 3:** I agree, that Cauchy-Schwarz yields a stronger condition (which was my initial thought). That said,  I still feel that the existence of $\gamma$ bounded away from $0$ is a strong (and rather unconventional) assumption.
>
> **Lemma 2:** I remain unclear about this point. The authors rebut: "Specifically, we set $\alpha$ so that it remains below the value $\psi(\lambda, \gamma)$...". This is ambiguous, as we do not know the structure of $\psi$ and whether or not this choice exists. Furthermore, if a choice exists, is it unique? If so, can we express $\lambda$ or $\gamma$ in terms of $\alpha$?
>
> I appreciate the author's efforts to help me understand various aspects of the paper. In its current form, I feel that the paper needs a careful restructuring / revising. Hence, I maintain my score.

---

### Note · Authors · 2025-12-04

**Comment:**

We sincerely appreciate the reviewer’s valuable comments, and we will incorporate the discussed points to improve the manuscript. In particular, we will strengthen the clarity and detail regarding the initial exploration phase and the arm-selection strategy. Nevertheless, we remain confident in the value of our central observation. Summarizing our key contributions:

- To the best of our knowledge, we are the first to discover and formalize the **implicit exploration** benefit that arises from the synergy between multi-objectivity and goodness. While prior multi-objective bandit algorithms often rely on complex exploration strategies, our work demonstrates that a simple and practical policy can achieve strong performance with provable regret guarantees.

- Unlike previous (single-objective) linear bandit studies on free exploration (e.g., Kannan et al., 2018; Raghavan et al., 2018; Bastani et al., 2021; Kim & Oh, 2025), which all assume distributional diversity in features, our work introduces a new exploration-free setting that operates **without** such assumptions.

- We propose the first formal **mathematical framework for fairness** in this domain and provide the first theoretical analysis of fairness guarantees in multi-objective bandit algorithms.

- We conduct **extensive numerical experiments** that validate our theoretical findings and demonstrate the practical effectiveness of our approach.

In short, our work is the first to address the previously overlooked phenomenon of implicit exploration induced by multi-objectivity, offering new insights for future work in this area. Furthermore, they are highly practical and fast in real-world recommendation scenarios where good arms are abundant and multiple objectives coexist. Given the significance of these contributions, we believe that our work offers meaningful insights, and we will further refine the manuscript so that these contributions are more clearly articulated.

**Withdrawal Confirmation:**

I have read and agree with the venue's withdrawal policy on behalf of myself and my co-authors.